# Direct extraction of signal and noise correlations from two-photon calcium imaging of ensemble neuronal activity

Anuththara Rupasinghe[1], Nikolas Francis[2,3], Ji Liu[2,3], Zac Bowen[2,3], Patrick O Kanold[2,3,4], Behtash Babadi[1]*

[1]Department of Electrical and Computer Engineering, University of Maryland, College Park, United States; [2]The Institute for Systems Research, University of Maryland, College Park, United States; [3]Department of Biology, University of Maryland, College Park, United States; [4]Department of Biomedical Engineering, Johns Hopkins University, Baltimore, United States

**Abstract** Neuronal activity correlations are key to understanding how populations of neurons collectively encode information. While two-photon calcium imaging has created a unique opportunity to record the activity of large populations of neurons, existing methods for inferring correlations from these data face several challenges. First, the observations of spiking activity produced by two-photon imaging are temporally blurred and noisy. Secondly, even if the spiking data were perfectly recovered via deconvolution, inferring network-level features from binary spiking data is a challenging task due to the non-linear relation of neuronal spiking to endogenous and exogenous inputs. In this work, we propose a methodology to explicitly model and directly estimate signal and noise correlations from two-photon fluorescence observations, without requiring intermediate spike deconvolution. We provide theoretical guarantees on the performance of the proposed estimator and demonstrate its utility through applications to simulated and experimentally recorded data from the mouse auditory cortex.

**\*For correspondence:**
behtash@umd.edu

**Competing interests:** The authors declare that no competing interests exist.

## Introduction

Neuronal activity correlations are essential in understanding how populations of neurons encode information. Correlations provide insights into the functional architecture and computations carried out by neuronal networks (*Abbott and Dayan, 1999*; *Averbeck et al., 2006*; *Cohen and Kohn, 2011*; *Hansen et al., 2012*; *Kohn et al., 2016*; *Kohn and Smith, 2005*; *Lyamzin et al., 2015*; *Montijn et al., 2014*; *Smith and Sommer, 2013*; *Sompolinsky et al., 2001*; *Yatsenko et al., 2015*). Neuronal activity correlations are often categorized in two groups: *signal* correlations and *noise* correlations (*Cohen and Kohn, 2011*; *Cohen and Maunsell, 2009*; *Gawne and Richmond, 1993*; *Josić et al., 2009*; *Lyamzin et al., 2015*; *Vinci et al., 2016*). Given two neurons, signal correlation quantifies the similarity of neural responses that are time-locked to a repeated stimulus across trials, whereas noise correlation quantifies the stimulus-independent trial-to-trial variability shared by neural responses that are believed to arise from common latent inputs.

Two-photon calcium imaging has become increasingly popular in recent years to record in vivo neural activity simultaneously from hundreds of neurons (*Ahrens et al., 2013*; *Romano et al., 2017*; *Stosiek et al., 2003*; *Svoboda and Yasuda, 2006*). This technology takes advantage of intracellular calcium flux mostly arising from spiking activity and captures calcium signaling in neurons in living animals using fluorescence microscopy. The observed fluorescence traces of calcium concentrations, however, are indirectly related to neuronal spiking activity. Extracting spiking activity from fluorescence traces is a challenging signal deconvolution problem and has been the focus of active research

(*Deneux et al., 2016*; *Friedrich et al., 2017*; *Grewe et al., 2010*; *Jewell et al., 2020*; *Jewell and Witten, 2018*; *Kazemipour et al., 2018*; *Pachitariu et al., 2018*; *Pnevmatikakis et al., 2016*; *Stringer and Pachitariu, 2019*; *Theis et al., 2016*; *Vogelstein et al., 2009*; *Vogelstein et al., 2010*).

The most commonly used approach to infer signal and noise correlations from two-photon data is to directly apply the classical definitions of correlations for firing rates (*Lyamzin et al., 2015*), to fluorescence traces (*Fallani et al., 2015*; *Francis et al., 2018*; *Rothschild et al., 2010*; *Winkowski and Kanold, 2013*). However, it is well known that fluorescence observations are noisy and blurred surrogates of spiking activity, because of dependence on observation noise, calcium dynamics and the temporal properties of calcium indicators. Due to temporal blurring, the resulting signal and noise correlation estimates are highly biased. An alternative approach is to carry out the inference in a two-stage fashion: first, infer spikes using a deconvolution technique, and then compute firing rates and evaluate the correlations (*Kerlin et al., 2019*; *Najafi et al., 2020*; *Ramesh et al., 2018*; *Soudry et al., 2015*; *Yatsenko et al., 2015*). These two-stage estimates are highly sensitive to the accuracy of spike deconvolution, and require high temporal resolution and signal-to-noise ratios (*Lütcke et al., 2013*; *Pachitariu et al., 2018*). Furthermore, these deconvolution techniques are biased toward obtaining accurate first-order statistics (i.e. spike timings) via spatiotemporal priors, which may be detrimental to recovering second-order statistics (i.e. correlations). Finally, both approaches also undermine the non-linear dynamics of spiking activity as governed by stimuli, past activity and other latent processes (*Truccolo et al., 2005*). There are a few existing studies that aim at improving estimation of neuronal correlations, but they either do not consider signal correlations (*Rupasinghe and Babadi, 2020*; *Yatsenko et al., 2015*), or aim at estimating surrogates of correlations from spikes such as the connectivity/coupling matrix (*Aitchison et al., 2017*; *Mishchenko et al., 2011*; *Soudry et al., 2015*; *Keeley et al., 2020*).

Here, we propose a methodology to *directly* estimate both signal and noise correlations from two-photon imaging observations, without requiring an intermediate step of spike deconvolution. We pose the problem under the commonly used experimental paradigm in which neuronal activity is recorded during trials of a repeated stimulus. We avoid the need to perform spike deconvolution by integrating techniques from point processes and state-space modeling that explicitly relate the signal and noise correlations to the observed fluorescence traces in a multi-tier model. Thus, we cast signal and noise correlations within a parameter estimation setting. To solve the resulting estimation problem in an efficient fashion, we develop a solution method based on variational inference (*Jordan et al., 1999*; *Blei et al., 2017*), by combining techniques from Pólya-Gamma augmentation (*Polson et al., 2013*) and compressible state-space estimation (*Rauch et al., 1965*; *Kazemipour et al., 2018*; *Ba et al., 2014*). We also provide theoretical guarantees on the bias and variance performance of the resulting estimator.

We demonstrate the utility of our proposed estimation framework through application to simulated and real data from the mouse auditory cortex during presentations of tones and acoustic noise. In application to repeated trials under spontaneous and stimulus-driven conditions within the same experiment, our method reliably provides noise correlation structures that are invariant across the two conditions. In addition, our joint analysis of signal and noise correlations corroborates existing hypotheses regarding the distinction between their structures (*Keeley et al., 2020*; *Rumyantsev et al., 2020*; *Bartolo et al., 2020*). Moreover, while application of our proposed method to spatial analysis of signal and noise correlations in the mouse auditory cortex is consistent with existing work (*Winkowski and Kanold, 2013*), it reveals novel and distinct spatial trends in the correlation structure of layers 2/3 and 4. In summary, our method improves on existing work by: (1) explicitly modeling the fluorescence observation process and the non-linearities involved in spiking activity, as governed by both the stimulus and latent processes, through a multi-tier Bayesian forward model, (2) joint estimation of signal and noise correlations directly from two-photon fluorescence observations through an efficient iterative procedure, without requiring intermediate spike deconvolution, (3) providing theoretical guarantees on the performance of the proposed estimator, and (4) gaining access to closed-form posterior approximations, with low-complexity and iterative update rules and minimal dependence on training data. Our proposed method can thus be used as a robust and scalable alternative to existing approaches for extracting signal and noise correlations from two-photon imaging data.

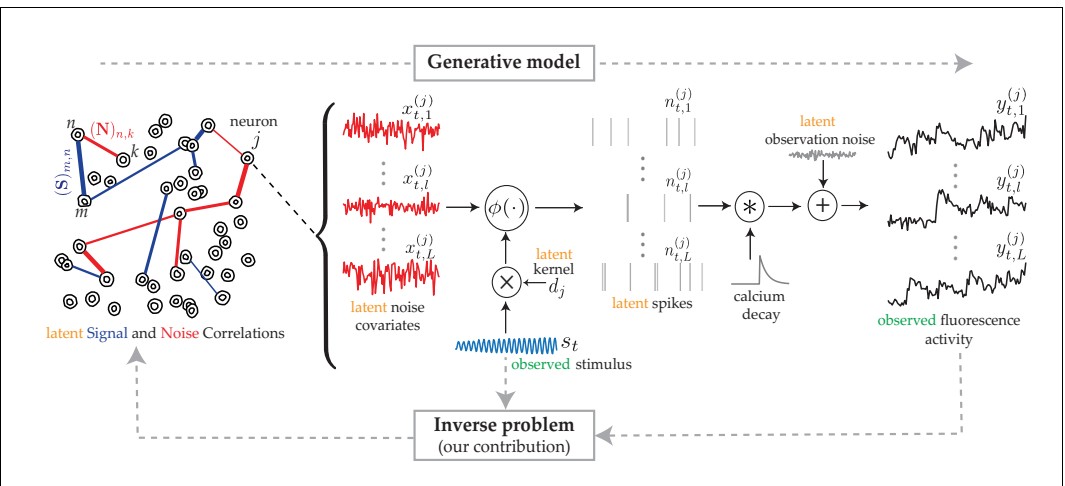

**Figure 1.** The proposed generative model and inverse problem. Observed (green) and latent (orange) variables pertinent to the $j^{th}$ neuron are indicated, according to the proposed model for estimating the signal (blue) and noise (red) correlations from two-photon calcium fluorescence observations. Calcium fluorescence traces $\left(y_{t,l}^{(j)}\right)$ of $L$ trials are observed, in which the repeated external stimulus ($s_t$) is known. The underlying spiking activity $\left(n_{t,l}^{(j)}\right)$, trial-to-trial variability and other intrinsic/extrinsic neural covariates that are not time-locked with the external stimulus $\left(x_{t,l}^{(j)}\right)$, and the stimulus kernel ($\mathbf{d}_j$) are latent. Our main contribution is to solve the inverse problem: recovering the underlying latent signal ($\mathbf{S}$) and noise ($\mathbf{N}$) correlations directly from the fluorescence observations, without requiring intermediate spike deconvolution.

## Results

In this section, we first demonstrate the utility of our proposed estimation framework through simulation studies as well as applications on experimentally recorded data from the mouse auditory cortex. Then, we present theoretical performance bounds on the proposed estimator. Before presenting the results, we will give an overview of the proposed signal and noise correlation inference framework, and outline our contributions and their relationship to existing work. For the ease of reproducibility, we have archived a MATLAB implementation of our proposed method in GitHub (*Rupasinghe, 2020*) and have deposited the data used in this work in the Digital Repository at the University of Maryland (*Rupasinghe et al., 2021*).

### Signal and noise correlations

We consider a canonical experimental setting in which the same external stimulus, denoted by $\mathbf{s}_t$, is repeatedly presented across $L$ independent trials and the spiking activity of a population of $N$ neurons are indirectly measured using two-photon calcium fluorescence imaging. *Figure 1* (forward arrow) shows the generative model that is used to quantify this procedure. The fluorescence observation in the $l^{th}$ trial from the $j^{th}$ neuron at time frame $t$, denoted by $y_{t,l}^{(j)}$, is a noisy surrogate of the intracellular calcium concentrations. The calcium concentrations in turn are temporally blurred surrogates of the underlying spiking activity $n_{t,l}^{(j)}$, as shown in *Figure 1*.

In modeling the spiking activity, we consider two main contributions: (1) the common known stimulus $\mathbf{s}_t$ affects the activity of the $j^{th}$ neuron via an unknown kernel $\mathbf{d}_j$, akin to the receptive field; (2) the trial-to-trial variability and other intrinsic/extrinsic neural covariates that are not time-locked to the stimulus $\mathbf{s}_t$ are captured by a trial-dependent latent process $x_{t,l}^{(j)}$. Then, we use a Generalized Linear Model to link these underlying neural covariates to spiking activity (*Truccolo et al., 2005*). More specifically, we model spiking activity as a Bernoulli process:

$$n_{t,l}^{(j)} \sim Bernoulli\left(\phi\left(x_{t,l}^{(j)}, \mathbf{d}_j^\top \mathbf{s}_t\right)\right),$$

where $\phi(\cdot)$ is a mapping function, which could in general be non-linear.

The *signal* correlations aim to measure the correlations in the temporal response that are time-locked to the repeated stimulus, $\mathbf{s}_t$. On the other hand, *noise* correlations in our setting quantify connectivity arising from covariates that are unrelated to the stimulus, including the trial-to-trial variability (*Keeley et al., 2020*). Based on the foregoing model, we propose to formulate the signal $\left((\mathbf{\Sigma}_s)_{i,j}\right)$ and noise $\left((\mathbf{\Sigma}_x)_{i,j}\right)$ covariance between the $i^{th}$ neuron and $j^{th}$ neuron as:

$$(\mathbf{\Sigma}_s)_{i,j} := \mathbf{d}_i^\top cov(\mathbf{s}_t, \mathbf{s}_t)\mathbf{d}_j, \quad (\mathbf{\Sigma}_x)_{i,j} := cov\left(x_{t,l}^{(i)}, x_{t,l}^{(j)}\right), \tag{1}$$

where $cov(\cdot, \cdot)$ is the empirical covariance function defined for two vector time series $\mathbf{u}_t$ and $\mathbf{v}_t$ as $cov(\mathbf{u}_t, \mathbf{v}_t) := \frac{1}{T}\sum_{t=1}^{T}\left(\mathbf{u}_t - \frac{1}{T}\sum_{t'=1}^{T}\mathbf{u}_{t'}\right)\left(\mathbf{v}_t - \frac{1}{T}\sum_{t'=1}^{T}\mathbf{v}_{t'}\right)^\top$, for a total observation duration of $T$ time frames.

Our main contribution is to provide an efficient solution for the so-called inverse problem: direct estimation of $\mathbf{\Sigma}_s$ and $\mathbf{\Sigma}_x$ from the fluorescence observations, without requiring intermediate spike deconvolution (*Figure 1*, backward arrow). The signal and noise correlation matrices, denoted by $\mathbf{S}$ and $\mathbf{N}$, can then be obtained by standard normalization of $\mathbf{\Sigma}_s$ and $\mathbf{\Sigma}_x$:

$$(\mathbf{S})_{i,j} := \frac{(\mathbf{\Sigma}_s)_{i,j}}{\sqrt{(\mathbf{\Sigma}_s)_{i,i}\cdot(\mathbf{\Sigma}_s)_{j,j}}}, \quad (\mathbf{N})_{i,j} := \frac{(\mathbf{\Sigma}_x)_{i,j}}{\sqrt{(\mathbf{\Sigma}_x)_{i,i}\cdot(\mathbf{\Sigma}_x)_{j,j}}}, \quad \forall i,j = 1, 2, \cdots, N. \tag{2}$$

We note that when spiking activity is directly observed using electrophysiology recordings, the conventional signal $\left((\mathbf{\Sigma}_s^{con})_{i,j}\right)$ and noise $\left((\mathbf{\Sigma}_x^{con})_{i,j}\right)$ covariances of spiking activity between the $i^{th}$ and $j^{th}$ neuron are defined as (*Lyamzin et al., 2015*):

$$(\mathbf{\Sigma}_s^{con})_{i,j} := cov\left(\frac{1}{L}\sum_{l=1}^{L}n_{t,l}^{(i)}, \frac{1}{L}\sum_{l=1}^{L}n_{t,l}^{(j)}\right), \quad (\mathbf{\Sigma}_x^{con})_{i,j} := \frac{1}{L}\sum_{l=1}^{L}cov\left(n_{t,l}^{(i)} - \frac{1}{L}\sum_{l'=1}^{L}n_{t,l'}^{(i)}, n_{t,l}^{(j)} - \frac{1}{L}\sum_{l'=1}^{L}n_{t,l'}^{(j)}\right), \tag{3}$$

which after standard normalization in *Equation 2* give the conventional signal $\left((\mathbf{S}^{con})_{i,j}\right)$ and noise $\left((\mathbf{N}^{con})_{i,j}\right)$ correlations. While at first glance, our definitions of signal and noise covariances in *Equation 1* seem to be a far departure from the conventional ones in *Equation 3*, we show that the conventional notions of correlation indeed approximate the same quantities as in our definitions:

$$\mathbf{S}^{con} \approx \mathbf{S} \quad \text{and} \quad \mathbf{N}^{con} \approx \mathbf{N},$$

under asymptotic conditions (i.e. $T$ and $L$ sufficiently large). We prove this assertion of asymptotic equivalence in Appendix 1, which highlights another facet of our contributions: our proposed estimators are designed to robustly operate in the regime of finite (and typically small) $T$ and $L$, aiming for the very same quantities that the conventional estimators could only recover accurately under ideal asymptotic conditions.

## Existing methods used for performance comparison

In order to compare the performance of our proposed method with existing work, we consider three widely available methods for extracting neuronal correlations. In simulation studies, we additionally benchmark these estimates with respect to the known ground truth. The existing methods considered are the following:

> Pearson correlations from the two-photon data
> In this method, fluorescence observations are assumed to be the direct measurements of spiking activity, and thus empirical Pearson correlations of the two-photon data are used to compute the signal and noise correlations (*Rothschild et al., 2010*; *Winkowski and Kanold, 2013*; *Francis et al., 2018*; *Bowen et al., 2020*). Explicitly, these estimates are obtained by simply replacing $n_{t,l}^{(j)}$ in *Equation 3* by $y_{t,l}^{(j)}$, without performing spike deconvolution.
> Two-stage Pearson estimation

Unlike the previous method, in this case spikes are first inferred using a deconvolution technique. Then, following temporal smoothing via a narrow Gaussian kernel the Pearson correlations are computed using the conventional definitions of *Equation 3*. For spike deconvolution, we primarily used the FCSS algorithm (*Kazemipour et al., 2018*). In order to also demonstrate the sensitivity of these estimates to the deconvolution technique that is used, we provide a comparison with the f-oopsi deconvolution algorithm (*Pnevmatikakis et al., 2016*) in *Figure 2—figure supplement 1*.

Two-stage GPFA estimation

Similar to the previous method, spikes are first inferred using a deconvolution technique. Then, a latent variable model called Gaussian Process Factor Analysis (GPFA) (*Yu et al., 2009*) is applied to the inferred spikes in order to estimate the latent covariates and receptive fields. Based on those estimates, the signal and residual noise correlations are derived through a formulation similar to *Equation 1* and *Equation 2* (*Ecker et al., 2014*).

## Simulation study 1: neuronal ensemble driven by external stimulus

We simulated calcium fluorescence observations according to the proposed generative model given in Proposed forward model, from an ensemble of $N = 8$ neurons for a duration of $T = 5000$ time frames. We considered $L = 20$ repeated trials driven by the same external stimulus, which we modeled by an autoregressive process (see Guidelines for model parameter settings for details). *Figure 2* shows the corresponding estimation results.

The first column of *Figure 2A* shows the ground truth noise (top) and signal (bottom) correlations (diagonal elements are all equal to one and omitted for visual convenience). The second column shows estimates of the noise and signal correlations using our proposed method, which closely match the ground truth. The third, fourth and fifth columns, respectively, show the results of the Pearson correlations from the two-photon data, two-stage Pearson, and two-stage GPFA estimation

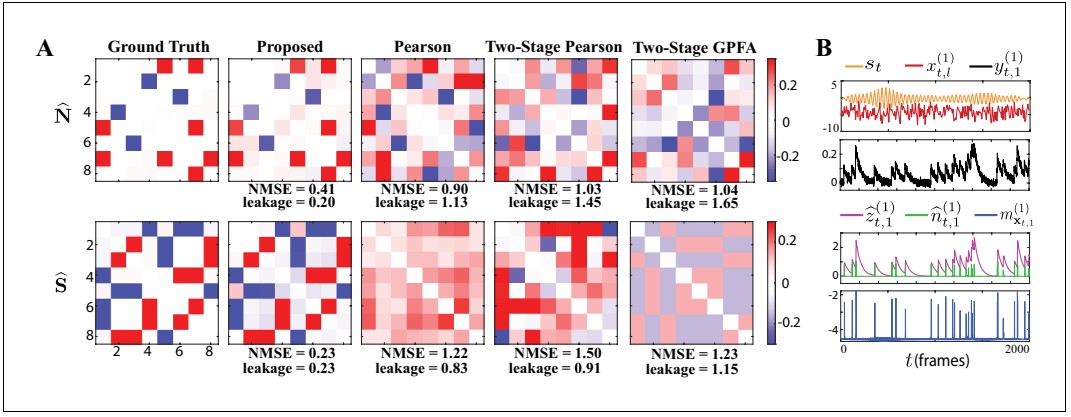

**Figure 2.** Results of simulation study 1. (**A**) Estimated noise and signal correlation matrices from different methods. Rows from left to right: ground truth, proposed method, Pearson correlations from two-photon recordings, two-stage Pearson estimates and two-stage GPFA estimates. The normalized mean squared error (NMSE) of each estimate with respect to the ground truth and the leakage effect quantified by the ratio between out-of-network and in-network power (leakage) are indicated below each panel. (**B**) Simulated external stimulus (orange), latent trial-dependent process (red), fluorescence observations (black), estimated calcium concentrations (purple), putative spikes (green), and estimated mean of the latent state (blue) by the proposed method, for the first trial of neuron 1.

The online version of this article includes the following figure supplement(s) for figure 2:

**Figure supplement 1.** Sensitivity of two-stage estimates to the choice of the underlying spike deconvolution technique.

**Figure supplement 2.** Performance of two-stage estimates based on ground truth spikes.

**Figure supplement 3.** Performance comparison under stimulus integration model mismatch.

**Figure supplement 4.** Performance under calcium decay model mismatch.

**Figure supplement 5.** Performance comparison under varying SNR levels and firing rates.

**Figure supplement 6.** Performance comparison under observation noise model mismatch.

methods. Through a qualitative visual inspection, it is evident that these methods incur high false alarms and mis-detections of the ground truth correlations.

To quantify these comparisons, the normalized mean square error (NMSE) of different estimates with respect to the ground truth are shown below each of the subplots (*Figure 2A*). Our proposed method achieves the lowest NMSE compared to the others. Furthermore, we observed a significant mixing between signal and noise correlations in these other estimates. To quantify this leakage effect, we first classified each of the correlation entries as in-network or out-of-network, based on being non-zero or zero in the ground truth, respectively (see Performance evaluation). We then computed the ratio between the power of out-of-network components and the power of in-network components as a measure of leakage. The leakage ratios are also reported in *Figure 2A*. The leakage of our proposed estimates is the lowest of all four techniques, in estimating both the signal and noise correlations. In order to further probe the performance of our proposed method, the simulated external stimulus $s_t$, latent trial-dependent process $x_{t,1}^{(1)}$, simulated observations $y_{t,1}^{(1)}$, estimated calcium concentration $\widehat{z}_{t,1}^{(1)}$, the putative spikes $\widehat{n}_{t,1}^{(1)} := \widehat{z}_{t,1}^{(1)} - \alpha\widehat{z}_{t-1,1}^{(1)}$, and the estimated mean of the latent state $m_{\mathbf{x}_{t,1}}^{(1)}$, for the first trial of the first neuron are shown in *Figure 2B*. These results demonstrate the ability of the proposed estimation framework in accurately identifying the latent processes, which in turn leads to an accurate estimation of the signal and noise correlations as shown in *Figure 2B*.

The main sources of the observed performance gap between our proposed method and the existing ones are the bias incurred by treating the fluorescence traces as spikes, low spiking rates, non-linearity of spike generation with respect to intrinsic and external covariates, and sensitivity to spike deconvolution. For the latter, we demonstrated the sensitivity of the two-stage Pearson estimates to the choice of the deconvolution technique in *Figure 2—figure supplement 1*. Furthermore, in order to isolate the effect of said non-linearities on the estimation performance, we applied the two-stage methods to ground truth spikes in *Figure 2—figure supplement 2*. Our analysis showed that both two-stage estimates incur significant estimation errors even if the spikes were recovered perfectly, mainly due to the limited number of trials ($L = 20$ here). In accordance with our theoretical analysis of the asymptotic behavior of the conventional signal and noise correlation estimates given in Appendix 1, we also showed in *Figure 2—figure supplement 2* that the performance of the two-stage Pearson estimates based on ground truth spikes, but using $L = 1000$ trials, dramatically improves. Our proposed method, however, was capable of producing reliable estimates with the number of trials as low as $L = 20$, which is typical in two-photon imaging experiments.

## Analysis of robustness with respect to modeling assumptions

While the preceding results are quite favorable to our proposed method, the underlying generative models were the same as those used to estimate signal and noise correlations, which is in contrast to conventional real data validation with known ground truth. Access to ground truth correlations in two-photon imaging experimental settings, however, is quite challenging. In order to further probe the robustness of our proposed method in the absence of ground truth data, we utilized surrogate data that parallel the setting of *Figure 2*, but deviate from our modeling assumptions.

1. Robustness to stimulus integration model mismatch. First, we considered surrogate data generated with a non-linear stimulus integration model by replacing the linear receptive field component $\mathbf{d}_j^\top\mathbf{s}_t$ with $\mathbf{d}_j^\top\mathbf{s}_t + (\widetilde{\mathbf{d}}_{j,1}^\top\mathbf{s}_t)^2 + (\widetilde{\mathbf{d}}_{j,2}^\top\mathbf{s}_t)^2$, where $\widetilde{\mathbf{d}}_{j,1}$ and $\widetilde{\mathbf{d}}_{j,2}$ are akin to *quadratic* receptive field components. We assumed a linear stimulus integration model in our estimation framework (i.e., $\widetilde{\mathbf{d}}_{j,1} = \widetilde{\mathbf{d}}_{j,2} = \mathbf{0}$). *Figure 2—figure supplement 3* shows the resulting correlation estimates. While the performance of our proposed signal correlation estimates degrade under this setting as compared to *Figure 2*, our proposed estimates still outperform existing methods. In addition, the model mismatch in the stimulus integration component does not affect the accuracy of noise correlation estimation in our method.

2. Robustness to calcium decay model mismatch. Next, we tested our proposed estimation framework on data simulated with a different calcium decay model. Specifically, we simulated data with second-order autoregressive calcium dynamics, and at a lower signal-to-noise ratio (SNR) compared to the setting of *Figure 2*, and used our inference framework which assumes first order calcium dynamics for estimation. *Figure 2—figure supplement 4* shows the corresponding noise and signal correlations estimated by the proposed method under these

conditions. Even though the performance slightly degrades (in terms of NMSE and leakage ratio), our method is able to recover the underlying correlations faithfully under this setting.

3. Robustness to SNR level and firing rate. Next, we compared the performance of Pearson and Two-Stage Pearson methods with our proposed method under varying SNR levels and average firing rates, as shown in *Figure 2—figure supplement 5*. While the performance of all methods degrades at low SNR levels or firing rates (SNR < 10 dB, firing rate < 0.5 Hz), our proposed method outperforms the existing methods for a wide range of SNR and firing rate values. To quantify this comparison, we have also indicated the mean and standard deviation of the relative performance gain of our proposed estimates across SNR levels and firing rates as insets in *Figure 2—figure supplement 5*.

4. Robustness to observation noise model mismatch. Finally, we repeated the foregoing comparisons under varying SNR levels and firing rates, only now we included an additional observation noise model mismatch. Similar to the treatment in *Deneux et al., 2016*, we considered two temporally correlated observation noise models: white noise with a low frequency drift (*Figure 2—figure supplement 6*, top panels) and pink noise (*Figure 2—figure supplement 6*, bottom panels). In accordance with the results in *Figure 2—figure supplement 5*, our proposed method outperforms the existing ones for a wide range of SNR and firing rate values and under both observation noise model mismatch conditions. From *Figure 2—figure supplement 6C and F*, it can be observed that the ground truth spikes are favorably recovered as a byproduct of our method, even though the estimated calcium concentrations are contaminated by the temporally correlated fluctuations in observation noise. This in turn results in accurate signal and noise correlation estimates.

## Simulation study 2: spontaneous activity

Next, we present the results of a simulation study in the absence of external stimuli (i.e. $\mathbf{s}_t = \mathbf{0}$), pertaining to the spontaneous activity condition. It is noteworthy that the proposed method can readily be applied to estimate noise correlations during spontaneous activity, by simply setting the external stimulus $\mathbf{s}_t$ and the receptive field $\mathbf{d}_j$ to zero in the update rules (see Proposed forward model for details). We simulated the ensemble spiking activity based on a Poisson process (*Smith and Brown, 2003*) using a discrete time-rescaling procedure (*Brown et al., 2002*; *Smith and Brown, 2003*), so that the data are generated using a different model than that used in our inference framework (i.e. Bernoulli process with a logistic link as outlined in Proposed forward model). As such, we eliminated potential performance biases in favor of our proposed method by introducing the aforementioned model mismatch. We simulated $L = 20$ independent trials of spontaneous activity of $N = 30$ neurons, observed for a time duration of $T = 5000$ time frames. The number of neurons in this study is notably larger than that used in the previous one, to examine the scalability of our proposed approach with respect to the ensemble size.

*Figure 3* shows the comparison of the noise correlation matrices estimated by our proposed method, Pearson correlations from two-photon recordings, two-stage Pearson, and two-stage GPFA estimates, with respect to the ground truth. The Pearson and the two-stage estimates are highly

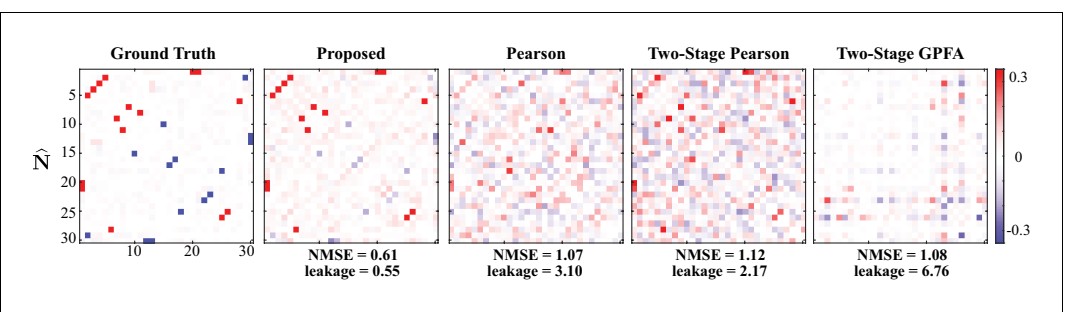

**Figure 3.** Results of simulation study 2. Estimated noise correlation matrices using different methods based from spontaneous activity data. Rows from left to right: ground truth, proposed method, Pearson correlations from two-photon recordings, two-stage Pearson and two-stage GPFA estimates. The normalized mean squared error (NMSE) of each estimate with respect to the ground truth and the ratio between out-of-network power and in-network power (leakage) are shown below each panel.

variable and result in excessive false detections. Our proposed estimate, however, closely follows the ground truth, which is also reflected by the comparatively lower NMSE and leakage ratios, in spite of the mismatch between the models used for data generation and inference. In addition, our proposed method exhibits favorable scaling with respect to the ensemble size, thanks to the underlying low-complexity variational updates (see Low-complexity parameter updates for details).

## Real data study 1: mouse auditory cortex under random tone presentation

We next applied our proposed method to experimentally recorded two-photon observations from the mouse primary auditory cortex (A1). The dataset consisted of recordings from 371 excitatory neurons in layer 2/3 A1, from which we selected $N = 16$ responsive neurons (i.e. neurons that exhibited at least one spiking event in at least half of the trials considered; see Guidelines for model parameter settings). A random sequence of four tones was presented to the mouse, with the same sequence being repeated for $L = 10$ trials. Each trial consisted of $T = 3600$ time frames, and each tone was 2 s long followed by a 4 s silent period (see Experimental procedures for details). We considered an integration window of $R = 25$ frames for stimulus encoding (see Guidelines for model parameter settings for details). The comparison of the noise and signal correlation estimates obtained by our proposed method, Pearson correlations from two-photon recordings, two-stage Pearson and two-stage GPFA methods is shown in *Figure 4A*. The spatial map of the 16 neurons considered in the analysis in the field of view is shown in *Figure 4B*. *Figure 4C* shows the stimulus tone sequence $s_t$ , two-photon observations $y_{t,1}^{(1)}$ , estimated calcium concentration $\widehat{z}_{t,1}^{(1)}$ , putative spikes $\widehat{n}_{t,1}^{(1)} := \widehat{z}_{t,1}^{(1)} - \alpha \widehat{z}_{t-1,1}^{(1)}$ and the estimated mean of the latent state $m_{\mathbf{x},1}^{(1)}$ , for the first trial of the first neuron.

We estimated the Best Frequency (BF) of each neuron as the tone that resulted in the highest level of fluorescence activity. The results in *Figure 4A* are organized such that the neurons with the same BF are neighboring, with the BF increasing along the diagonal. Thus, expectedly (*Bowen et al., 2020*) our proposed method as well as the Pearson and two-stage Pearson estimates show high signal correlations along the diagonal. However, the two-stage GPFA estimates do not reveal such a structure. By visual inspection, as also observed in the simulation studies, the Pearson correlations from two-photon recordings, two-stage Pearson and two-stage GPFA estimates have significant leakage between the signal and noise correlations, whereas our proposed signal and noise correlation estimates in *Figure 4A* suggest distinct spatial structures.

To quantify this visual comparison, we used a statistic based on the Tanimoto similarity metric (*Lipkus, 1999*), denoted by $T_s(\mathbf{X}, \mathbf{Y})$ for two matrices $\mathbf{X}$ and $\mathbf{Y}$. As a measure of dissimilarity, we used $T_d(\mathbf{X}, \mathbf{Y}) := 1 - T_s(\mathbf{X}, \mathbf{Y})$ (see Performance evaluation for details). The comparison of $T_d(\widehat{\mathbf{S}}, \widehat{\mathbf{N}})$ for the four estimates is presented in the second column of *Table 1*. To assess statistical significance, for each comparison we obtained null distributions corresponding to chance occurrence of dissimilarities using a shuffling procedure as shown in *Figure 4D*, and then computed one-tailed $p$-values from those distributions (see Performance evaluation for details). *Table 1* and *Figure 4D* includes these p-values, which show that the proposed estimates (boldface numbers in *Table 1*, second column) indeed have the highest dissimilarity between signal and noise correlations. The higher leakage effect in the other three estimates is also reflected in their smaller $T_d(\widehat{\mathbf{S}}, \widehat{\mathbf{N}})$ values.

To further investigate this effect, we have depicted the scatter plots of signal vs. noise correlations estimated by each method in *Figure 4E*. To examine the possibility of the leakage effect on a pairwise basis, we performed linear regression in each case. The slope of the model fit, the p-value for the corresponding t-test, and the $R^2$ values are reported in the third and fourth columns of *Table 1* (the slope and p-values are also shown as insets in *Figure 4E*). Consistent with the results of *Winkowski and Kanold, 2013*, the Pearson estimates suggest a significant correlation between the signal and noise correlation pairs (as indicated by the higher slope in *Figure 4E*). However, none of the other estimates (including the proposed estimates) in *Figure 4E* register a significant trend between signal and noise correlations. This further corroborates our assessment of the high leakage between signal and noise correlations in Pearson estimates, since such a leakage effect could result in overestimation of the trend between the signal and noise correlation pairs. The signal and noise correlations estimated by our proposed method show no pairwise trend, suggesting distinct

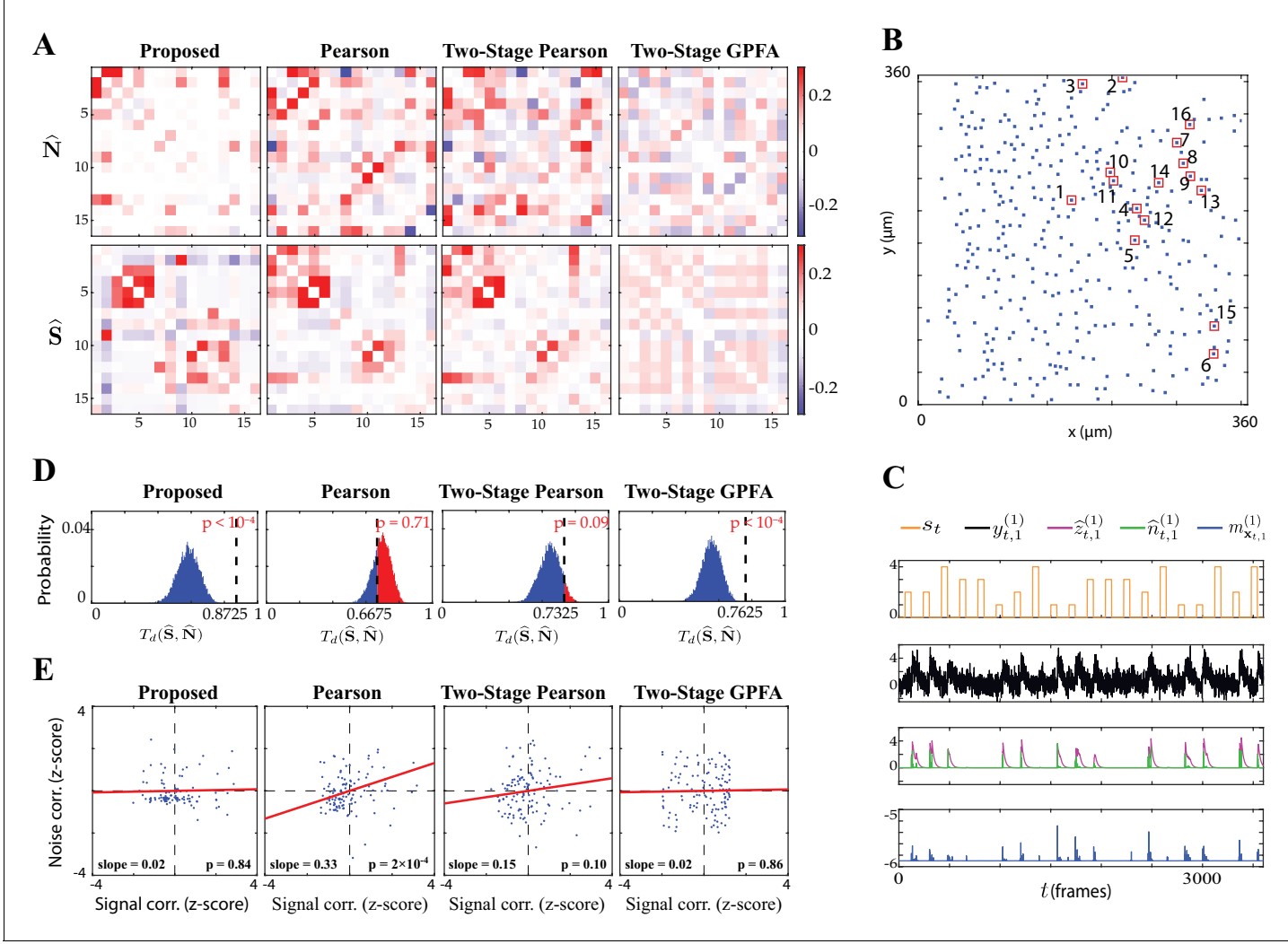

**Figure 4.** Application to experimentally-recorded data from the mouse A1. (**A**) Estimated noise (top) and signal (bottom) correlation matrices using different methods. Rows from left to right: proposed method, Pearson correlations from two-photon data, two-stage Pearson and two-stage GPFA estimates. (**B**) Location of the selected neurons with the highest activity in the field of view. (**C**) Presented tone sequence (orange), observations (black), estimated calcium concentrations (purple), putative spikes (green) and estimated mean latent state (blue) in the first trial of the first neuron. (**D**) Null distributions of chance occurrence of dissimilarities between signal and noise correlation estimates using different methods. The observed test statistic in each case is indicated by a dashed vertical line. (**E**) Scatter plots of signal vs. noise correlations for individual cell pairs (blue dots) corresponding to each method. Data were normalized for comparison by computing z-scores. For each case, the linear regression model fit is shown in red, and the slope and p-value of the t-test are indicated as insets.

The online version of this article includes the following figure supplement(s) for figure 4:

**Figure supplement 1.** Probing the effect of stimulus integration window length on the performance of the proposed estimates.

**Figure supplement 2.** Inspecting the inferred latent processes under high fluorescence activity due to rapid increase in firing rate.

patterns of stimulus-dependent and stimulus-independent functional connectivity (**Kohn et al., 2016**; **Montijn et al., 2014**; **Rothschild et al., 2010**; **Keeley et al., 2020**).

A key advantage of our proposed method over the Pearson and two-stage approaches is the explicit modeling of stimulus integration. The relevant parameter in this regard is the length of the stimulus integration window $R$. While in our simulation studies the value of $R$ was known, it needs to be set by the user in real data applications. To this end, domain knowledge or data-driven methods such as cross-validation and model order selection can be utilized (see Guidelines for model parameter settings for details). Noting that the number of parameters to be estimated linearly scales with $R$, it must be chosen large enough to capture the stimulus effects, yet small enough to result in

**Table 1.** Dissimilarity metric statistics for the estimates in *Figure 4A* (also illustrated in *Figure 4D*), linear regression statistics of the comparison between signal and noise correlations in *Figure 4E*, and the average NMSE across 50 trials used in the shuffling procedure illustrated in *Figure 5A*.

| Estimate | Dissimilarity $T_d(\widehat{S}, \widehat{N})$ Figure 4D | Regression statistics (*Figure 4E*) Slope (p-value) | $R^2$ Value | Shuffling test (*Figure 5*) NMSE in $\widehat{N}$ | NMSE in $\widehat{S}$ |
|---|---|---|---|---|---|
| Proposed | **0.8725** $(p<10^{-4})$ | 0.02 $(p = 0.84)$ | $4 \times 10^{-4}$ | **1.07 ± 0.16** | **1.32 ± 0.19** |
| Pearson | 0.6675 $(p = 0.71)$ | 0.33 $(p = 2 \times 10^{-4})$ | 0.11 | 0 | 0 |
| Two-stage Pearson | 0.7325 $(p = 0.09)$ | 0.15 $(p = 0.10)$ | 0.02 | 1.84 ± 0.34 | 0.55 ± 0.12 |
| Two-stage GPFA | 0.7625 $(p<10^{-4})$ | 0.02 $(p = 0.86)$ | $3 \times 10^{-4}$ | 2.32 ± 0.52 | 2.26 ± 0.51 |

favorable computational complexity. Here, given that the typical tone response duration of mouse A1 neurons is <1 s (*Linden et al., 2003*; *DeWeese et al., 2003*; *Petrus et al., 2014*), with a sampling frequency of $f_s = 30$ Hz, we surmised that a choice of $R \sim 30$ suffices to capture the stimulus effects. We further examined the effect of varying $R$ on the proposed correlation estimates in *Figure 4—figure supplement 1*. As shown, small values of $R$ (e.g. $R = 1$ or 10) may not be adequate to fully capture stimulus integration effects. By considering values of $R$ in the range $25 - 50$, we observed that the correlation estimates remain stable. We thus chose $R = 25$ for our analysis.

Careful inspection of the second panel in *Figure 4C* shows that the fluorescence activity often saturates to ~4 times its baseline value. This effect is due to successive closely spaced spikes, which implies the occurrence of more than one spike per frame and thus violates our Bernoulli modeling assumption. To inspect the performance of our method more carefully under this scenario, we show in *Figure 4—figure supplement 2* a zoomed-in view of the estimated latent processes $\widehat{z}_{t,1}^{(1)}$ (calcium concentration) and $\widehat{n}_{t,1}^{(1)}$ (putative spikes) for a sample data segment with high fluorescence activity. The estimated latent processes reveal two mechanisms leveraged by our inference method to mitigate the aforementioned model mismatch: first, our proposed method predicts spiking events in adjacent time frames to compensate for rapid increase in firing rate and thus infers calcium concentration levels that match the observed fluorescence; secondly, even though our generative model

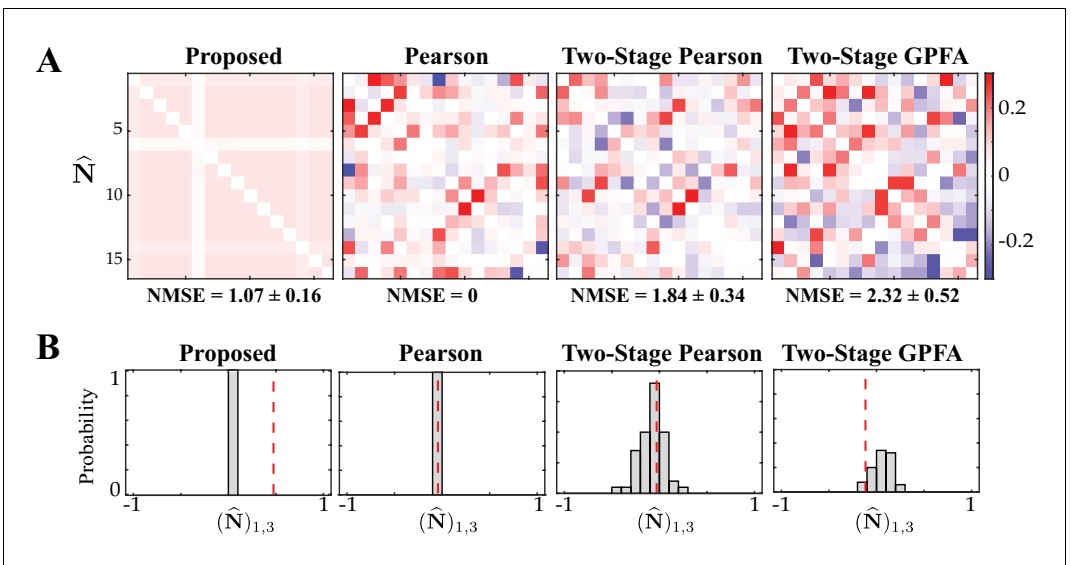

**Figure 5.** Assessing the specificity of different estimation results shown in *Figure 4*. Rows from left to right: proposed method, Pearson correlations from two-photon data, two-stage Pearson and two-stage GPFA estimates. (**A**) The estimated noise correlations using different methods after random temporal shuffling of the observations. The mean and standard deviation of the NMSE across 50 trials are indicated below each panel. (**B**) Histograms of the noise correlation estimates between the first and third neurons over the 50 temporal shuffling trials. The estimate based on the original (un-shuffled) data in each case is indicated by a dashed vertical line.

assumes that there is only one spiking event in a given time frame, this restriction is mitigated in our inference framework by relaxing the constraint $\widehat{n}_{t,l}^{(j)} := \widehat{z}_{t,l}^{(j)} - \alpha \widehat{z}_{t-1,l}^{(j)} \leq 1$, as explained in Low-complexity parameter updates. While this relaxation was performed for the sake of tractability of the inverse solution, it in fact leads to improved estimation results under episodes of rapid increase in firing rate, by allowing the putative spike magnitudes $\widehat{n}_{t,l}^{(j)}$ to be greater than 1. The latter is evident in the magnitude of the inferred spikes in *Figure 4—figure supplement 2*, following the rise of fluorescence activity.

Given that the ground truth correlations are not available for a direct comparison, we instead performed a test of specificity that reveals another key limitation of existing methods. Fluorescence observations exhibit structured dynamics due to the exponential intracellular calcium concentration decay (as shown in *Figure 4C*, for example), which are in turn related to the underlying spikes that are driven non-linearly by intrinsic/extrinsic stimuli as well as the properties of the indicator used. As such, an accurate inference method is expected to be specific to this temporal structure. To test this, we randomly shuffled the $T$ time frames consistently in the same order in all trials, in order to fully break the temporal structure governing calcium decay dynamics, and then estimated correlations from these shuffled data using the different methods. The resulting estimates of noise correlations are shown in *Figure 5A* for one instance of such shuffled data. The average NMSE for a total of 50 shuffled samples with respect to the original un-shuffled estimates (in *Figure 4A*) are tabulated in the fifth and sixth columns of *Table 1*, and are also indicated below each panel in *Figure 5A*.

A visual inspection of *Figure 5A* shows that the Pearson correlations from two-photon recordings expectedly remain unchanged. Since this method treats each time frame to be independent, temporal shuffling does not impact the correlations in anyway. On the other extreme, both of the two-stage estimates seem to detect highly variable and large correlation values, despite operating on data that lacks any relevant temporal structure. Our proposed method, however, remarkably produces negligible correlation estimates. Although both the two-stage and proposed estimates show variability with respect to the shuffled data (*Table 1*, fifth column), the standard deviation of the NMSE values of our proposed method are considerably smaller than those of the two-stage methods (*Table 1*, fifth column). For further inspection, the histograms of a single element $((\widehat{\mathbf{N}})_{1,3})$ of the estimated correlation matrices across the 50 shuffling trials are shown in *Figure 5B*. The original un-shuffled estimates are marked by the dashed vertical lines in each case. The proposed estimate in *Figure 5B* is highly concentrated around zero, even though the un-shuffled estimate is non-zero. However, the two-stage estimates produce correlations that are widely variable across the shuffling trials. This analysis demonstrates that our proposed method is highly specific to the temporal structure of fluorescence observations, whereas the Pearson correlations from two-photon recordings, two-stage Pearson and two-stage GPFA methods fail to be specific.

## Real data study 2: spontaneous vs. stimulus-driven activity in the mouse A1

To further validate the utility of our proposed methodology, we applied it to another experimentally-recorded dataset from the mouse A1 layer 2/3. This experiment pertained to trials of presenting a sequence of short white noise stimuli, randomly interleaved with silent trials of the same duration. *Figure 6A* shows a sample trial sequence. The two-photon recordings thus contained episodes of stimulus-driven and spontaneous activity (see Experimental procedures for details). Under this experimental setup, it is expected that the noise correlations are invariant across the spontaneous and stimulus-driven conditions. In accordance with the foregoing results of real data study 1, we also expect the signal and noise correlation patterns to be distinct. Each trial considered in the analysis consisted of $T = 765$ frames (see Experimental procedures for details). We selected $N = 10$ responsive neurons (according to the criterion described in Guidelines for model parameter settings), each with $L = 10$ trials. Similar to real data study 1, we chose a stimulus integration window of length $R = 25$ frames.

*Figure 6B* shows the resulting noise and signal correlation estimates under the spontaneous ($\widehat{\mathbf{N}}_{spon}$, top) and stimulus-driven ($\widehat{\mathbf{N}}_{stim}$ and $\widehat{\mathbf{S}}_{stim}$, bottom) conditions. *Figure 6C* shows the spatial map of the 10 neurons considered in the analysis in the field of view. A visual inspection of the first column of *Figure 6B* indeed suggests that $\widehat{\mathbf{N}}_{spon}$ and $\widehat{\mathbf{N}}_{stim}$ are saliently similar, and distinct from $\widehat{\mathbf{S}}_{stim}$.

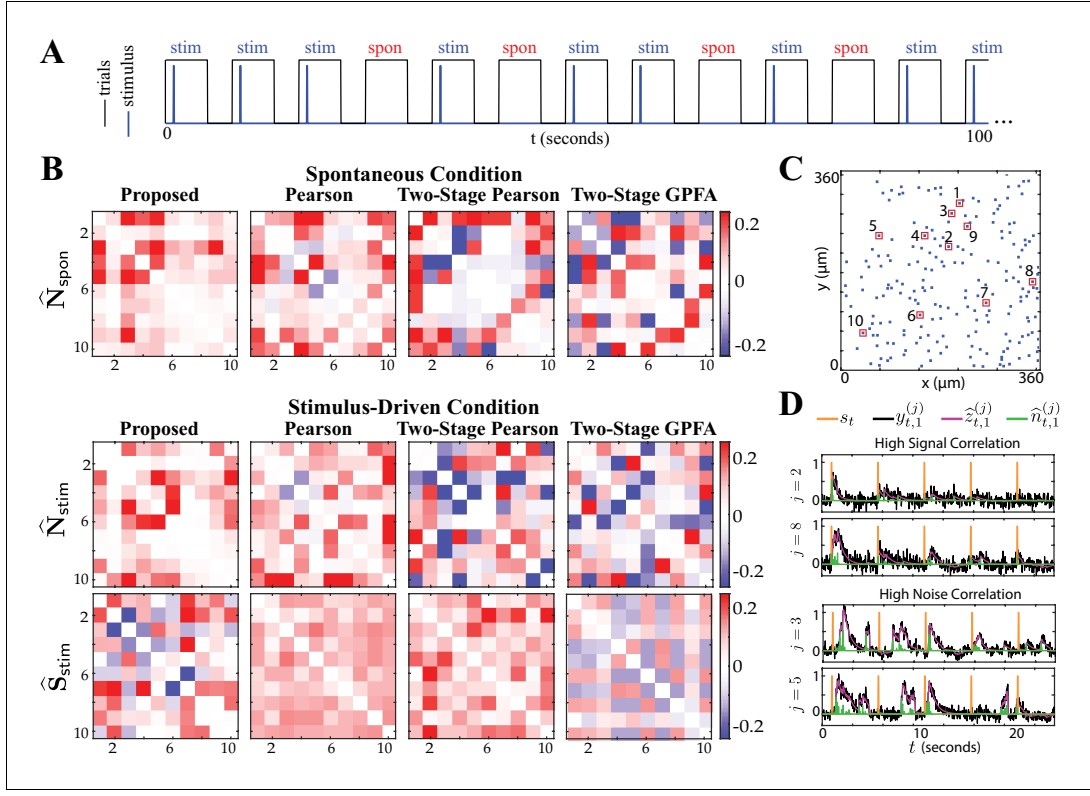

**Figure 6.** Comparison of spontaneous and stimulus-driven activity in the mouse A1. (**A**) A sample trial sequence in the experiment. Stimulus-driven (stim) trials were recorded with randomly interleaved spontaneous (spon) trials of the same duration. (**B**) Estimated noise and signal correlation matrices under spontaneous (top) and stimulus-driven (bottom) conditions. Rows from left to right: proposed method, Pearson correlations from two-photon data, two-stage Pearson and two-stage GPFA estimates. (**C**) Location of the selected neurons with highest activity in the field of view. (**D**) Stimulus onsets (orange), observations (black), estimated calcium concentrations (purple) and putative spikes (green) for the first trial from two pairs of neurons with high signal correlation (top) and high noise correlation (bottom), as identified by the proposed estimates.

The online version of this article includes the following figure supplement(s) for figure 6:

**Figure supplement 1.** Histograms of the similarity/dissimilarity metrics under the shuffling procedure.

The Pearson correlations obtained from two-photon data (second column) and the two-stage Pearson and GPFA estimates (third and fourth columns, respectively), however, evidently lack this structure. As in the previous study, we quantified this visual comparison using the similarity metric $T_s(\mathbf{X}, \mathbf{Y})$ and the dissimilarity metric $T_d(\mathbf{X}, \mathbf{Y})$ (see Performance evaluation for details). These statistics are reported in *Table 2* along with the p-values (null distributions are shown in *Figure 6—figure supplement 1*), which show that the only significant outcomes (boldface numbers) are those of our proposed method. While it is expected from the experiment design for the noise correlations under the two settings to be similar, the only method that detects this expected outcome with statistical significance is our proposed method. Moreover, the statistically significant dissimilarity between the

**Table 2.** Similarity/dissimilarity metric statistics for the estimates in *Figure 6*.

| Estimation method | $T_s(\widehat{\mathbf{N}}_{spon}, \widehat{\mathbf{N}}_{stim})$ | $T_d(\widehat{\mathbf{S}}_{stim}, \widehat{\mathbf{N}}_{stim})$ |
|---|---|---|
| **Proposed** | **0.5716 ($p = 0.003$)** | **0.7946 ($p = 0.004$)** |
| Pearson | 0.3031 ($p = 0.61$) | 0.5032 ($p = 0.92$) |
| Two-stage Pearson | 0.2790 ($p = 0.05$) | 0.7862 ($p = 0.39$) |
| Two-stage GPFA | 0.2008 ($p = 0.50$) | 0.7792 ($p = 0.22$) |

signal and noise correlations of our proposed estimates corroborate the hypothesis that signal and noise are encoded by distinct functional networks (*Kohn et al., 2016*; *Montijn et al., 2014*; *Rothschild et al., 2010*; *Keeley et al., 2020*).

Furthermore, *Figure 6D* shows the time course of the stimulus, observations, estimated calcium concentrations and putative spikes for the first trial from two pairs of neurons with high signal correlation ($j = 2, 8$, top) and high noise correlation ($j = 3, 5$, bottom). As expected, the putative spiking activity of the neurons with high signal correlation (top) are closely time-locked to the stimulus onsets. The activity of the two neurons with high noise correlation (bottom), however, is not time-locked to the stimulus onsets, even though the two neurons exhibit highly correlated activity. The correlations estimated via the proposed method thus encode substantial information about the inter-dependencies of the spiking activity of the neuronal ensemble.

## Real data study 3: spatial analysis of signal and noise correlations in the mouse A1

Lastly, we applied our proposed method to examine the spatial distribution of signal and noise correlations in the mouse A1 layers 2/3 and 4 (data from *Bowen et al., 2020*). The dataset included fluorescence activity recorded during multiple experiments of presenting sinusoidal amplitude-modulated tones, with each stimulus being repeated across several trials (see Experimental procedures and *Bowen et al., 2020* for experimental details). In each experiment, we selected on average around 20 responsive neurons for subsequent analysis (according to the criterion described in Guidelines for model parameter settings). For brevity, we compare the estimates of signal and noise correlations using our proposed method only with those obtained by Pearson correlations from the two-photon data. The latter method was also used in previous analyses of data from this experimental paradigm (*Winkowski and Kanold, 2013*).

In parallel to the results reported in *Winkowski and Kanold, 2013*, *Figure 7A* and *Figure 7B* illustrate the correlation between the signal and noise correlations in layers 2/3 and 4, respectively. Consistent with the results of *Winkowski and Kanold, 2013*, the signal and noise correlations exhibit positive correlation in both layers, regardless of the method used. However, the correlation coefficients (i.e. slopes in the insets) identified by our proposed method are notably smaller than those obtained from Pearson correlations, in both layer 2/3 (*Figure 7A*) and layer 4 (*Figure 7B*). Comparing this result with our simulation studies suggests that the stronger linear trend between the signal and noise correlations observed using the Pearson correlation estimates is likely due to the mixing between the estimates of signal and noise correlations. As such, our method suggests that the signal and noise correlations may not be as highly correlated with one another as indicated in previous studies of layer 2/3 and 4 in mouse A1 (*Winkowski and Kanold, 2013*).

Next, to evaluate the spatial distribution of signal and noise correlations, we plotted the correlation values for pairs of neurons as a function of their distance for layer 2/3 (*Figure 7C*) and layer 4 (*Figure 7D*). The distances were discretized using bins of length $10\,\mu m$. The scatter of the correlations along with their median at each bin are shown in all panels. Then, to examine the spatial trend of the correlations, we performed linear regression in each case. The slope of the model fit, the p-value for the corresponding t-test, and the $R^2$ values are reported in *Table 3* (the slope and p-values are also shown as insets in *Figure 7C and D*).

From *Table 3* and *Figure 7C and D* (upper panels), it is evident that the signal correlations show a significant negative trend with respect to distance, using both methods and in both layers. However, the slope of these negative trends identified by our method (boldface numbers in *Table 3*) is notably steeper than those identified by Pearson correlations. On the other hand, the trends of the noise correlations with distance (bottom panels) are different between our proposed method and Pearson correlations: our proposed method shows a significant negative trend in layer 2/3, but not in layer 4, whereas the Pearson correlations of the two-photon data suggest a significant negative trend in layer 4, but not in layer 2/3. In addition, the slopes of these negative trends identified by our method (boldface numbers in *Table 3*) are steeper than or equal to those identified by Pearson correlations.

Our proposed estimates also indicate that noise correlations are sparser and less widespread in layer 4 (*Figure 7D*) than in layer 2/3 (*Figure 7C*). To further investigate this observation, we depicted the two-dimensional spatial spread of signal and noise correlations in both layers and for both methods in *Figure 7E and F*, by centering each neuron at the origin and overlaying the individual spatial

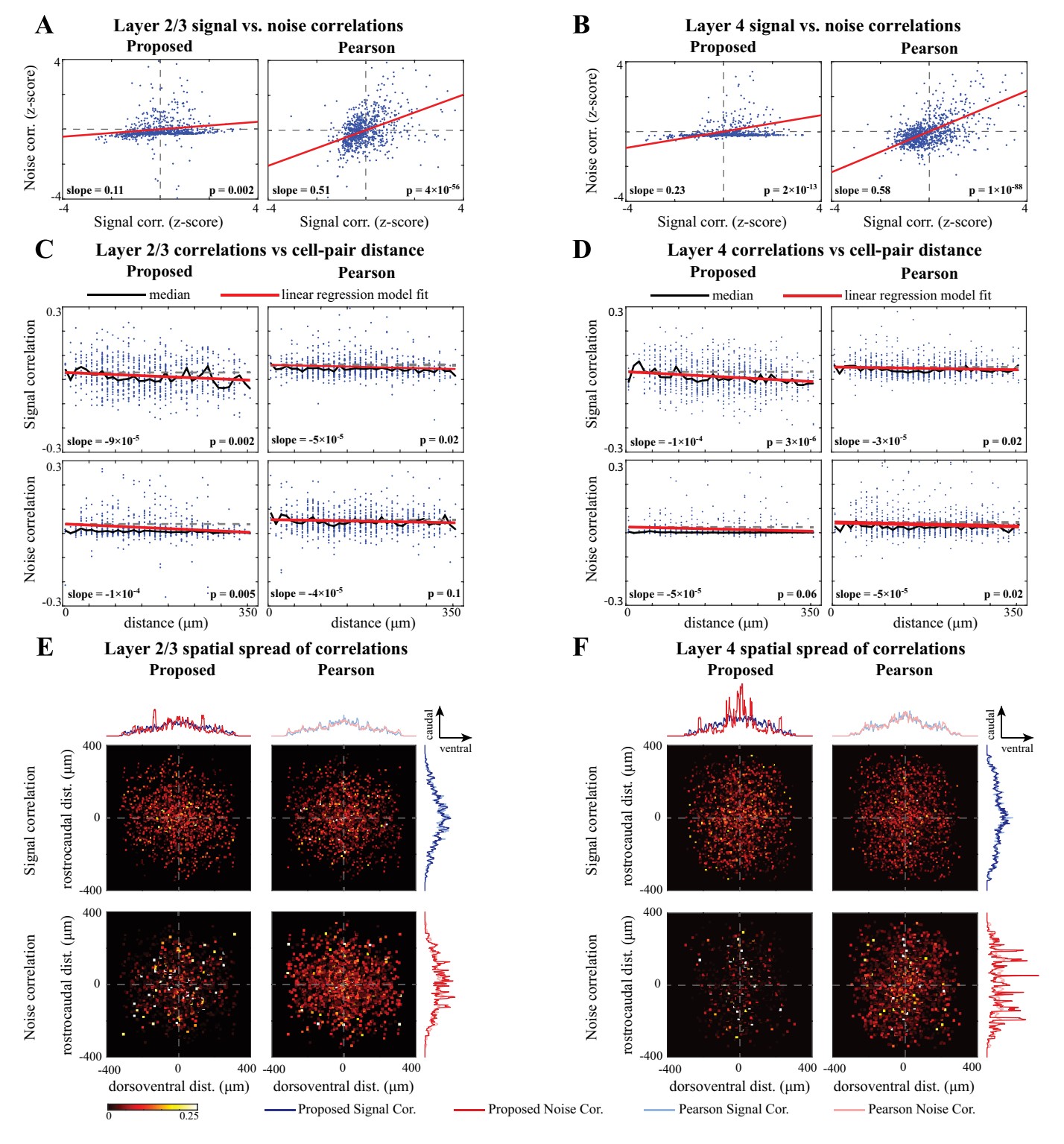

**Figure 7.** Comparison of signal and noise correlations across layers 2/3 and 4. (**A**) Scatter-plot of noise vs. signal correlations (blue) for individual cell-pairs in layer 2/3, based on the proposed (left) and Pearson estimates (right). Data were normalized for comparison by computing z-scores. The linear model fits are shown in red, and the slope and p-value of the t-tests are indicated as insets. Panel (**B**) corresponds to layer 4 in the same organization as panel A. (**C**) Signal (top) and noise (bottom) correlations vs. cell-pair distance in layer 2/3, based on the proposed (left) and Pearson estimates (right). Distances were binned to $10\,\mu m$ intervals. The median of the distributions (black) and the linear model fit (red) are shown in each panel. The slope of the linear model fit, and the p-value of the t-test are also indicated as insets. Dashed horizontal lines indicate the zero-slope line for ease of visual

*Figure 7 continued on next page*

Figure 7 continued

comparison. Panel **D** corresponds to layer 4 in the same organization as panel C. (**E**) Spatial spread of signal (top) and noise (bottom) correlations in layer 2/3, based on the proposed (left) and Pearson estimates (right). The horizontal and vertical axes in each panel respectively represent the relative dorsoventral and rostrocaudal distances between each cell-pair, and the heat-map indicates the magnitude of correlations. Marginal distributions of the signal (blue) and noise (red) correlations along the dorsoventral and rostrocaudal axes for the proposed method (darker colors) and Pearson method (lighter colors) are shown at the top and right sides of the sub-panels. Panel **F** corresponds to layer 4 in the same organization as panel E.

The online version of this article includes the following figure supplement(s) for figure 7:

**Figure supplement 1.** Comparing the marginal distributions of signal and noise correlations along the dorsoventral and rostrocaudal axes.

**Figure supplement 2.** Marginal angular distributions of signal and noise correlations.

spreads. The horizontal and vertical axes in each panel represent the relative dorsoventral and rostrocaudal distances, respectively, and the heat-maps represent the magnitude of correlations. Comparing the proposed noise correlation spread in *Figure 7E* with the corresponding spread in *Figure 7F*, we observe that the noise correlations in layer 2/3 are indeed more widespread and abundant than in layer 4, as can be expected by more extensive intralaminar connections in layer 2/3 vs. 4 (*Watkins et al., 2014*; *Meng et al., 2017a*; *Meng et al., 2017b*; *Kratz and Manis, 2015*).

The spatial spreads of signal and noise correlations based on the Pearson estimates are remarkably similar in both layers (*Figure 7E and F*, right panels), whereas they are saliently different for our proposed estimates (*Figure 7E and F*, left panels). This further corroborates our hypothesis on the possibility of high mixing between the signal and noise correlation estimates obtained by the Pearson correlation of two-photon data. To further examine the differences between the signal and noise correlations, the marginal distributions along the dorsoventral and rostrocaudal axes are shown in *Figure 7E and F*, selectively overlaid for ease of visual comparison. To quantify the differences between the spatial distributions of signal and noise correlations estimated by each method, we performed Kolmogorov-Smirnov (KS) tests on each pair of marginal distributions, which are summarized in *Figure 7—figure supplement 1*. Although the marginal distributions of signal and noise correlations are significantly different in all cases from both methods, the effect sizes of their difference (KS statistics) are higher for our proposed estimates compared to those of the Pearson estimates.

Finally, the spatial spreads of correlations for either method and in each layer suggest non-uniform angular distributions with possibly directional bias. To test this effect, we computed the angular marginal distributions and performed KS tests for non-uniformity, which are reported in *Figure 7—figure supplement 2*. These tests indicate that all distributions are significantly non-uniform. In addition, the angular distributions of both signal and noise correlations in layer 4 exhibit salient modes in the rostrocaudal direction, whereas they are less directionally selective in layer 2/3 (*Figure 7—figure supplement 2*).

In summary, the spatial trends identified by our proposed method are consistent with empirical observations of spatially heterogeneous pure-tone frequency tuning by individual neurons in auditory cortex (*Winkowski and Kanold, 2013*). The improved correspondence of our proposed method compared to results obtained using Pearson correlations could be the result of the demixing of signal and noise correlations in our method. As a result of the demixing, our proposed method also suggests that noise correlations have a negative trend with distance in layer 2/3, but are much sparser and spatially flat in layer 4. In addition, the spatial spread patterns of signal and noise correlations are more structured and remarkably more distinct for our proposed method than those obtained by the Pearson estimates.

**Table 3.** Linear regression statistics for the analysis of correlations vs. cell-pair distance.

| | Statistics of layer 2/3 correlations | | Statistics of layer 4 correlations | |
| --- | --- | --- | --- | --- |
| Correlations | Slope (p-value) | $R^2$ Value | Slope (p-value) | $R^2$ Value |
| Proposed Signal Corr. | $-9 \times 10^{-5}$ ($p = 0.002$) | 0.012 | $-1 \times 10^{-4}$ ($p = 3 \times 10^{-6}$) | 0.023 |
| Pearson Signal Corr. | $-5 \times 10^{-5}$ ($p = 0.02$) | 0.007 | $-3 \times 10^{-5}$ ($p = 0.02$) | 0.005 |
| Proposed Noise Corr. | $-1 \times 10^{-4}$ ($p = 0.005$) | 0.010 | $-5 \times 10^{-5}$ ($p = 0.06$) | 0.004 |
| Pearson Noise Corr. | $-4 \times 10^{-5}$ ($p = 0.1$) | 0.003 | $-5 \times 10^{-5}$ ($p = 0.02$) | 0.005 |

## Theoretical analysis of the bias and variance of the proposed estimators

Finally, we present a theoretical analysis of the bias and variance of the proposed estimator. Note that our proposed estimation method has been developed as a scalable alternative to the intractable maximum likelihood (ML) estimation of the signal and noise covariances (see Overview of the proposed estimation method). In order to benchmark our estimates, we thus need to evaluate the quality of said ML estimates. To this end, we derived bounds on the bias and variance of the ML estimators of the kernel $\mathbf{d}_j$ for $j = 1, \cdots, N$ and the noise covariance $\boldsymbol{\Sigma}_x$. In order to simplify the treatment, we posit the following mild assumptions:

Assumption (1). We assume a scalar time-varying external stimulus (i.e. $\mathbf{s}_t = s_t$, and hence $\mathbf{d}_j = d_j, \mathbf{d} = [d_1, d_2, \cdots, d_N]^\top$). Furthermore, we set the observation noise covariance to be $\boldsymbol{\Sigma}_w = \sigma_w^2 \mathbf{I}$, for notational convenience.

Assumption (2). We derive the performance bounds in the regime where $T$ and $L$ are large, and thus do not impose any prior distribution on the correlations, which are otherwise needed to mitigate overfitting (see Preliminary assumptions).

Assumption (3). We assume the latent trial-dependent process and stimulus to be slowly varying signals, and thus adopt a piece-wise constant model in which these processes are constant within consecutive windows of length $W$ (i.e. $\mathbf{x}_{t,l} = \mathbf{x}_{W_k,l}$ and $s_t = s_{W_k}$, for $(k-1)W + 1 \leq t < kW$ and $k = 1, \cdots, K$ with $W_k = (k-1)W + 1$ and $KW = T$) for our theoretical analysis, as is usually done in spike count calculations for conventional noise correlation estimates.

Our main theoretical result is as follows:

Theorem 1 (Performance Bounds) Let $q > \frac{1}{64}$, $0 < \epsilon < 1/2$, and $0 < \eta \leq 1/2$ be fixed constants, $\sigma_m^2 := \max_i (\boldsymbol{\Sigma}_x)_{i,i}$ and $\sigma_s^2 := \frac{1}{K} \sum_{k=1}^{K} s_{W_k}^2$. Then, under Assumptions (1 - 3), the bias and variance of the maximum likelihood estimators $\widehat{\mathbf{d}}$ and $\widehat{\boldsymbol{\Sigma}}_x$, conditioned on an event $\mathscr{A}_W$ with $\mathbb{P}(\mathscr{A}_W) \geq 1 - \eta$ satisfy:

$$\left| bias_{\mathscr{A}_W}\left(\widehat{d}_j\right) \right| \leq \frac{1}{\sqrt{W^{1-2\epsilon}}} C_1 \left(2\sigma_w \sqrt{1 + \alpha^2} + 1\right) + \tau_j,$$

$$\sqrt{Var_{\mathscr{A}_W}\left(\widehat{d}_j\right)} \leq \sqrt{\frac{(\boldsymbol{\Sigma}_x)_{j,j}}{KL\sigma_s^2(1-\eta)} + \frac{1}{\sqrt{W^{1-2\epsilon}}} C_2 \left(2\sigma_w \sqrt{1 + \alpha^2} + 1\right) + \widetilde{\tau}_j},$$

$$\left| bias_{\mathscr{A}_W}\left((\widehat{\boldsymbol{\Sigma}}_x)_{i,j}\right) \right| \leq \frac{|(\boldsymbol{\Sigma}_x)_{i,j}|}{KL(1-\eta)} + \sqrt{\frac{\log W}{W^{1-2\epsilon}}} C_3 \left(14\sigma_w \sqrt{1 + \alpha^2} + 3\right) + \xi_{i,j},$$

$$\sqrt{Var_{\mathscr{A}_W}\left((\widehat{\boldsymbol{\Sigma}}_x)_{i,j}\right)} \leq \sqrt{\frac{(KL-1)\left((\boldsymbol{\Sigma}_x)_{i,j}^2 + (\boldsymbol{\Sigma}_x)_{i,i}(\boldsymbol{\Sigma}_x)_{j,j}\right)}{K^2 L^2(1-\eta)} + \sqrt{\frac{\log W}{W^{1-2\epsilon}}} C_4 \left(2\sigma_w \sqrt{1 + \alpha^2} + 1\right) + \widetilde{\xi}_{i,j}},$$

for all $i, j = 1, 2, \cdots, N$, if

$$\log W \geq \max \left\{ \frac{\log(8KLN/\eta)}{q}, \frac{32\sigma_m^2 q}{\epsilon^2}, \frac{2\log(64q)}{1-2\epsilon}, \frac{\max\{6.25, 4(\|\boldsymbol{\mu}_x\|_\infty + \max_{k,j}\{|s_{W_k}d_j|\})^2\}}{8q\sigma_m^2}, \log 2 \right\},$$

where $\tau_j$ and $\widetilde{\tau}_j$ denote bounded terms that are $\mathcal{O}(\sigma_w^2)$ or $\mathcal{O}\left(\frac{1}{W}\right)$, $\xi_{i,j}$ and $\widetilde{\xi}_{i,j}$ denote bounded terms that are $\mathcal{O}(\sigma_w^2)$ or $\mathcal{O}\left(\frac{1}{W^{1-2\epsilon}}\right)$ and $C_1, C_2, C_3$ and $C_4$ are bounded constants given in Appendix 2.

Proof. The proof of Theorem 1 is provided in Appendix 2.

∎

In order to discuss the implications of this theoretical result, several remarks are in order:

Remark 1: Achieving near Oracle performance
A common benchmark in estimation theory is the performance of the idealistic *oracle* estimator, in which an oracle directly observes the true latent process $\mathbf{x}_{t,l}$ and the true kernel $d_j$ and forms the correlation estimates. In this case, the oracle would incur zero bias and variance of order $\mathcal{O}(1/KL)$ in estimating $d_j$, and outputs an estimate of $\boldsymbol{\Sigma}_x$ with bias and variance in the order of $\mathcal{O}(1/KL)$. Theorem 1 indeed states that for sufficiently large $W$ and small $\sigma_w$, the bias and variance of the ML estimators are arbitrarily close to those of the oracle estimator. Recall that our variational inference framework is in fact a solution technique for the regularized ML problem. Hence, the bounds in Theorem 1 provide a benchmark for the expected

performance of the proposed estimators, by quantifying the excess bias and variance over the performance of the oracle estimator.

Remark 2: Effect of the observation noise and observation duration

As the assumed window of stationarity $W \to \infty$ (and hence the observation duration $T \to \infty$), the loss of performance of the proposed estimators only depends on $\sigma_w^2$, the variance of the observation noise. As a result, at a given observation noise variance $\sigma_w^2$, these bounds provide a sufficient upper bound on the time duration of the observations required for attaining a desired level of estimation accuracy. It is noteworthy that $\sigma_w^2$ is typically small in practice, as it pertains to the effective observation noise and is significantly diminished by pixel averaging of the fluorescence traces following cell segmentation.

Remark 3: Effect of the number of trials

Finally, note that the bounds in Theorem 1 have terms that also drop as the number of trials $L$ grows. These terms in fact pertain to the performance of the oracle estimator. As the number of trials grows ($L \to \infty$), the oracle estimates become arbitrarily close to the true parameters $\boldsymbol{\Sigma}_x$ and $\mathbf{d}_j$. Thus, our theoretical performance bounds also provide a sufficient upper bound on the number of trials $L$ required for the oracle estimator to attain a desired level of estimation accuracy.

## Discussion

We developed a novel approach for the joint estimation of signal and noise correlations of neuronal activities directly from two-photon calcium imaging observations and tested our method with experimental data. Existing widely used methods either take the fluorescence traces as surrogates of spiking activity, or first recover the unobserved spikes using deconvolution techniques, both followed by computing Pearson correlations or connectivity matrices. As such, they typically result in estimates that are highly biased and are heavily dependent on the choice of the spike deconvolution technique. We addressed these issues by explicitly relating the signal and noise covariances to the observed two-photon data via a multi-tier Bayesian model that accounts for the observation process and non-linearities involved in spiking activity. We developed an efficient estimation framework by integrating techniques from variational inference and state-space estimation. We also established performance bounds on the bias and variance of the proposed estimators, which revealed favorable scaling with respect to the observation noise and trial length.

We demonstrated the utility of our proposed estimation framework on both simulated and experimentally recorded data from the mouse auditory cortex. In our simulation studies, we evaluated the robustness of our proposed method with respect to several model mismatch conditions induced by the stimulus integration model, calcium decay, SNR level, firing rate, and temporally correlated observation noise. In all cases, we observed that our proposed estimates outperform the existing methods in recovering the signal and noise correlations.

There are two main sources for the observed performance gap between our proposed method and existing approaches. The first source is the favorable *soft* decisions on the timing of spikes achieved by our method as a byproduct of the iterative variational inference procedure. An accurate probabilistic decoding of spikes results in better estimates of the signal and noise correlations, and conversely having more accurate estimates of the signal and noise covariances improves the probabilistic characterization of spiking events. This is in contrast with both the Pearson correlations computed from two-photon data and two-stage methods: in computing the Pearson correlations from two-photon data, spike timing is heavily blurred by the calcium decay; in the two-stage methods, erroneous *hard* decisions on the timing of spikes result in biases that propagate to and contaminate the downstream signal and noise correlation estimation and thus results in significant errors.

The second source of performance improvement is the explicit modeling of the non-linear mapping from stimulus and latent covariates to spiking through a canonical point process model, which is in turn tied to a two-photon observation model in a multi-tier Bayesian fashion. Our theoretical analysis in Theorem 1 corroborates that this virtue of our proposed methodology results in robust performance under limited number of trials. As we have shown in Appendix 1, as the number of trials $L$ and trial duration $T$ tend to infinity, conventional notions of signal and noise correlation indeed recover the ground truth signal and noise correlations, as the biases induced by non-linearities average out across trial repetitions. However, as exemplified in *Figure 2—figure supplement 2*, in order

to achieve comparable performance to our method using few trials (e.g. $L = 20$), the conventional correlation estimates require considerably more trials (e.g. $L = 1000$).

Application to two-photon data recorded from the mouse primary auditory cortex showed that unlike the aforementioned existing methods, our estimates provide noise correlation structures that are expectedly invariant across spontaneous and stimulus-driven conditions within an experiment, while producing signal correlation structures that are largely distinct from those given by noise correlation. These results provide evidence for the involvement of distinct functional neuronal network structures in encoding the stimulus-dependent and stimulus-independent information.

Our analysis of the relationship between the signal and noise correlations in layers 2/3 and 4 in mouse A1 indicated a smaller correlation between signal and noise correlations than previously reported (*Winkowski and Kanold, 2013*). Thus, our proposed method suggests that the signal and noise correlations reflect distinct circuit mechanisms of sound processing in layers 2/3 vs 4. The spatial distribution of signal correlations obtained by our method was consistent with previous work showing significant negative trends with distance (*Winkowski and Kanold, 2013*). However, in addition, our proposed method revealed a significant negative trend of noise correlations with distance in layer 2/3, but not in layer 4, in contrast to the outcome of Pearson correlation analysis. The lack of a negative trend in layer 4 could be attributed to the sparse nature of the noise correlation spread in layer 4, as revealed by our analysis of two-dimensional spatial spreads. The latter analysis indeed revealed that the noise correlations in layer 2/3 are more widespread than those in layer 4, consistent with existing work based on whole-cell patch recordings (*Meng et al., 2017a*; *Meng et al., 2017b*).

The two-dimensional spatial spreads of signal and noise correlations obtained by our method are more distinct than those obtained by Pearson correlations. The spatial spreads also allude to directionality of the functional connectivity patterns, with a notable rostrocaudal preference in layer 4. This result seems surprising in light of existing evidence for quasi-rostrocaudal organization of the tonotopic axis in mouse A1 (*Romero et al., 2020*). However, given the heterogeneity of tuning in both layers 2/3 and 4 with a best frequency interqartile range of ~1–1.5 octaves over the imaging field (*Bowen et al., 2020*) and using supra-threshold tones, we expect that the tones will drive not only neurons with the corresponding best frequency, but also neurons tuned to neighboring frequencies. Moreover, there is high connectivity between layer 4 cells within a few 100 µm across the tonotopic axis (*Kratz and Manis, 2015*; *Meng et al., 2017a*), potentially amplifying and broadening the effect of supra-threshold tones.

Our proposed method can scale up favorably to larger populations of neurons, thanks to the underlying low-complexity variational updates in the inference procedure. Due to its minimal dependence on training data, our estimation framework is also applicable to single-session analysis of two-photon data with limited number of trials and duration. Another useful byproduct of the proposed framework is gaining access to approximate posterior densities in closed-form, which allows further statistical analyses such as construction of confidence intervals. Our proposed methodology can thus be used as a robust and scalable alternative to existing approaches for extracting neuronal correlations from two-photon calcium imaging data.

A potential limitation of our proposed generative model is the assumption that there is at most one spiking event per time frame for each neuron, in light of the fact that typical two-photon imaging frame durations are in the range of 30–100 ms. Average spike rates of excitatory neurons in mouse A1 layers 2/3 and 4 are of the order of < 10 Hz (*Petrus et al., 2014*; *Forli et al., 2018*) and thus our model is reasonable for the current study, although it might not be optimal during bursting activity. It is noteworthy that we relax this assumption in the inference framework by allowing the magnitude of putative spikes to be greater than one, thus alleviating the model mismatch during episodes of rapid increase in firing rate. This assumption can also be made more precise by adopting a Poisson model, but that would render closed-form variational density updates intractable.

Furthermore, in the regime of extremely low spiking rate and high observation noise, the proposed method may fail to capture the underlying correlations faithfully and its performance degrades to those of existing methods based on Pearson correlations, as we have shown through our simulation studies. Nevertheless, our method addresses key limitations of conventional signal and noise correlation estimators that persist even in high spiking rate and high SNR conditions.

Our proposed estimation framework can be used as groundwork for incorporating other notions of correlation such as the connected correlation function (*Martin et al., 2020*), and to account for non-Gaussian and higher order structures arising from spatiotemporal interactions (*Kadirvelu et al., 2017*; *Yu et al., 2011*). Other possible extensions of this work include leveraging variational inference beyond the mean-field regime (*Wang and Blei, 2013*), extension to time-varying correlations that underlie rapid task-dependent dynamics, and extension to non-linear models such as those parameterized by neural networks (*Aitchison et al., 2017*). In the spirit of easing reproducibility, a MATLAB implementation of our proposed method as well as the data used in this work are made publicly available (*Rupasinghe, 2020*; *Rupasinghe et al., 2021*).

# Materials and methods

## Proposed forward model

Suppose we observe fluorescence traces of $N$ neurons, for a total duration of $T$ discrete-time frames, corresponding to $L$ independent trials of repeated stimulus. Let $\mathbf{y}_{t,l} := [y_{t,l}^{(1)}, y_{t,l}^{(2)}, \cdots, y_{t,l}^{(N)}]^\top$, $\mathbf{z}_{t,l} := [z_{t,l}^{(1)}, z_{t,l}^{(2)}, \cdots, z_{t,l}^{(N)}]^\top$, and $\mathbf{n}_{t,l} := [n_{t,l}^{(1)}, n_{t,l}^{(2)}, \cdots, n_{t,l}^{(N)}]^\top$ be the vectors of noisy observations, intracellular calcium concentrations, and ensemble spiking activities, respectively, at trial $l$ and frame $t$. We capture the dynamics of $\mathbf{y}_{t,l}$ by the following state-space model:

$$\mathbf{y}_{t,l} = \mathbf{A}\,\mathbf{z}_{t,l} + \mathbf{w}_{t,l}, \qquad \mathbf{z}_{t,l} = \alpha\,\mathbf{z}_{t-1,l} + \mathbf{n}_{t,l},$$

where $\mathbf{A} \in \mathbb{R}^{N \times N}$ represents the scaling of the observations, $\mathbf{w}_{t,l}$ is zero-mean i.i.d. Gaussian noise with covariance $\mathbf{\Sigma}_w$, and $0 \leq \alpha < 1$ is the state transition parameter capturing the calcium dynamics through a first order model. Note that this state-space is non-Gaussian due to the binary nature of the spiking activity, that is, $n_{t,l}^{(j)} \in \{0,1\}$. We model the spiking data as a point process or Generalized Linear Model with Bernoulli statistics (*Eden et al., 2004*; *Paninski, 2004*; *Smith and Brown, 2003*; *Truccolo et al., 2005*):

$$n_{t,l}^{(j)} \sim Bernoulli\left(\lambda_{t,l}^{(j)}\right), \qquad \lambda_{t,l}^{(j)} = \phi\left(x_{t,l}^{(j)}, \mathbf{d}_j^\top \mathbf{s}_t\right),$$

where $\lambda_{t,l}^{(j)}$ is the conditional intensity function (*Truccolo et al., 2005*), which we model as a non-linear function of the known external stimulus $\mathbf{s}_t$ and the other latent intrinsic and extrinsic trial-dependent covariates, $\mathbf{x}_{t,l} := [x_{t,l}^{(1)}, x_{t,l}^{(2)}, \cdots, x_{t,l}^{(N)}]^\top$. While we assume the stimulus $\mathbf{s}_t \in \mathbb{R}^M$ to be common to all neurons, we model the distinct effect of this stimulus on the $j^{th}$ neuron via an unknown kernel $\mathbf{d}_j \in \mathbb{R}^M$, akin to the receptive field.

The non-linear mapping of our choice is the logistic link, which is also the canonical link for a Bernoulli process in the point process and Generalized Linear Model frameworks (*Truccolo et al., 2005*). Thus, we assume:

$$\phi\left(x_{t,l}^{(j)}, \mathbf{d}_j^\top \mathbf{s}_t\right) = \frac{\exp\left(x_{t,l}^{(j)} + \mathbf{d}_j^\top \mathbf{s}_t\right)}{1 + \exp\left(x_{t,l}^{(j)} + \mathbf{d}_j^\top \mathbf{s}_t\right)}.$$

Finally, we assume the latent trial dependent covariates to be a Gaussian process $\mathbf{x}_{t,l} \sim \mathcal{N}(\mathbf{\mu}_x, \mathbf{\Sigma}_x)$, with mean $\mathbf{\mu}_x := [\mu_x^{(1)}, \mu_x^{(2)}, \cdots, \mu_x^{(N)}]^\top$ and covariance $\mathbf{\Sigma}_x$.

The probabilistic graphical model in *Figure 8* summarizes the main components of the aforementioned forward model. According to this forward model, the underlying noise covariance matrix that captures trial-to-trial variability can be identified as $\mathbf{\Sigma}_x$. The signal covariance matrix, representing the covariance of the neural activity arising from the repeated application of the stimulus $\mathbf{s}_t$, is given by $\mathbf{\Sigma}_s := \mathbf{D}^\top cov(\mathbf{s}_t, \mathbf{s}_t)\,\mathbf{D}$, where $\mathbf{D} := [\mathbf{d}_1, \mathbf{d}_2, \cdots, \mathbf{d}_N] \in \mathbb{R}^{M \times N}$. The signal and noise correlation matrices, denoted by $\mathbf{S}$ and $\mathbf{N}$, can then be obtained by standard normalization of $\mathbf{\Sigma}_s$ and $\mathbf{\Sigma}_x$:

$$(\mathbf{S})_{i,j} := \frac{(\mathbf{\Sigma}_s)_{i,j}}{\sqrt{(\mathbf{\Sigma}_s)_{i,i} \cdot (\mathbf{\Sigma}_s)_{j,j}}}, \quad (\mathbf{N})_{i,j} := \frac{(\mathbf{\Sigma}_x)_{i,j}}{\sqrt{(\mathbf{\Sigma}_x)_{i,i} \cdot (\mathbf{\Sigma}_x)_{j,j}}}, \quad \forall i,j = 1, 2, \cdots, N.$$

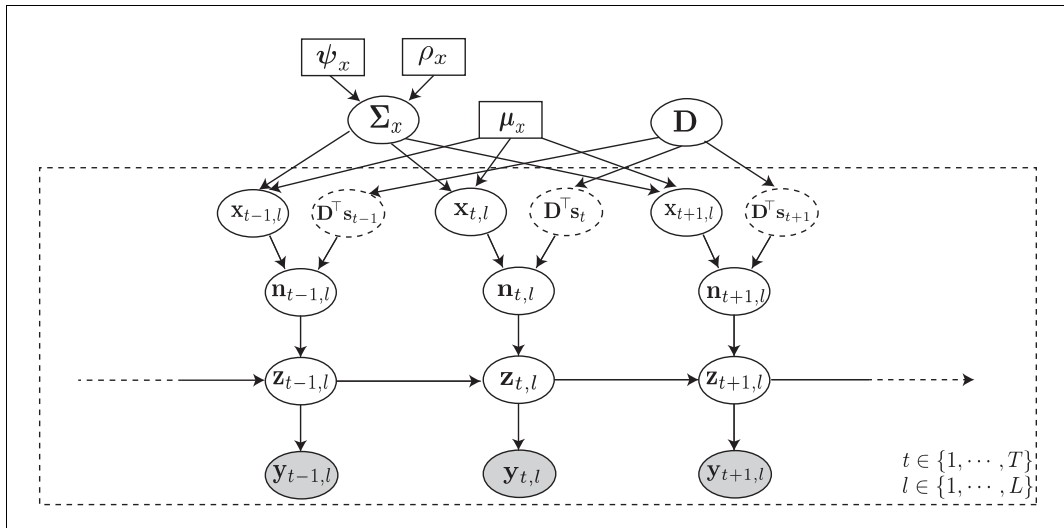

**Figure 8.** Probabilistic graphical model of the proposed forward model. The fluorescence observations at the $t^{th}$ time frame and $l^{th}$ trial: $\mathbf{y}_{t,l}$, are noisy surrogates of the intracellular calcium concentrations: $\mathbf{z}_{t,l}$. The calcium concentration at time $t$ is a function of the spiking activity $\mathbf{n}_{t,l}$, and the calcium activity at the previous time point $\mathbf{z}_{t-1,l}$. The spiking activity is driven by two independent mechanisms: latent trial-dependent covariates $\mathbf{x}_{t,l}$, and contributions from the known external stimulus $\mathbf{s}_t$, which we model by $\mathbf{D}^\top \mathbf{s}_t$ (in which the receptive field $\mathbf{D}$ is unknown). Then, we model $\mathbf{x}_{t,l}$ as a Gaussian process with constant mean $\boldsymbol{\mu}_x$, and unknown covariance $\boldsymbol{\Sigma}_x$. Finally, we assume the covariance $\boldsymbol{\Sigma}_x$ to have an inverse Wishart prior distribution with hyper-parameters $\boldsymbol{\psi}_x$ and $\rho_x$. Based on this forward model, the inverse problem amounts to recovering the signal and noise correlations by directly estimating $\boldsymbol{\Sigma}_x$ and $\mathbf{D}$ (top layer) from the fluorescence observations $\{\mathbf{y}_{t,l}\}_{t=1,l=1}^{T,L}$ (bottom layer).

The main problem is thus to estimate $\{\boldsymbol{\Sigma}_x, \mathbf{D}\}$ from the noisy and temporally blurred data $\{\mathbf{y}_{t,l}\}_{t=1,l=1}^{T,L}$ .

## Overview of the proposed estimation method

First, given a limited number of trials $L$ from an ensemble with typically low spiking rates, we need to incorporate suitable prior assumptions to avoid overfitting. Thus, we impose a prior $p_{pr}(\boldsymbol{\Sigma}_x)$ on the noise covariance, to compensate sparsity of data. A natural estimation method to estimate $\{\boldsymbol{\Sigma}_x, \mathbf{D}\}$ in a Bayesian framework is to maximize the observed data likelihood $p\big(\{\mathbf{y}_{t,l}\}_{t,l=1}^{T,L} \big| \boldsymbol{\Sigma}_x, \mathbf{D}\big)$, that is maximum likelihood (ML). Thus, we consider the joint likelihood of the observed data and latent processes to perform Maximum a Posteriori (MAP) estimation:

$$p(\mathbf{y}, \mathbf{z}, \mathbf{x}, \boldsymbol{\Sigma}_x | \mathbf{D}) = p_{pr}(\boldsymbol{\Sigma}_x) \prod_{t,l=1}^{T,L} \frac{1}{\sqrt{(2\pi)^N |\boldsymbol{\Sigma}_w|}} \exp\left(-\frac{1}{2}(\mathbf{y}_{t,l} - \mathbf{A}\mathbf{z}_{t,l})^\top \boldsymbol{\Sigma}_w^{-1}(\mathbf{y}_{t,l} - \mathbf{A}\mathbf{z}_{t,l})\right)$$

$$\times \prod_{t,l,j=1}^{T,L,N} \frac{\left(\exp\left(x_{t,l}^{(j)} + \mathbf{d}_j^\top \mathbf{s}_t\right)\right)^{z_{t,l}^{(j)} - \alpha z_{t-1,l}^{(j)}}}{1 + \exp\left(x_{t,l}^{(j)} + \mathbf{d}_j^\top \mathbf{s}_t\right)} \prod_{t,l=1}^{T,L} \frac{1}{\sqrt{(2\pi)^N |\boldsymbol{\Sigma}_x|}} \exp\left(-\frac{1}{2}(\mathbf{x}_{t,l} - \boldsymbol{\mu}_x)^\top \boldsymbol{\Sigma}_x^{-1}(\mathbf{x}_{t,l} - \boldsymbol{\mu}_x)\right).$$

(4)

Inspecting this MAP problem soon reveals that estimating $\boldsymbol{\Sigma}_x$ and $\mathbf{D}$ is a challenging task: (1) standard approaches such as Expectation-Maximization (EM) (*Shumway and Stoffer, 1982*) are intractable due to the complexity of the model, arising from the hierarchy of latent processes and the nonlinearities involved in their mappings and (2) the temporal coupling of the likelihood in the calcium concentrations makes any potential direct solver scale poorly with $T$.

Thus, we propose an alternative solution based on Variational Inference (VI) (*Beal, 2003*; *Blei et al., 2017*; *Jordan et al., 1999*). VI is a method widely used in Bayesian statistics to approximate unwieldy posterior densities using optimization techniques, as a low-complexity alternative strategy to Markov Chain Monte Carlo sampling (*Hastings, 1970*) or empirical Bayes techniques such as EM. To this end, we treat $\{\mathbf{x}_{t,l}\}_{t,l=1}^{T,L}$ and $\boldsymbol{\Sigma}_x$ as latent variables and $\{\mathbf{z}_{t,l}\}_{t,l=1}^{T,L}$ and $\mathbf{D}$ as unknown

parameters to be estimated. We introduce a framework to update the latent variables and parameters sequentially, with straightforward update rules. We will describe the main ingredients of the proposed framework in the following subsections. Hereafter, we use the shorthand notations $\mathbf{y} := \{\mathbf{y}_{t,l}\}_{t,l=1}^{T,L}$, $\mathbf{z} := \{\mathbf{z}_{t,l}\}_{t,l=1}^{T,L}$, and $\mathbf{x} := \{\mathbf{x}_{t,l}\}_{t,l=1}^{T,L}$.

## Preliminary assumptions

For the sake of simplicity, we assume that the constants $\alpha$, $\mathbf{A}$, $\mathbf{\Sigma}_w$ and $\mathbf{\mu}_x$ are either known or can be consistently estimated from pilot trials. Next, we take $p_{pr}(\mathbf{\Sigma}_x)$ to be an Inverse Wishart density:

$$\mathbf{\Sigma}_x \sim InvWish_N(\mathbf{\psi}_x, \rho_x),$$

which turns out to be the conjugate prior in our model. Thus, $\mathbf{\psi}_x$ and $\rho_x$ will be the hyper-parameters of our model. Procedures for hyper-parameter tuning and choosing the key model parameters are given in subsections Hyper-parameter tuning and Guidelines for model parameter settings, respectively.

## Decoupling via Pólya-Gamma augmentation

Direct application of VI to problems containing both discrete and continuous random variables results in intractable densities. Specifically, finding a variational distribution for $\mathbf{x}_{t,l}$ in our model with a standard distribution is not straightforward, due to the complicated posterior arising from co-dependent Bernoulli and Gaussian random variables. In order to overcome this difficulty, we employ Pólya-Gamma (PG) latent variables (*Pillow and Scott, 2012*; *Polson et al., 2013*; *Linderman et al., 2016*). We observe from *Equation 4* that the posterior density, $p(\mathbf{x}|\mathbf{z}, \mathbf{D}, \mathbf{\Sigma}_x)$ is conditionally independent in $t, l$ with:

$$p(\mathbf{x}_{t,l}|\mathbf{z}, \mathbf{D}, \mathbf{\Sigma}_x) \propto p(\mathbf{x}_{t,l}|\mathbf{\Sigma}_x) \prod_{j=1}^{N} \frac{\left(\exp\left(x_{t,l}^{(j)} + \mathbf{d}_j^{\top}\mathbf{s}_t\right)\right)^{z_{t,l}^{(j)} - \alpha z_{t-1,l}^{(j)}}}{1 + \exp\left(x_{t,l}^{(j)} + \mathbf{d}_j^{\top}\mathbf{s}_t\right)}.$$

Thus, upon careful inspection, we see that this density has the desired form for the PG augmentation scheme (*Polson et al., 2013*). Accordingly, we introduce a set of auxiliary PG-distributed i.i.d. latent random variables $\mathbf{\omega}_{t,l} := [\omega_{t,l}^{(1)}, \omega_{t,l}^{(2)}, \cdots, \omega_{t,l}^{(N)}]^{\top}$, $\omega_{t,l}^{(j)} \sim PG(1,0)$ for $1 \le j \le N$, $1 \le t \le T$ and $1 \le l \le L$, to derive the complete data log-likelihood:

$$\log p(\mathbf{y}, \mathbf{z}, \mathbf{x}, \mathbf{\omega}, \mathbf{\Sigma}_x|\mathbf{D})$$
$$= -\frac{TL}{2}\log|\mathbf{\Sigma}_x| + \log p_{pr}(\mathbf{\Sigma}_x) + \sum_{t,l=1}^{T,L}\left\{ -\frac{1}{2}\left(\mathbf{y}_{t,l} - \mathbf{A}\mathbf{z}_{t,l}\right)^{\top}\mathbf{\Sigma}_w^{-1}\left(\mathbf{y}_{t,l} - \mathbf{A}\mathbf{z}_{t,l}\right) - \frac{1}{2}\left(\mathbf{x}_{t,l} - \mathbf{\mu}_x\right)^{\top}\mathbf{\Sigma}_x^{-1}\left(\mathbf{x}_{t,l} - \mathbf{\mu}_x\right) \right.$$
$$\left. + \sum_{j=1}^{N}\left\{\left(z_{t,l}^{(j)} - \alpha z_{t-1,l}^{(j)} - \frac{1}{2}\right)\left(x_{t,l}^{(j)} + \mathbf{d}_j^{\top}\mathbf{s}_t\right) - \frac{1}{2}\omega_{t,l}^{(j)}\left(x_{t,l}^{(j)} + \mathbf{d}_j^{\top}\mathbf{s}_t\right)^2 + \log p_{PG(1,0)}\left(\omega_{t,l}^{(j)}\right)\right\}\right\} + C,$$

(5)

where $\mathbf{\omega} := \{\mathbf{\omega}_{t,l}\}_{t,l=1}^{T,L}$ and $C$ accounts for terms not depending on $\mathbf{y}, \mathbf{z}, \mathbf{x}, \mathbf{\omega}, \mathbf{\Sigma}_x$ and $\mathbf{D}$. The complete data log-likelihood is notably *quadratic* in $\mathbf{z}_{t,l}$, which as we show later admits efficient estimation procedures with favorable scaling in $T$.

## Deriving the optimal variational densities

In this section, we will outline the procedure of applying VI to the latent variables $\mathbf{x} = \{\mathbf{x}_{t,l}\}_{t,l=1}^{T,L}$, $\mathbf{\omega} = \{\mathbf{\omega}_{t,l}\}_{t,l=1}^{T,L}$ and $\mathbf{\Sigma}_x$, assuming that the parameter estimates $\widehat{\mathbf{z}}$ and $\widehat{\mathbf{D}}$ of the previous iteration are available. The methods that we propose to update the parameters $\widehat{\mathbf{z}}$ and $\widehat{\mathbf{D}}$ subsequently, will be discussed in the next section.

The objective of variational inference is to posit a family of approximate densities $\mathcal{Q}$ over the latent variables, and to find the member of that family that minimizes the Kullback-Leibler (KL) divergence to the exact posterior:

$$q^*(\mathbf{x}, \mathbf{\omega}, \mathbf{\Sigma}_x|\widehat{\mathbf{z}}, \widehat{\mathbf{D}}) = \underset{q \in \mathcal{Q}}{argmin}\, KL\left(q(\mathbf{x}, \mathbf{\omega}, \mathbf{\Sigma}_x|\widehat{\mathbf{z}}, \widehat{\mathbf{D}}) \,\middle\|\, p(\mathbf{x}, \mathbf{\omega}, \mathbf{\Sigma}_x|\mathbf{y}, \widehat{\mathbf{z}}, \widehat{\mathbf{D}})\right).$$

However, evaluating the KL divergence is intractable, and it has been shown (*Blei et al., 2017*) that an equivalent result to this minimization can be obtained by maximizing the alternative objective function, called the evidence lower bound (ELBO):

$$ELBO(q) = \mathbb{E}[\log p(\mathbf{x}, \boldsymbol{\omega}, \boldsymbol{\Sigma}_x, \mathbf{y} | \widehat{\mathbf{z}}, \widehat{\mathbf{D}})] - \mathbb{E}[\log q(\mathbf{x}, \boldsymbol{\omega}, \boldsymbol{\Sigma}_x | \widehat{\mathbf{z}}, \widehat{\mathbf{D}})].$$

Further, we assume $\mathcal{Q}$ to be a mean-field variational family (*Blei et al., 2017*), resulting in the overall variational density of the form:

$$q(\mathbf{x}, \boldsymbol{\omega}, \boldsymbol{\Sigma}_x) = q(\boldsymbol{\Sigma}_x) \prod_{t,l=1}^{T,L} \left( q(\mathbf{x}_{t,l}) \prod_{j=1}^{N} q\left(\omega_{t,l}^{(j)}\right) \right). \tag{6}$$

Under the mean field assumptions, the maximization of the ELBO can be derived using the optimization algorithm 'Coordinate Ascent Variational Inference' (CAVI) (*Bishop, 2006*; *Blei et al., 2017*). Accordingly, we see that the optimal variational densities in *Equation 6* take the forms:

$$\begin{aligned}
\log q^*(\mathbf{x}_{t,l}) &\propto \mathbb{E}_{q^*(\boldsymbol{\Sigma}_x) q^*(\boldsymbol{\omega}_{t,l})}\left[\log p\left(\mathbf{x}_{t,l} | \boldsymbol{\omega}_{t,l}, \boldsymbol{\Sigma}_x, \mathbf{y}, \widehat{\mathbf{z}}, \widehat{\mathbf{D}}\right)\right], \\
\log q^*\left(\omega_{t,l}^{(j)}\right) &\propto \mathbb{E}_{q^*(\mathbf{x}_{t,l})}\left[\log p\left(\omega_{t,l}^{(j)} | \mathbf{x}_{t,l}, \boldsymbol{\Sigma}_x, \mathbf{y}, \widehat{\mathbf{z}}, \widehat{\mathbf{D}}\right)\right], \\
\log q^*(\boldsymbol{\Sigma}_x) &\propto \mathbb{E}_{q^*(\mathbf{x})}\left[\log p\left(\boldsymbol{\Sigma}_x | \mathbf{x}, \mathbf{y}, \widehat{\mathbf{z}}, \widehat{\mathbf{D}}\right)\right].
\end{aligned}$$

Upon evaluation of these expectations, we derive the optimal variational distributions as:

$$q^*(\mathbf{x}_{t,l}) \sim \mathcal{N}(\mathbf{m}_{\mathbf{x}_{t,l}}, \mathbf{Q}_{\mathbf{x}_{t,l}}), \quad q^*(\omega_{t,l}^{(j)}) \sim PG(1, c_{t,l}^{(j)}), \quad q^*(\boldsymbol{\Sigma}_x) \sim \mathrm{InvWish}_N(\mathbf{P}_x, \gamma_x),$$

whose parameters $\mathbf{m}_{\mathbf{x}_{t,l}} := [m_{\mathbf{x}_{t,l}}^{(1)}, m_{\mathbf{x}_{t,l}}^{(2)}, \cdots, m_{\mathbf{x}_{t,l}}^{(N)}]^T$, $\mathbf{Q}_{\mathbf{x}_{t,l}}$, $c_{t,l}^{(j)}$, $\mathbf{P}_x$, and $\gamma_x$ can be updated given parameter estimates $\widehat{\mathbf{D}}$ and $\widehat{\mathbf{z}}$:

$$\mathbf{Q}_{\mathbf{x}_{t,l}} = (\widetilde{\boldsymbol{\Omega}}_{t,l} + \gamma_x \mathbf{P}_x^{-1})^{-1}, \quad \mathbf{m}_{\mathbf{x}_{t,l}} = \mathbf{Q}_{\mathbf{x}_{t,l}}\left(\widehat{\mathbf{z}}_{t,l} - \alpha\widehat{\mathbf{z}}_{t-1,l} - \frac{1}{2}\mathbf{1} - \widetilde{\boldsymbol{\Omega}}_{t,l}\widehat{\mathbf{D}}^\top \mathbf{s}_t + \gamma_x \mathbf{P}_x^{-1}\boldsymbol{\mu}_x\right),$$

$$\mathbf{P}_x := \boldsymbol{\psi}_x + \sum_{t,l=1}^{T,L}\left\{\mathbf{Q}_{\mathbf{x}_{t,l}} + \mathbf{m}_{\mathbf{x}_{t,l}}\mathbf{m}_{\mathbf{x}_{t,l}}^\top - \boldsymbol{\mu}_x\mathbf{m}_{\mathbf{x}_{t,l}}^\top - \mathbf{m}_{\mathbf{x}_{t,l}}\boldsymbol{\mu}_x^\top + \boldsymbol{\mu}_x\boldsymbol{\mu}_x^\top\right\}, \quad c_{t,l}^{(j)} = \sqrt{\left(\mathbf{Q}_{\mathbf{x}_{t,l}}\right)_{j,j} + \left(m_{\mathbf{x}_{t,l}}^{(j)} + \widehat{\mathbf{d}}_j^\top \mathbf{s}_t\right)^2},$$

and $\gamma_x := \rho_x + TL$, with $\widetilde{\boldsymbol{\Omega}}_{t,l} \in \mathbb{R}^{N \times N}$ denoting a diagonal matrix with entries $(\widetilde{\boldsymbol{\Omega}}_{t,l})_{j,j} := \frac{1}{2c_{t,l}^{(j)}} tanh\left(\frac{c_{t,l}^{(j)}}{2}\right)$

and $\mathbf{1} \in \mathbb{R}^N$ denoting the vector of all ones.

## Low-complexity parameter updates

Note that even though $\mathbf{z}$ is composed of the latent processes $\mathbf{z}_{t,l}$, we do not use VI for its inference, and instead consider it as an unknown parameter. This choice is due to the temporal dependencies arising from the underlying state-space model in *Equation 4*, which hinders a proper assignment of variational densities under the mean field assumption. We thus seek to estimate both $\mathbf{z}$ and $\mathbf{D}$ using the updated variational density $q^*(\mathbf{x}, \boldsymbol{\omega}, \boldsymbol{\Sigma}_x)$.

First, note that the log-likelihood in *Equation 5* is decoupled in $l$, which admits independent updates to $\{\mathbf{z}_{t,l}\}_{t=1}^{T}$, for $l = 1, \cdots, L$. As such, given an estimate $\widehat{\mathbf{D}}$, we propose to estimate $\{\mathbf{z}_{t,l}\}_{t=1}^{T}$ as:

$$\begin{aligned}
\{\widehat{\mathbf{z}}_{t,l}\}_{t=1}^{T} &= \underset{\{\mathbf{z}_{t,l}\}_{t=1}^{T}}{argmax} \quad \mathbb{E}_{q^*(\mathbf{x}, \boldsymbol{\omega}, \boldsymbol{\Sigma}_x)}\left[\log p\left(\mathbf{y}, \mathbf{z}, \mathbf{x}, \boldsymbol{\omega}, \boldsymbol{\Sigma}_x | \widehat{\mathbf{D}}\right)\right] \\
&= \underset{\{\mathbf{z}_{t,l}\}_{t=1}^{T}}{argmin} \quad \sum_{t=1}^{T}\left\{\frac{1}{2}\left(\mathbf{y}_{t,l} - \mathbf{A}\mathbf{z}_{t,l}\right)^\top \boldsymbol{\Sigma}_w^{-1}\left(\mathbf{y}_{t,l} - \mathbf{A}\mathbf{z}_{t,l}\right) - \sum_{j=1}^{N}\left(m_{\mathbf{x}_{t,l}}^{(j)} + \widehat{\mathbf{d}}_j^\top \mathbf{s}_t\right)\left(z_{t,l}^{(j)} - \alpha z_{t-1,l}^{(j)}\right)\right\},
\end{aligned}$$

under the constraints $0 \leq z_{t,l}^{(j)} - \alpha z_{t-1,l}^{(j)} \leq 1$, for $t = 1, \cdots, T$ and $j = 1, \cdots, N$. These constraints are a direct consequence of $n_{t,l}^{(j)} = z_{t,l}^{(j)} - \alpha z_{t-1,l}^{(j)}$ being a Bernoulli random variable with $\mathbb{E}\left[n_{t,l}^{(j)}\right] \in [0, 1]$. While this problem is a quadratic program and can be solved using standard techniques, it is not readily decoupled in $t$, and thus standard solvers would not scale favorably in $T$.

Instead, we consider an alternative solution that admits a low-complexity recursive solution by relaxing the constraints. To this end, we relax the constraint $\mathbf{z}_{t,l} - \alpha\mathbf{z}_{t-1,l} \preceq \mathbf{1}$ and replace the

constraint $\mathbf{z}_{t,l} - \alpha\mathbf{z}_{t-1,l} \succeq \mathbf{0}$ by penalty terms proportional to $|z_{t,l}^{(j)} - \alpha z_{t-1,l}^{(j)}|$. The resulting relaxed problem is thus given by:

$$\min_{\{\mathbf{z}_{t,l}\}_{t=1}^T} \quad \sum_{t=1}^T \left\{ \frac{1}{2}\left(\mathbf{y}_{t,l} - \mathbf{A}\mathbf{z}_{t,l}\right)^\top \boldsymbol{\Sigma}_w^{-1}\left(\mathbf{y}_{t,l} - \mathbf{A}\mathbf{z}_{t,l}\right) + \sum_{j=1}^N \nu_{t,l}^{(j)}\left|z_{t,l}^{(j)} - \alpha z_{t-1,l}^{(j)}\right| \right\}, \tag{7}$$

where $\nu_{t,l}^{(j)} := \beta\,|m_{\mathbf{x}_{t,l}}^{(j)} + \widehat{\mathbf{d}}_j^\top \mathbf{s}_t|$ with $\beta \geq 1$ being a hyper-parameter. Given that the typical spiking rates are quite low in practice, $m_{\mathbf{x}_{t,l}}^{(j)} + \widehat{\mathbf{d}}_j^\top \mathbf{s}_t$ is expected to be a negative number. Thus, we have assumed that $-m_{\mathbf{x}_{t,l}}^{(j)} - \widehat{\mathbf{d}}_j^\top \mathbf{s}_t = |m_{\mathbf{x}_{t,l}}^{(j)} + \widehat{\mathbf{d}}_j^\top \mathbf{s}_t|$.

The problem of *Equation 7* pertains to *compressible* state-space estimation, for which fast recursive solvers are available (*Kazemipour et al., 2018*). The solver utilizes the Iteratively Re-weighted Least Squares (IRLS) (*Ba et al., 2014*) framework to transform the absolute value in the second term of the cost function into a quadratic form in $\mathbf{z}_{t,l}$, followed by Fixed Interval Smoothing (FIS) (*Rauch et al., 1965*) to find the minimizer. At iteration $k$, given a current estimate $\mathbf{z}^{[k-1]}$, the problem reduces to a Gaussian state-space estimation of the form:

$$\mathbf{y}_{t,l} = \mathbf{A}\mathbf{z}_{t,l} + \mathbf{w}_{t,l}, \quad \mathbf{z}_{t,l} = \alpha\mathbf{z}_{t-1,l} + \mathbf{v}_{t,l}, \tag{8}$$

with $\mathbf{w}_{t,l} \sim \mathcal{N}(0, \boldsymbol{\Sigma}_w)$ and $\mathbf{v}_{t,l} \sim \mathcal{N}(0, \boldsymbol{\Sigma}_{\mathbf{v}_{t,l}}^{[k]})$, where $\boldsymbol{\Sigma}_{\mathbf{v}_{t,l}}^{[k]} \in \mathbb{R}^{N \times N}$ is a diagonal matrix with $(\boldsymbol{\Sigma}_{\mathbf{v}_{t,l}}^{[k]})_{j,j} := \sqrt{\left(\widehat{z}_{t,l}^{(j)[k-1]} - \alpha\widehat{z}_{t-1,l}^{(j)[k-1]}\right)^2 + \varepsilon^2}\,/\,\nu_{t,l}^{(j)}$, for some small constant $\varepsilon > 0$. This problem can be efficiently solved using FIS, and the iterations proceed for a total of $K$ times or until a standard convergence criterion is met (*Kazemipour et al., 2018*). It is noteworthy that our proposed estimator of the calcium concentration $\mathbf{z}_{t,l}$ can be thought of as *soft* spike deconvolution, which naturally arises from our variational framework, as opposed to the *hard* spike deconvolution step used in two-stage estimators.

Finally, given $q^*(\mathbf{x}, \boldsymbol{\omega}, \boldsymbol{\Sigma}_x)$ and the updated $\widehat{\mathbf{z}}$, the estimate of $\mathbf{d}_j$ for $j = 1, 2, \cdots, N$ can be updated in closed-form by maximizing the expected complete log-likelihood $\mathbb{E}_{q^*(\mathbf{x}, \boldsymbol{\omega}, \boldsymbol{\Sigma}_x)}[\log p(\mathbf{y}, \widehat{\mathbf{z}}, \mathbf{x}, \boldsymbol{\omega}, \boldsymbol{\Sigma}_x | \mathbf{D})]$:

$$\widehat{\mathbf{d}}_j = \left( \sum_{t,l=1}^{T,L} \left( (\widetilde{\boldsymbol{\Omega}}_{t,l})_{j,j}\mathbf{s}_t\mathbf{s}_t^\top \right) \right)^{-1} \left( \sum_{t,l=1}^{T,L} \left\{ \left( \widehat{z}_{t,l}^{(j)} - \alpha\widehat{z}_{t-1,l}^{(j)} - \frac{1}{2} \right)\mathbf{s}_t - (\widetilde{\boldsymbol{\Omega}}_{t,l})_{j,j}m_{\mathbf{x}_{t,l}}^{(j)}\mathbf{s}_t \right\} \right).$$

The VI procedure iterates between updating the variational densities and parameters until convergence, upon which we estimate the noise and signal covariances as:

$$\widehat{\boldsymbol{\Sigma}}_x := mode\{q^*(\boldsymbol{\Sigma}_x)\} = \frac{\mathbf{P}_x}{\gamma_x + N + 1}, \qquad \widehat{\boldsymbol{\Sigma}}_s := \widehat{\mathbf{D}}^\top \mathbb{E}[\mathbf{s}_t\mathbf{s}_t^\top]\widehat{\mathbf{D}}.$$

The overall combined iterative procedure is outlined in Algorithm 1. Furthermore, a MATLAB implementation of this algorithm is publicly available in *Rupasinghe, 2020*. It is worth noting that a special case of our proposed variational inference procedure can be used to estimate signal and noise correlations from electrophysiology recordings. Given that spiking activity, that is $\{\mathbf{n}_{t,l}\}_{t,l=1}^{T,L}$, is directly observed in this case, the solution to the optimization problem in *Equation (7)* is no longer required. Thus, the parameters $\boldsymbol{\Sigma}_x$ and $\mathbf{D}$ can be estimated using a simplified variational procedure, which is outlined in Algorithm 2 in Appendix 3.

## Guidelines for model parameter settings

There are several key model parameters that need to be set by the user prior to the application of our proposed method. Here, we provide our rationale and criteria for choosing these parameters, which could also serve as guidelines in facilitating the applicability and adoption of our method by future users. We will also provide the specific choices of these parameters used in our simulation studies and real data analyses.

### Number of neurons selected for the analysis (N)

While our proposed method scales-up well with the population size due to low-complexity update rules involved, including neurons with negligible spiking activity in the analysis would only increase

the complexity and potentially contaminate the correlation estimates. Thus, we performed an initial pre-processing step to extract $N$ neurons that exhibited at least one spiking event in at least half of the trials considered.

## Stimulus integration window length ($R$)

The number of lags $R$ considered in stimulus integration is a key parameter that can be set through data-driven approaches or using prior domain knowledge. Examples of common data-driven criteria include cross-validation, Akaike Information Criterion (AIC) and Bayesian Information Criterion (BIC), which balance the estimation accuracy and model complexity (**Arlot and Celisse, 2010**; **Ding et al., 2018**).

To quantify the effect of $R$ on model complexity, we first describe the stimulus encoding model in our framework. Suppose that the onset of the $p^{th}$ tone in the stimulus set ($p = 1, \cdots, P$, where $P$ is the number of distinct tones) is given by a binary sequence $f_t^{(p)} \in \{0, 1\}$. The choice of $R$ implies that the response at time $t$ post-stimulus depends only on the $R$ most recent time lags. As such, the effective stimulus at time $t$ corresponding to tone $p$ is given by $\mathbf{s}_t^{(p)} := [f_t^{(p)}, f_{t-1}^{(p)}, \cdots, f_{t-R+1}^{(p)}]^\top \in \mathbb{R}^R$. By including all the $P$ tones, the overall effective stimulus at the $t^{th}$ time frame is given by $\mathbf{s}_t := [\mathbf{s}_t^{(1)\top}, \cdots, \mathbf{s}_t^{(P)\top}]^\top \in \mathbb{R}^{RP}$. The stimulus modulation vector $\mathbf{d}_j$ would thus be $RP$-dimensional. As a result, the number of parameters ($M = RP$) to be estimated linearly increases with $R$. By using additional domain knowledge, we chose $R$ to be large enough to capture the stimulus effects, and at the same time to be small enough to control the complexity of the algorithm.

As an example, given that the typical tone response duration of mouse primary auditory neurons is < 1 s (**Linden et al., 2003**; **DeWeese et al., 2003**; **Petrus et al., 2014**), with a sampling frequency of $f_s = 30$ Hz, a choice of $R \sim 30$ would suffice to capture the stimulus effects. By further examining the effect of varying $R$ on the proposed correlation estimates in **Figure 4—figure supplement 1**, we chose $R = 25$ for our real data analyses.

## Observation noise covariance ($\boldsymbol{\Sigma}_w$) and scaling matrix ($\mathbf{A}$)

We assumed that the observation noise covariance $\boldsymbol{\Sigma}_w$ is diagonal, and estimated the diagonal elements using the background fluorescence in the absence of spiking events, for each neuron. We set $\mathbf{A} = a\mathbf{I}$, where $\mathbf{I} \in \mathbb{R}^{N \times N}$ represents the identity matrix, and estimated $a$ by considering the average increase in fluorescence after the occurrence of isolated spiking events. Specifically, we derived the average fluorescence activity of multiple trials triggered to the fluorescence rise onset, and set $a$ as the increment in the magnitude of this average fluorescence immediately following the rise onset.

## State transition parameter ($\alpha$)

We chose $\alpha$ in the range $[0.95, 0.98]$, which match the slow dynamics of the calcium indicator in our data. We tested the robustness of our estimates under different choices of $\alpha$ in this range through the method outlined in Hyper-parameter tuning, and accordingly chose the optimal value of $\alpha$.

## Mean of the latent trial-dependent process ($\boldsymbol{\mu}_x$)

We estimated $\boldsymbol{\mu}_x$ as a constant that is proportional to the average firing rate. To this end, we parametrized each component of $\boldsymbol{\mu}_x$ as $\mu_x^{(j)} = -a_\mu + b_\mu \frac{1}{LT} \sum_{t,l=1}^{T,L} y_{t,l}^{(j)}$, for $j = 1, \cdots, N$. The constants $a_\mu$ and $b_\mu$ were chosen such that $-2 \leq \mu_x^{(j)} \leq -10$, which gives the range of baseline parameters compatible with observed firing rates in our experimental data.

## Parameter choices for simulation study 1

In the first simulation study, we set $\alpha = 0.98$, $\beta = 8$, $\mathbf{A} = 0.1\mathbf{I}$, $\boldsymbol{\mu}_x = -4.5\mathbf{1}$ and $\boldsymbol{\Sigma}_w = 2 \times 10^{-4}\mathbf{I}$ ($\mathbf{I} \in \mathbb{R}^{8 \times 8}$ represents the identity matrix and $\mathbf{1} \in \mathbb{R}^8$ represents the vector of all ones), so that the SNR of simulated data was in the same range as that of experimentally-recorded data. We used a $6^{th}$ order autoregressive process with a mean of -1 as the stimulus ($s_t$), and considered $R = 2$ ($M = 2$) lags of the stimulus (i.e. $\mathbf{s}_t = [s_t, s_{t-1}]^\top$) in both the generative model and inference procedure. The components of the linear and quadratic stimulus modulation vectors, that is $\mathbf{d}_j$, $\widetilde{\mathbf{d}}_{j,1}$ and $\widetilde{\mathbf{d}}_{j,2}$, were chosen at random uniformly in the range $[-0.5, 0.5]$. The variance of $s_t$ was set in each case such that

the average power of the overall signal component ($\mathbf{d}_j^\top \mathbf{s}_t$ for the linear model, and $\mathbf{d}_j^\top \mathbf{s}_t + (\widetilde{\mathbf{d}}_{j,1}{}^\top \mathbf{s}_t)^2 + (\widetilde{\mathbf{d}}_{j,2}{}^\top \mathbf{s}_t)^2$ for the non-linear model) was comparable to the average power of the noise component ($x_{t,l}^{(j)}$).

---

**Algorithm 1 Estimation of $\boldsymbol{\Sigma}_x$ and $\mathbf{D}$ through the proposed iterative procedure**

---

**Inputs:** Ensemble of fluorescence measurements $\{\mathbf{y}_{t,l}\}_{t,l=1}^{T,L}$, constants $\alpha, \mathbf{A}, \boldsymbol{\Sigma}_w$ and $\boldsymbol{\mu}_x$, hyper-parameters $\boldsymbol{\psi}_x, \rho_x, \beta$ and $\epsilon$, tolerance at convergence $\delta$ and the external stimulus $\mathbf{s}_t$

**Outputs:** $\widehat{\boldsymbol{\Sigma}}_x$ and $\widehat{\mathbf{D}}$

**Initialization:** Initial choice of $\boldsymbol{\Sigma}_{\mathbf{v}_{t,l}}, \widetilde{\boldsymbol{\Omega}}_t, \widehat{\boldsymbol{\Sigma}}_x$ and $\widehat{\mathbf{D}}$, residual $= 10\delta$, $\gamma_x = \rho_x + LT$

1: **while** residual $\geq \delta$ **do**

Estimate calcium concentrations using Fixed Interval Smoothing

2: **for** $l = 1, \cdots, L$ **do**

Forward filter:

3: **for** $t = 1, ..., T$ **do**

4: $\mathbf{z}_{(t|t-1),l} = \alpha\, \mathbf{z}_{(t-1|t-1),l}$

5: $\mathbf{P}_{(t|t-1),l} = \alpha^2\, \mathbf{P}_{(t-1|t-1),l} + \boldsymbol{\Sigma}_{\mathbf{v}_{t,l}}$

6: $\mathbf{B}_{t,l} = \mathbf{P}_{(t|t-1),l}\mathbf{A}^\top (\mathbf{A}\mathbf{P}_{(t|t-1),l}\mathbf{A}^\top + \boldsymbol{\Sigma}_w)^{-1}$

7: $\mathbf{z}_{(t|t),l} = \mathbf{z}_{(t|t-1),l} + \mathbf{B}_{t,l}(\mathbf{y}_{t,l} - \mathbf{A}\,\mathbf{z}_{(t|t-1),l})$

8: $\mathbf{P}_{(t|t),l} = (\mathbf{I} - \mathbf{B}_{t,l}\,\mathbf{A})\mathbf{P}_{(t|t-1),l}$

9: **end for**

Backward smoother:

10: **for** $t = T-1, ..., 1$ **do**

11: $\hat{\mathbf{z}}_{t,l} = \mathbf{z}_{(t|t),l} + \alpha\, \mathbf{P}_{(t|t),l}\, \mathbf{P}_{(t+1|t),l}^{-1}(\hat{\mathbf{z}}_{t+1,l} - \mathbf{z}_{(t+1|t),l})$

12: **end for**

13: **end for**

Update variational parameters

14: **for** $t = 1, \cdots, T$ and $l = 1, \cdots, L$ **do**

15: $\mathbf{Q}_{\mathbf{x}_{t,l}} = (\widetilde{\boldsymbol{\Omega}}_{t,l} + \gamma_x \mathbf{P}_x^{-1})^{-1}$

16: $\mathbf{m}_{\mathbf{x}_{t,l}} = \mathbf{Q}_{\mathbf{x}_{t,l}}(\hat{\mathbf{z}}_{t,l} - \alpha\hat{\mathbf{z}}_{t-1,l} - \frac{1}{2}\mathbf{1} - \widetilde{\boldsymbol{\Omega}}_{t,l}\widehat{\mathbf{D}}^\top \mathbf{s}_t + \gamma_x \mathbf{P}_x^{-1}\boldsymbol{\mu}_x)$

17: $v_{t,l}^{(j)} := \beta|m_{\mathbf{x}_{t,l}}^{(j)} + \hat{\mathbf{d}}_j^\mathrm{T}\mathbf{s}_t|$

18: **for** $j = 1, \cdots, N$ **do**

19: $c_{t,l}^{(j)} = \sqrt{\left(\mathbf{Q}_{\mathbf{x}_{t,l}}\right)_{j,j} + \left(m_{\mathbf{x}_{t,l}}^{(j)} + \hat{\mathbf{d}}_j{}^\top \mathbf{s}_t\right)^2}$

20: $(\widetilde{\boldsymbol{\Omega}}_{t,l})_{j,j} := \dfrac{1}{2c_{t,l}^{(j)}}\tanh\left(\dfrac{c_{t,l}^{(j)}}{2}\right)$

21: **end for**

22: **end for**

23: $\mathbf{P}_x := \boldsymbol{\psi}_x + \sum\limits_{t,l=1}^{T,L}\left\{\mathbf{Q}_{\mathbf{x}_{t,l}} + \mathbf{m}_{\mathbf{x}_{t,l}}\mathbf{m}_{\mathbf{x}_{t,l}}^\top - \boldsymbol{\mu}_x\mathbf{m}_{\mathbf{x}_{t,l}}^\top - \mathbf{m}_{\mathbf{x}_{t,l}}\boldsymbol{\mu}_x^\top + \boldsymbol{\mu}_x\boldsymbol{\mu}_x^\top\right\}$

Update IRLS covariance approximation

24: **for** $l = 1, \cdots, L, t = 1, \cdots, T$ and $j = 1, \cdots, N$ **do**

25: $(\boldsymbol{\Sigma}_{\mathbf{v}_{t,l}})_{j,j} := \sqrt{(\hat{z}_{t,l}^{(j)} - \alpha\hat{z}_{t-1,l}^{(j)})^2 + \varepsilon^2}/v_{t,l}^{(j)}$

26: **end for**

Update outputs and the convergence criterion

27: **for** $j = 1, \cdots, N$ **do**

28: $\hat{\mathbf{d}}_j = \left(\sum\limits_{t,l=1}^{T,L}\left((\widetilde{\boldsymbol{\Omega}}_{t,l})_{j,j}\mathbf{s}_t\mathbf{s}_t^\top\right)\right)^{-1}\left(\sum\limits_{t,l=1}^{T,L}\left\{\left(\hat{z}_{t,l}^{(j)} - \alpha\hat{z}_{t-1,l}^{(j)} - \frac{1}{2}\right)\mathbf{s}_t - (\widetilde{\boldsymbol{\Omega}}_{t,l})_{j,j}m_{\mathbf{x}_{t,l}}^{(j)}\mathbf{s}_t\right\}\right)$

29: **end for**

30: $(\widehat{\mathbf{D}})_{prev} = \widehat{\mathbf{D}}, \widehat{\mathbf{D}} = \left[\hat{\mathbf{d}}_1, \hat{\mathbf{d}}_2, \cdots, \hat{\mathbf{d}}_N\right]$

31: $(\widehat{\boldsymbol{\Sigma}}_x)_{prev} = \widehat{\boldsymbol{\Sigma}}_x, \widehat{\boldsymbol{\Sigma}}_x = \dfrac{\mathbf{P}_x}{\gamma_x + N + 1}$

32: residual $= \|(\widehat{\boldsymbol{\Sigma}}_x)_{prev} - \widehat{\boldsymbol{\Sigma}}_x\|_2/\|(\widehat{\boldsymbol{\Sigma}}_x)_{prev}\|_2 + \|(\widehat{\mathbf{D}})_{prev} - \widehat{\mathbf{D}}\|_2/\|(\widehat{\mathbf{D}})_{prev}\|_2$

33: **end while**

34: Return $\widehat{\boldsymbol{\Sigma}}_x$ and $\widehat{\mathbf{D}}$

---

## Parameter choices for simulation study 2

In the second simulation study, we set $\alpha = 0.98$, $\mathbf{A} = 0.1\mathbf{I}$, $\boldsymbol{\mu}_x = -4.5\mathbf{1}$ and $\boldsymbol{\Sigma}_w = 10^{-4}\mathbf{I}$ ($\mathbf{I} \in \mathbb{R}^{30\times30}$ represents the identity matrix and $\mathbf{1} \in \mathbb{R}^{30}$ represents the vector of all ones) when generating the fluorescence traces $\{\mathbf{y}_{t,l}\}_{t,l=1}^{T,L}$, so that the SNR of the simulated data was in the same range as of real

calcium imaging observations. Furthermore, we simulated the spike trains based on a Poisson process (*Smith and Brown, 2003*) using the discrete time re-scaling procedure (*Brown et al., 2002*; *Smith and Brown, 2003*). Following the assumptions in *Brown et al., 2002*, we used an exponential link to simulate the observations:

$$n_{t,l}^{(j)} \sim Poisson\left(\lambda_{t,l}^{(j)}\right), \quad \lambda_{t,l}^{(j)} = \exp\left(x_{t,l}^{(j)}\right),$$

as opposed to the Bernoulli-logistic assumption in our recognition model. Then, we estimated the noise covariance $\widehat{\boldsymbol{\Sigma}}_x$ using the Algorithm 1, with a slight modification. Since there are no external stimuli, we set $\mathbf{s}_t = \mathbf{0}$ and $\mathbf{D} = \mathbf{0}$. Accordingly, in Algorithm 1, we initialized $\widehat{\mathbf{D}} = \mathbf{0}$ and did not perform the update on $\widehat{\mathbf{D}}$ in the subsequent iterations.

## Parameter choices for real data study 1

The dataset consisted of recordings from 371 excitatory neurons, from which we selected $N = 16$ responsive neurons for the analysis. Each trial consisted of $T = 3600$ time frames (the sampling frequency was 30 Hz, and each trial had a duration of 120 s), with the presentation of a random sequence of four tones ($P = 4$). The spiking events were very sparse and infrequent, and hence this dataset fits our model with at most one spiking event in a time frame.

We considered $R = 25$ ($M = 100$) time lags in this analysis and further examined the effect of varying $R$ in *Figure 4—figure supplement 1*. We set $\alpha = 0.95$ and $\mathbf{A} = \mathbf{I}$ ($\mathbf{I} \in \mathbb{R}^{16 \times 16}$ represents the identity matrix).

## Parameter choices for real data study 2

Each trial consisted of $T = 765$ frames (25.5 s) at a sampling frequency of 30 Hz. The A1 neurons studied here had low response rates (in both time and space), with only $\sim 10$ neurons exhibiting spiking activity in at least half of the trials. Thus, we selected $N = 10$ neurons and $L = 10$ trials for the analysis, and chose $R = 25$ lags of the stimulus ($M = 25$) in the model for the stimulus-driven condition. We set $\alpha = 0.95$ and $\mathbf{A} = 0.75\mathbf{I}$ ($\mathbf{I} \in \mathbb{R}^{10 \times 10}$ represents the identity matrix).

## Parameter choices for real data study 3

Each experiment consisted of $L = 5$ trials of $P = 9$ different tone frequencies repeated at four different amplitude levels, resulting in each concatenated trial being $\sim 180$ s long (see *Bowen et al., 2020* for more details). We set the number of stimulus time lags considered to be $R = 25$ ($M = 225$). For each layer, we analyzed fluorescence observations from six experiments. In each experiment, we selected the most responsive $N \sim 20$ neurons for the subsequent analysis. We set $\alpha = 0.95$ and $\mathbf{A} = \mathbf{I}$.

# Performance evaluation

## Simulation studies

Since the ground truth is known in simulations, we directly compared the performance of each signal and noise correlation estimate with the ground truth signal and noise correlations, respectively. Suppose the ground truth correlations are given by the matrix $\mathbf{X}$ and the estimated correlations are given by the matrix $\widehat{\mathbf{X}}$. To quantify the similarity between $\mathbf{X}$ and $\widehat{\mathbf{X}}$, we defined the following two metrics:

Normalized Mean Squared Error (NMSE): The NMSE computes the mean squared error of $\widehat{\mathbf{X}}$ with respect to $\mathbf{X}$ using the Frobenius Norm:

$$\text{NMSE} := \frac{\|\mathbf{X} - \widehat{\mathbf{X}}\|_F^2}{\|\mathbf{X}\|_F^2}.$$

Ratio between out-of-network power and in-network power (leakage): First, we identified the in-network and out-of-network components from the ground truth correlation matrix $\mathbf{X}$. Suppose that if the true correlation between the $i^{th}$ neuron and the $j^{th}$ neuron is non-zero, then $\left|(\mathbf{X})_{i,j}\right| > \delta_x$, for some $\delta_x > 0$. Thus, we formed a matrix $\mathbf{X}^{in}$ that masks the in-network components, by setting $(\mathbf{X}^{in})_{i,j} = 1$ if

$\left|(\mathbf{X})_{i,j}\right| > \delta_x$ and $(\mathbf{X}^{in})_{i,j} = 0$ if $\left|(\mathbf{X})_{i,j}\right| \leq \delta_x$. Likewise, we also formed a matrix $\mathbf{X}^{out}$ that masks the out-of-network components, by setting $(\mathbf{X}^{out})_{i,j} = 1$ if $\left|(\mathbf{X})_{i,j}\right| \leq \delta_x$ and $(\mathbf{X}^{out})_{i,j} = 0$ if $\left|(\mathbf{X})_{i,j}\right| > \delta_x$. Then, using these two matrices we quantified the leakage effect of $\widehat{\mathbf{X}}$ comparative to $\mathbf{X}$ by:

$$leakage := \frac{\|\widehat{\mathbf{X}} \cdot \mathbf{X}^{out}\|_F^2}{\|\widehat{\mathbf{X}} \cdot \mathbf{X}^{in}\|_F^2},$$

where $(\cdot)$ denotes element-wise multiplication.

## Real data studies

To quantify the similarity and dissimilarity between signal and noise correlation estimates, we used a statistic based on the Tanimoto similarity metric (*Lipkus, 1999*), denoted by $T_s(\mathbf{X}, \mathbf{Y})$ for two matrices $\mathbf{X}$ and $\mathbf{Y}$. For two vectors $\mathbf{a}$ and $\mathbf{b}$ with *non-negative* entries, the Tanimoto coefficient (*Lipkus, 1999*) is defined as:

$$T(\mathbf{a}, \mathbf{b}) := \frac{\mathbf{a}^T \mathbf{b}}{\mathbf{a}^T \mathbf{a} + \mathbf{b}^T \mathbf{b} - \mathbf{a}^T \mathbf{b}}.$$

The Tanimoto similarly metric between two matrices can be defined in a similar manner, by vectorizing the matrices. Thus, we formulated a similarity metric between two correlation matrices $\mathbf{X}$ and $\mathbf{Y}$ as follows. Let $\mathbf{X}^+ := \max\{\mathbf{X}, 0\mathbf{I}\}$ and $\mathbf{X}^- := \max\{-\mathbf{X}, 0\mathbf{I}\}$, with the $\max\{\cdot, \cdot\}$ operator interpreted element-wise. Note that $\mathbf{X} = \mathbf{X}^+ - \mathbf{X}^-$, and $\mathbf{X}^+, \mathbf{X}^-$ have non-negative entries. We then defined the similarity matrix by combining those of the positive and negative parts as follows:

$$T_s(\mathbf{X}, \mathbf{Y}) := \varepsilon\, T(\mathbf{X}^+, \mathbf{Y}^+) + (1 - \varepsilon)\, T(\mathbf{X}^-, \mathbf{Y}^-)$$

where $\varepsilon \in [0, 1]$ denotes the percentage of positive entries in $\mathbf{X}$ and $\mathbf{Y}$. As a measure of dissimilarity, we used $T_d(\mathbf{X}, \mathbf{Y}) := 1 - T_s(\mathbf{X}, \mathbf{Y})$. The values of $T_d(\widehat{\mathbf{S}}, \widehat{\mathbf{N}})$ in **Table 1** and $T_s(\widehat{\mathbf{N}}_{spon}, \widehat{\mathbf{N}}_{stim})$ and $T_d(\widehat{\mathbf{S}}_{stim}, \widehat{\mathbf{N}}_{stim})$ reported in **Table 2** were obtained based on the foregoing definitions.

To further assess the statistical significance of these results, we performed following randomized tests. To test the significance of $T_s(\widehat{\mathbf{N}}_{spon}, \widehat{\mathbf{N}}_{stim})$, for each comparison and each algorithm, we fixed the first matrix (i.e. $\widehat{\mathbf{N}}_{spon}$) and randomly shuffled the entries of the second one (i.e. $\widehat{\mathbf{N}}_{stim}$) while respecting symmetry. We repeated this procedure for 10000 trials, to derive the null distributions that represented the probabilities of chance occurrence of similarities between two random groups of neurons.

To test the significance of $T_d(\widehat{\mathbf{S}}, \widehat{\mathbf{N}})$ and $T_d(\widehat{\mathbf{S}}_{stim}, \widehat{\mathbf{N}}_{stim})$, for each comparison and each algorithm, again we fixed the first matrix (i.e. signal correlations). Then, we formed the elements of the second matrix (akin to noise correlations) as follows. For each element of the second matrix, we assigned either the same element as the signal correlations (in order to model the leakage effect) or a random noise (with same variance as the elements in the noise correlation matrix) with equal probability. As before, we repeated this procedure for 10,000 trials, to derive the null distributions that represent the probabilities of chance occurrence of dissimilarities between two matrices that have some leakage between them.

## Hyper-parameter tuning

The hyper-parameters that directly affect the proposed estimation are the inverse Wishart prior hyper-parameters: $\psi_x$ and $\rho_x$. Given that $\rho_x$ appears in the form of $\gamma_x := TL + \rho_x$, we will consider $\psi_x$ and $\gamma_x$ as the main hyper-parameters for simplicity. Here, we propose a criterion for choosing these two hyper-parameters in a data-driven fashion, which will then be used to construct the estimates of the noise covariance matrix $\widehat{\boldsymbol{\Sigma}}_x$ and weight matrix $\widehat{\mathbf{D}}$. Due to the hierarchy of hidden layers in our model, an empirical Bayes approach for hyper-parameter selection using a likelihood-based performance metric is not straightforward. Hence, we propose an alternative empirical method for hyper-parameter selection as follows.

For a given choice of $\boldsymbol{\psi}_x$ and $\gamma_x$, we estimate $\widehat{\boldsymbol{\Sigma}}_x$ and $\widehat{\mathbf{D}}$ following the proposed method. Then, based on the generative model in Proposed forward model, and using the estimated values of $\widehat{\boldsymbol{\Sigma}}_x$ and $\widehat{\mathbf{D}}$, we sample an ensemble of simulated fluorescence traces $\widehat{\mathbf{y}} = \{\widehat{\mathbf{y}}_t^{(l)}\}_{t,l=1}^{T,L}$, and compute the metric $d(\boldsymbol{\psi}_x, \gamma_x)$:

$$d(\boldsymbol{\psi}_x, \gamma_x) := D_{frob}(cov(\widehat{\mathbf{y}}, \widehat{\mathbf{y}}), cov(\mathbf{y}, \mathbf{y})),$$

where $cov(\cdot, \cdot)$ denotes the empirical covariance and $D_{frob}(\mathbf{X}, \mathbf{Y}) := \|\mathbf{X} - \mathbf{Y}\|_F^2$. Note that $D_{frob}(\mathbf{X}, \mathbf{Y})$ is strictly convex in $\mathbf{X}$. Thus, minimizing $D_{frob}(\mathbf{X}, \mathbf{Y})$ over $\mathbf{X}$ for a given $\mathbf{Y}$ has a unique solution. Accordingly, we observe that $d(\boldsymbol{\psi}_x, \gamma_x)$ is minimized when $cov(\widehat{\mathbf{y}}, \widehat{\mathbf{y}})$ is nearest to $cov(\mathbf{y}, \mathbf{y})$. Therefore, the corresponding estimates $\widehat{\boldsymbol{\Sigma}}_x$ and $\widehat{\mathbf{D}}$ that generated $\widehat{\mathbf{y}}$, best match the second-order statistics of $\mathbf{y}$ that was generated by the true parameters $\boldsymbol{\Sigma}_x$ and $\mathbf{D}$.

The typically low spiking rate of sensory neurons observed in practice may render the estimation problem ill-posed. It is thus important to have an accurate choice of the scale matrix $\boldsymbol{\psi}_x$ in the prior distribution. However, an exhaustive search for optimal tuning of $\boldsymbol{\psi}_x$ is not computationally feasible, given that it has $N(N+1)/2$ free variables. Thus, the main challenge here is finding the optimal choice of the scale matrix $\boldsymbol{\psi}_{x,opt}$.

To address this challenge, we propose the following method. First, we fix $\boldsymbol{\psi}_{x,init} = \tau \mathbf{I}$, where $\tau$ is a scalar and $\mathbf{I} \in \mathbb{R}^{N \times N}$ is the identity matrix. Next, given $\boldsymbol{\psi}_{x,init}$ we find the optimal choice of $\gamma_x$ as:

$$\gamma_{x,init} = \underset{\gamma_x \in \mathcal{S}_\gamma}{argmin}\, d(\boldsymbol{\psi}_{x,init}, \gamma_x),$$

where $\mathcal{S}_\gamma$ is a finite set of candidate solutions for $\gamma_x > N - 1$. Let $\widehat{\boldsymbol{\Sigma}}_{x,init}$ denote the noise covariance estimate corresponding to hyper-parameters $(\boldsymbol{\psi}_{x,init}, \gamma_{x,init})$. We will next use $\widehat{\boldsymbol{\Sigma}}_{x,init}$ to find a suitable choice of $\boldsymbol{\psi}_x$. To this end, we first fix $\gamma_{x,opt} := TL + \widetilde{\rho}_x$, for some $N - 1 < \widetilde{\rho}_x \ll TL$. Note that by choosing $\widetilde{\rho}_x$ to be much smaller than $TL$, the final estimates become less sensitive to the choice of $\gamma_x$. Then, we construct a candidate set $\mathcal{S}_\psi$ for $\boldsymbol{\psi}_{x,opt}$ by scaling $\widehat{\boldsymbol{\Sigma}}_{x,init}$ with a finite set of scalars $\eta \in \mathbb{R}^+$, i. e. $\mathcal{S}_\psi := \left\{ \eta \widehat{\boldsymbol{\Sigma}}_{x,init}, \eta \in \mathbb{R}^+ \right\}$. To select $\boldsymbol{\psi}_{x,opt}$, we match it with the choice of $\gamma_{x,opt}$ by solving:

$$\boldsymbol{\psi}_{x,opt} = \underset{\boldsymbol{\psi}_x \in \mathcal{S}_\psi}{argmin}\, d(\boldsymbol{\psi}_x, \gamma_{x,opt}).$$

Finally, we use these hyper-parameters $(\boldsymbol{\psi}_{x,opt}, \gamma_{x,opt})$ to obtain the estimators $\widehat{\boldsymbol{\Sigma}}_x$ and $\widehat{\mathbf{D}}$ as the output of the algorithm.

## Experimental procedures

All procedures were approved by the University of Maryland Institutional Animal Care and Use Committee. Imaging experiments were performed on a P60 (for real data study 1) and P83 (for real data study 2) female F1 offspring of the CBA/CaJ strain (The Jackson Laboratory; stock #000654) crossed with transgenic C57BL/6J-Tg(Thy1-GCaMP6s)GP4.3Dkim/J mice (The Jackson Laboratory; stock #024275) (CBAxThy1), and F1 (CBAxC57). The third real data study was performed on data from P66-P93 and P166-P178 mice (see *Bowen et al., 2020* for more details). We used the F1 generation of the crossed mice because they have good hearing into adulthood (*Frisina et al., 2011*).

We performed cranial window implantation and two-photon imaging as previously described in *Francis et al., 2018*; *Liu et al., 2019*; *Bowen et al., 2019*. Briefly, we implanted a cranial window of 3 mm in diameter over the left auditory cortex. We used a scanning microscope (Bergamo II series, B248, Thorlabs) coupled to Insight X3 laser (Spectra-physics) (study 1) or pulsed femtosecond Ti:Sapphire two-photon laser with dispersion compensation (Vision S, Coherent) (studies 2 and 3) to image GCaMP6s fluorescence from individual neurons in awake head-fixed mice with an excitation wavelengths of $\lambda = 920$ nm and $\lambda = 940$ nm, respectively. The microscope was controlled by ThorImageLS software. The size of the field of view was $370 \times 370\, \mu m$. Imaging frames of $512 \times 512$ pixels (pixel size $0.72\, \mu m$) were acquired at 30 Hz by bidirectional scanning of an 8 kHz resonant scanner. The imaging depth was around $200\, \mu m$ below pia.

## Data pre-processing

A circular ROI was manually drawn over each cell body to extract raw fluorescence traces from individual cells. Neuropil contamination subtraction and baseline correction were performed on the raw fluorescence traces of each cell (*Francis et al., 2018*; *Liu et al., 2019*; *Bowen et al., 2020*) according to $\frac{F_{cell} - \alpha_n F_{neuropil} - baseline}{baseline}$, where $\alpha_n$ was set to 0.7 in real data study 1 (*Francis et al., 2018*), 0.8 in real data study 2 (*Liu et al., 2019*) and 0.9 in real data study 3 (*Bowen et al., 2020*). The two-photon observations $\{\mathbf{y}_{t,l}\}_{t,l=1}^{T,L}$ used in our analyses are the output of this pre-processing step.

## Stimuli for real data study 1

During imaging experiments, we presented four tones (4, 8, 16, and 32 kHz) at 70 dB SPL. The tones were 2 s in duration with an inter-trial silence of 4 s. For the sequence of tones, we first generated a randomized sequence that consisted of five repeats for each tone (20 tones in total) and then the same sequence was repeated for 10 trials.

## Stimuli for real data study 2

During imaging experiments, we presented a 75 dB SPL 100 ms broadband noise (4–48 kHz) as the auditory stimulus. Each trial was 5.1 s long (1 s pre-stimulus silence + 0.1 s stimulus + 3 s post-stimulus silence), and the inter-trial duration was 3 s. Spontaneous neuronal activity was collected from activity during randomly interleaved no-stimuli trials of the same duration, and these trials had complete silence throughout the trial duration (5.1 s long).

Then, we extracted 50 such trials from each type, and formed 10 ($L = 10$) trials each of 25.5 s duration ($T = 765$ frames) for the subsequent analysis, by concatenating five 5.1 s trials. This final step was performed to increase the effective trial duration.

## Stimuli for real data study 3

During imaging experiments, sounds were played at four sound levels (20, 40, 60, and 80 dB SPL). Auditory stimuli consisted of sinusoidal amplitude-modulated (SAM) tones (20 Hz modulation, cosine phase), ranging from 3 to 48 kHz. The frequency resolution was two tones/octave (0.5 octave spacing) and each of these tonal stimuli was 1 s long, repeated five times with a 4−6 s inter-stimulus interval (see *Bowen et al., 2020* for details).

## Acknowledgements

The authors thank Daniel E Winkowski for collecting the data in *Bowen et al., 2019* that was also used in this work.

## Additional information

### Funding

| Funder | Grant reference number | Author |
|---|---|---|
| National Science Foundation | 1807216 | Behtash Babadi |
| National Science Foundation | 2032649 | Behtash Babadi |
| National Institutes of Health | 1U19NS107464 | Patrick O Kanold<br>Behtash Babadi |

The funders had no role in study design, data collection and interpretation, or the decision to submit the work for publication.

### Author contributions

Anuththara Rupasinghe, Ji Liu, Conceptualization, Data curation, Software, Formal analysis, Validation, Investigation, Visualization, Methodology, Writing - original draft, Writing - review and editing; Nikolas Francis, Conceptualization, Resources, Data curation, Formal analysis, Supervision, Funding acquisition, Validation, Investigation, Visualization, Methodology, Writing - original draft, Project

administration, Writing - review and editing; Zac Bowen, Data curation, Investigation, Writing - review and editing; Patrick O Kanold, Conceptualization, Resources, Supervision, Funding acquisition, Validation, Investigation, Project administration, Writing - review and editing; Behtash Babadi, Conceptualization, Resources, Formal analysis, Supervision, Funding acquisition, Validation, Investigation, Visualization, Methodology, Writing - original draft, Project administration, Writing - review and editing

### Author ORCIDs

Anuththara Rupasinghe (iD) http://orcid.org/0000-0003-2143-8709
Behtash Babadi (iD) https://orcid.org/0000-0002-9856-006X

### Ethics

Animal experimentation: All procedures, under Kanold lab protocol R-JAN-19-06, conformed to the guidelines of the University of Maryland Institutional Animal Care and Use Committee and the Guide for the Care and Use of Laboratory Animals of the National Institutes of Health.

### Decision letter and Author response

Decision letter https://doi.org/10.7554/eLife.68046.sa1
Author response https://doi.org/10.7554/eLife.68046.sa2

## Additional files

### Supplementary files

• Transparent reporting form

### Data availability

A MATLAB implementation of the proposed method has been archived in Github at https://github.com/Anuththara-Rupasinghe/Signal-Noise-Correlation (copy archived at https://archive.softwareheritage.org/swh:1:rev:7397cc8d751a128f41df81f8af160014b22974d6). The data used in this work have been deposited in the Digital Repository at the University of Maryland at http://hdl.handle.net/1903/26917.

The following dataset was generated:

| Author(s) | Year | Dataset title | Dataset URL | Database and Identifier |
|---|---|---|---|---|
| Rupasinghe A, Francis N, Liu J, Bowen Z, Kanold PO, Babadi B | 2021 | Experimental Data from 'Direct Extraction of Signal and Noise Correlations from Two-Photon Calcium Imaging of Ensemble Neuronal Activity' | http://hdl.handle.net/1903/26917 | Digital Repository at the University of Maryland, 1903/26917 |

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

## Appendix 1

### Relationship to existing definitions of Signal and Noise correlations

Recall that the conventional definitions of signal and noise covariance of spiking activity between the $i^{th}$ and $j^{th}$ neuron are (*Lyamzin et al., 2015*):

$$(\Sigma_s^{con})_{i,j} = cov\left(\frac{1}{L}\sum_l n_{t,l}^{(i)}, \frac{1}{L}\sum_l n_{t,l}^{(j)}\right),$$

$$(\Sigma_x^{con})_{i,j} = \frac{1}{L}\sum_l cov\left(n_{t,l}^{(i)} - \frac{1}{L}\sum_{l'} n_{t,l'}^{(i)}, n_{t,l}^{(j)} - \frac{1}{L}\sum_{l'} n_{t,l'}^{(j)}\right),$$

where $cov(\mathbf{u}_t, \mathbf{v}_t) := \frac{1}{T}\sum_{t=1}^{T}\left(\mathbf{u}_t - \frac{1}{T}\sum_{t'=1}^{T}\mathbf{u}_{t'}\right)\left(\mathbf{v}_t - \frac{1}{T}\sum_{t'=1}^{T}\mathbf{v}_{t'}\right)^{\top}$, is the empirical covariance. The correlations, are then derived by the standard normalization:

$$(\mathbf{S}^{con})_{i,j} := \frac{(\mathbf{\Sigma}_s^{con})_{i,j}}{\sqrt{(\mathbf{\Sigma}_s^{con})_{i,i}\cdot(\mathbf{\Sigma}_s^{con})_{j,j}}}, \quad (\mathbf{N}^{con})_{i,j} := \frac{(\mathbf{\Sigma}_x^{con})_{i,j}}{\sqrt{(\mathbf{\Sigma}_x^{con})_{i,i}\cdot(\mathbf{\Sigma}_x^{con})_{j,j}}}, \quad \forall i,j = 1,2,\cdots,N. \tag{9}$$

Suppose that the spiking events follow the forward model:

$$n_{t,l}^{(j)} \sim Bernoulli\left(\lambda_{t,l}^{(j)}\right), \quad \lambda_{t,l}^{(j)} = \phi\left(x_{t,l}^{(j)}, \mathbf{d}_j^{\top}\mathbf{s}_t\right),$$

where $\phi : \mathbb{R}^2 \to [0,1]$ is a differentiable non-linear mapping. We assume $\mathbf{x}_{t,l}$ and $\mathbf{s}_t$ to be independent. Without loss of generality, let $\mathbb{E}[\mathbf{s}_t] = \mathbf{0}$ and $\mathbb{E}[\mathbf{x}_{t,l}] = \boldsymbol{\mu}_x$. Further, we define the notation $X_t \approx Y_t$ to denote almost sure equivalence, that is $X_t \xrightarrow{a.s.} Z$ and $Y_t \xrightarrow{a.s.} Z$ for some random variable $Z$.

First, let us consider $(\mathbf{S}^{con})_{i,j}$. Noting that $\mathbb{E}\left[n_{t,l}^{(j)}\right] = \mathbb{E}\left[\lambda_{t,l}^{(j)}\right]$ and $\mathbb{E}\left[n_{t,l}^{(i)}n_{t,l}^{(j)}\right] = \mathbb{E}\left[\lambda_{t,l}^{(i)}\lambda_{t,l}^{(j)}\right]$, we conclude as $T \to \infty$:

$$(\Sigma_s^{con})_{i,j} \approx cov\left(\frac{1}{L}\sum_l \lambda_{t,l}^{(i)}, \frac{1}{L}\sum_l \lambda_{t,l}^{(j)}\right),$$

from the law of large numbers. Then, if we consider the Taylor series expansion of $\phi\left(x_{t,l}^{(j)}, \mathbf{d}_j^{\top}\mathbf{s}_t\right)$ around the mean $(\mu_x^{(j)}, 0)$, we get:

$$(\Sigma_s^{con})_{i,j} \approx cov\Bigg(\phi\left(\mu_x^{(i)},0\right) + \frac{1}{L}\sum_l\left(x_{t,l}^{(i)} - \mu_x^{(i)}\right)\phi_{\left(x_{t,l}^{(i)}\right)}\left(\mu_x^{(i)},0\right) + (\mathbf{d}_i^{\top}\mathbf{s}_t)\phi_{\left(\mathbf{d}_i^{\top}\mathbf{s}_t\right)}\left(\mu_x^{(i)},0\right) + \epsilon_{t,l}^{(i)},$$
$$\phi\left(\mu_x^{(j)},0\right) + \frac{1}{L}\sum_l\left(x_{t,l}^{(j)} - \mu_x^{(j)}\right)\phi_{\left(x_{t,l}^{(j)}\right)}\left(\mu_x^{(j)},0\right) + (\mathbf{d}_j^{\top}\mathbf{s}_t)\phi_{\left(\mathbf{d}_j^{\top}\mathbf{s}_t\right)}\left(\mu_x^{(j)},0\right) + \epsilon_{t,l}^{(j)}\Bigg),$$

where $\epsilon_{t,l}^{(i)}$ and $\epsilon_{t,l}^{(j)}$ represent the higher order terms. Then, as $L \to \infty$, we get:

$$(\Sigma_s^{con})_{i,j} \approx cov\left((\mathbf{d}_i^{\top}\mathbf{s}_t)\phi_{\left(\mathbf{d}_i^{\top}\mathbf{s}_t\right)}\left(\mu_x^{(i)},0\right), (\mathbf{d}_j^{\top}\mathbf{s}_t)\phi_{\left(\mathbf{d}_j^{\top}\mathbf{s}_t\right)}\left(\mu_x^{(j)},0\right)\right) + \epsilon_{t,l},$$

since $\lim_{L\to\infty}\frac{1}{L}\sum_{l=1}^{L}\left(x_{t,l}^{(j)}\right) = \mu_x^{(j)}$ by the Law of Large numbers. Thus, we see that:

$$(\Sigma_s^{con})_{i,j} \approx C_S \mathbf{d}_i^{\top} cov(\mathbf{s}_t, \mathbf{s}_t)\mathbf{d}_j + \epsilon_{t,l}$$
$$= C_S(\Sigma_s)_{i,j} + \epsilon_{t,l},$$

where $C_S$ is a constant and $\epsilon_{t,l}$ is typically small if the latent process $\mathbf{x}_{t,l}$ and the stimulus $\mathbf{s}_t$ are concentrated around their means. Then, the signal correlations are obtained by normalization of the signal covariance as in *Equation 9*, through which the scaling factor $C_S$ cancels and we get:

$$(\mathbf{S}^{con})_{i,j} \approx (\mathbf{S})_{i,j}.$$

Thus, as $T, L \to \infty$, we see that $\mathbf{S}$ is indeed the signal correlation matrix that is aimed to be approximated by the conventional definitions.

Next, let us consider $(\mathbf{N}^{con})_{i,j}$. Similar to foregoing analysis of the signal covariance, as $T \to \infty$ we get:

$$(\Sigma_x^{con})_{i,j} \approx \frac{1}{L} \sum_l cov\left( \lambda_{t,l}^{(i)} - \frac{1}{L} \sum_{l'} \lambda_{t,l'}^{(i)}, \lambda_{t,l}^{(j)} - \frac{1}{L} \sum_{l'} \lambda_{t,l'}^{(j)} \right).$$

Then, from a Taylor series expansion, we get:

$$(\Sigma_x^{con})_{i,j} \approx \frac{1}{L} \sum_l cov\left( x_{t,l}^{(i)} \phi_{(x_{t,l}^{(i)})}\left(\mu_x^{(i)}, 0\right) - \frac{1}{L} \sum_{l'} x_{t,l'}^{(i)} \phi_{(x_{t,l'}^{(i)})}\left(\mu_x^{(i)}, 0\right) + \xi_{t,l}^{(i)}, \right.$$
$$\left. x_{t,l}^{(j)} \phi_{(x_{t,l}^{(j)})}\left(\mu_x^{(j)}, 0\right) - \frac{1}{L} \sum_{l'} x_{t,l'}^{(j)} \phi_{(x_{t,l'}^{(j)})}\left(\mu_x^{(j)}, 0\right) + \xi_{t,l}^{(j)} \right),$$

where $\xi_{t,l}^{(i)}$ and $\xi_{t,l}^{(j)}$ represent the higher order terms. Then, as $L \to \infty$:

$$(\Sigma_x^{con})_{i,j} \approx \frac{1}{L} \sum_l cov\left( \left( x_{t,l}^{(i)} - \mu_x^{(i)} \right) \phi_{(x_{t,l}^{(i)})}\left(\mu_x^{(i)}, 0\right), \left( x_{t,l}^{(j)} - \mu_x^{(j)} \right) \phi_{(x_{t,l}^{(j)})}\left(\mu_x^{(j)}, 0\right) \right) + \xi_{t,l},$$

from the law of large numbers. Accordingly, we see that:

$$\begin{aligned} (\Sigma_x^{con})_{i,j} &\approx C_N \frac{1}{L} \sum_l cov\left( x_{t,l}^{(i)} - \mu_x^{(i)}, x_{t,l}^{(j)} - \mu_x^{(j)} \right) + \xi_{t,l} \\ &= C_N \left( \Sigma_x \right)_{i,j} + \xi_{t,l}, \end{aligned}$$

where $C_N$ is a constant and $\xi_{t,l}$ is typically small if the latent process $\mathbf{x}_{t,l}$ and the stimulus $\mathbf{s}_t$ are concentrated around their means. Then, the noise correlations are derived by normalization of the noise covariance given in *Equation 9*. This cancels out the scaling factor $C_N$, and we get:

$$(\mathbf{N}^{con})_{i,j} \approx (\mathbf{N})_{i,j}.$$

Thus, we similarly conclude that as $T, L \to \infty$, the conventional definition of noise correlation $\mathbf{N}^{con}$ indeed aims to approximate $\mathbf{N}$.

As a numerical illustration, we demonstrated in *Figure 2—figure supplement 2* that the conventional definitions of the correlations indeed approximate our proposed definitions, but require much larger number of trials to be accurate. More specifically, in order to achieve comparable performance to our method using $L = 20$ trials, the conventional correlation estimates require $L = 1000$ trials.

## Appendix 2

### Proof of Theorem 1

In what follows, we present a comprehensive proof of Theorem 1. Recall the following key assumptions:

Assumption (1). We assume a scalar time-varying external stimulus (i.e. $\mathbf{s}_t = s_t$, and hence $\mathbf{d}_j = d_j$, $\mathbf{d} = [d_1, d_2, \cdots, d_N]^\top$ ). Furthermore, we set the observation noise covariance to be $\boldsymbol{\Sigma}_w = \sigma_w^2 \mathbf{I}$, for notational convenience.

Assumption (2). We derive the performance bounds in the regime where $T$ and $L$ are large, and thus do not impose any prior distribution on the correlations (i.e. $p_{pr}(\boldsymbol{\Sigma}_x) \propto 1$), which are otherwise needed to mitigate overfitting (see Preliminary assumptions).

Assumption (3). We assume the latent trial-dependent process and stimulus to be slowly varying signals, and thus adopt a piece-wise constant model in which these processes are constant within consecutive windows of length $W$ (i.e., $\mathbf{x}_{t,l} = \mathbf{x}_{W_k,l}$ and $s_t = s_{W_k}$, for $(k-1)W + 1 \leq t < kW$ and $k = 1, \cdots, K$ with $W_k = (k-1)W + 1$ and $KW = T$) for our theoretical analysis, as is usually done in spike count calculations for conventional noise correlation estimates.

Proof of Theorem 1. First, recall the proposed forward model (see Proposed forward model) under Assumption (1 – 3):

$$\begin{aligned}
\mathbf{y}_{t,l} &= \mathbf{A}\mathbf{z}_{t,l} + \mathbf{w}_{t,l}, \\
\mathbf{z}_{t,l} &= \alpha\,\mathbf{z}_{t-1,l} + \mathbf{n}_{t,l}, \\
n_{t,l}^{(j)} &\sim Bernoulli\left(\phi\left(x_{W_k,l}^{(j)}\right)\right), \\
\mathbf{x}_{W_k,l} &\sim \mathcal{N}\left(\boldsymbol{\mu}_x + s_{W_k}\mathbf{d}, \boldsymbol{\Sigma}_x\right),
\end{aligned}$$

where $\phi(\cdot) := \frac{\exp(\cdot)}{1+\exp(\cdot)}$, is the logistic function. Note that we have re-defined the latent process $\mathbf{x}_{t,l}$ by absorbing the stimulus activity $s_t \mathbf{d}$ to the mean of $\mathbf{x}_{t,l}$ for notational convenience, without loss of generality. Hereafter, we also assume that $\mathbf{A} = \mathbf{I}$ without loss of generality. For a truncation level $B$ (to be specified later), consider the event

$$\mathscr{A}_W := \left\{ \left|x_{W_k,l}^{(j)}\right| \leq B \text{ and } \frac{1}{2(1+\exp(B))} \leq \bar{n}_{W_k,l}^{(j)} \leq 1 - \frac{1}{2(1+\exp(B))} \text{ for } j=1,\cdots,N, k=1,\cdots,K \text{ and } l=1,\cdots,L \right\},$$

such that $\bar{\mathbf{n}}_{W_k,l} = \left[\bar{n}_{W_k,l}^{(1)}, \bar{n}_{W_k,l}^{(2)}, \cdots, \bar{n}_{W_k,l}^{(N)}\right]^\top := \frac{1}{W}\sum_{w=1}^{W} \mathbf{n}_{(k-1)W+w,l}$.

First, we derive convenient forms of the maximum likelihood estimators via the Laplace's approximations and asymptotic expansions (**Wong, 2001**) through the following lemma:

### Lemma 1

Conditioned on event $\mathscr{A}_W$, the maximum likelihood estimators of the stimulus kernel of the $j^{\text{th}}$ neuron and the noise covariance between the $i^{\text{th}}$ and $j^{\text{th}}$ neurons take the forms:

$$\begin{aligned}
\widehat{d}_j &= \widetilde{d}_j\left(1 + \mathcal{O}(\sigma_w^2)\right)\left(1 + \mathcal{O}\left(\frac{1}{W}\right)\right) \quad \text{and} \\
(\widehat{\boldsymbol{\Sigma}}_x)_{i,j} &= (\widetilde{\boldsymbol{\Sigma}}_x)_{i,j}\left(1 + \mathcal{O}(\sigma_w^2)\right)\left(1 + \mathcal{O}\left(\frac{1}{W}\right)\right),
\end{aligned}$$

where

$$\begin{aligned}
\widetilde{d}_j &= \frac{1}{L\sum_{k=1}^{K} s_{W_k}^2} \sum_{k,l=1}^{K,L} s_{W_k}\left(\phi^{-1}\left(\widetilde{n}_{W_k,l}^{(j)}\right) - \mu_x^{(j)}\right) \quad \text{and} \\
(\widetilde{\boldsymbol{\Sigma}}_x)_{i,j} &= \frac{1}{KL}\sum_{k,l=1}^{K,L}\left(\phi^{-1}\left(\widetilde{n}_{W_k,l}^{(i)}\right) - \mu_x^{(i)} - s_{W_k}\widetilde{d}_i\right)\left(\phi^{-1}\left(\widetilde{n}_{W_k,l}^{(j)}\right) - \mu_x^{(j)} - s_{W_k}\widetilde{d}_j\right),
\end{aligned}$$

with

$$\widetilde{\mathbf{n}}_{W_k,l} = \left[\widetilde{n}^{(1)}_{W_k,l}, \widetilde{n}^{(2)}_{W_k,l}, \cdots, \widetilde{n}^{(N)}_{W_k,l}\right]^{\top} := \frac{1}{W}\sum_{w=1}^{W}\left(\mathbf{y}_{(k-1)W+w,l} - \alpha\mathbf{y}_{(k-1)W+w-1,l}\right) \quad \text{and} \quad \phi^{-1}(z) := \ln(z/(1-z)).$$

Proof of Lemma 1.

First, maximizing the data likelihood, we derive the estimators:

$$\widehat{d}_j = \underset{d_j}{argmax} \ p(\mathbf{y}|\boldsymbol{\Sigma}_x,\mathbf{d}) = \frac{\int\left(\frac{1}{L\sum_{k=1}^{K}s^2_{W_k}}\sum_{k,l=1}^{K,L}s_{W_k}\left(x^{(j)}_{W_k,l}-\mu^{(j)}_x\right)\right)p(\mathbf{y}|\mathbf{n})p(\mathbf{n}|\mathbf{x})p(\mathbf{x}|\boldsymbol{\Sigma}_x,\mathbf{d})\,d\mathbf{n}d\mathbf{x}}{\int p(\mathbf{y}|\mathbf{n})p(\mathbf{n}|\mathbf{x})p(\mathbf{x}|\boldsymbol{\Sigma}_x,\mathbf{d})\,d\mathbf{n}d\mathbf{x}}, \tag{10}$$

and

$$(\widehat{\boldsymbol{\Sigma}}_x)_{i,j} = \underset{(\boldsymbol{\Sigma}_x)_{i,j}}{argmax} \ p(\mathbf{y}|\boldsymbol{\Sigma}_x,\mathbf{d})$$

$$= \frac{\int\left(\frac{1}{KL}\sum_{k,l=1}^{K,L}\left(x^{(i)}_{W_k,l}-\mu^{(i)}_x-s_{W_k}\widehat{d}_i\right)\left(x^{(j)}_{W_k,l}-\mu^{(j)}_x-s_{W_k}\widehat{d}_j\right)\right)p(\mathbf{y}|\mathbf{n})p(\mathbf{n}|\mathbf{x})p(\mathbf{x}|\boldsymbol{\Sigma}_x,\mathbf{d})\,d\mathbf{n}d\mathbf{x}}{\int p(\mathbf{y}|\mathbf{n})p(\mathbf{n}|\mathbf{x})p(\mathbf{x}|\boldsymbol{\Sigma}_x,\mathbf{d})\,d\mathbf{n}d\mathbf{x}}, \tag{11}$$

where $W_k = (k-1)W+1$. Then, we simplify these integrals based on the saddle point method of asymptotic expansions (*Wong, 2001*). To that end, first consider the numerator of *Equation 10* denoted by $I^{(1)}_{\mathrm{num}}$. First, we evaluate the integration in $I^{(1)}_{\mathrm{num}}$ with respect to the variable $\mathbf{n}$. To that end, note:

$$I^{(1)}_{\mathrm{num}} = \int h^{(1)}_{\mathrm{num}}(\mathbf{n})\exp\left(A_1 f_1(\mathbf{n})\right)d\mathbf{n},$$

where $h^{(1)}_{\mathrm{num}}(\mathbf{n}) = \frac{1}{\sqrt{(2\pi)^{TNL}\sigma^{2TNL}_w}}\int\left(\frac{1}{L\sum_{k=1}^{K}s^2_{W_k}}\sum_{k,l=1}^{K,L}s_{W_k}\left(x^{(j)}_{W_k,l}-\mu^{(j)}_x\right)\right)p(\mathbf{n}|\mathbf{x})p(\mathbf{x}|\boldsymbol{\Sigma}_x,\mathbf{d})\,d\mathbf{x},$ $A_1 = \frac{1}{\sigma^2_w}$, $f_1(\mathbf{n}) = -\frac{1}{2}\sum_{t,l,j}\left(y^{(j)}_{t,l}-\sum_{k=1}^{t}\alpha^{t-k}n^{(j)}_{t,l}\right)^2$ and $d\mathbf{n}$ is shorthand notation for the product measure of the discrete random vector $\mathbf{n}$. Observing that $\nabla f_1(\widehat{\mathbf{n}}) = \mathbf{0}$ for $\widehat{\mathbf{n}} := \left\{\widehat{n}_{t,l} = \mathbf{y}_{t,l} - \alpha\mathbf{y}_{t-1,l}\right\}^{T,L}_{t,l=1}$, using the method of asymptotic expansions, $I^{(1)}_{\mathrm{num}}$ can be evaluated as:

$$I^{(1)}_{\mathrm{num}} = h^{(1)}_{\mathrm{num}}(\widehat{\mathbf{n}}) \times \exp\left(A_1 f_1(\widehat{\mathbf{n}})\right)\sqrt{\frac{(2\pi)^{TLN}}{-A_1|H(f_1)|}}\left(1+\mathcal{O}\left(\frac{1}{A_1}\right)\right), \tag{12}$$

where the determinant of the Hessian matrix $|H(f_1)|$, is a negative function of $\alpha$. Note that the covariance of this Gaussian integral $\left(-(H(f_1))^{-1}\right)$ is a function of $\alpha \in (0,1)$, and hence is bounded. Thus, all higher order error terms in *Equation 12* are also bounded, as higher order moments of Gaussian distributions are functions of the covariance.

Next, we simplify the integral $h^{(1)}_{\mathrm{num}}(\widehat{\mathbf{n}})$ in *Equation 12* using a similar procedure. We have:

$$h^{(1)}_{\mathrm{num}}(\widehat{\mathbf{n}}) = \int r^{(1)}_{\mathrm{num}}(\mathbf{x})\exp\left(A_2 f_2(\mathbf{x})\right)d\mathbf{x}, \tag{13}$$

where $f_2(\mathbf{x}) = \sum_{k,l,j}\left(\widetilde{n}^{(j)}_{W_k,l}x^{(j)}_{W_k,l} - \log\left(1+\exp\left(x^{(j)}_{W_k,l}\right)\right)\right)$ with

$$\widetilde{\mathbf{n}}_{W_k,l} = \left[\widetilde{n}^{(1)}_{W_k,l}, \widetilde{n}^{(2)}_{W_k,l}, \cdots, \widetilde{n}^{(N)}_{W_k,l}\right]^{\top} := \frac{1}{W}\sum_{w=1}^{W}\widehat{\mathbf{n}}_{(k-1)W+w,l},$$

$$r^{(1)}_{\mathrm{num}}(\mathbf{x}) = \frac{1}{\sqrt{(2\pi)^{(W+1)KLN}\sigma^{2TNL}_w|\Sigma_x|^{KL}}}\exp\left(-\frac{1}{2}\sum_t\left(\mathbf{x}_{W_k,l}-\boldsymbol{\mu}_x-s_{W_k}\mathbf{d}\right)^{\top}\Sigma^{-1}_x\left(\mathbf{x}_{W_k,l}-\boldsymbol{\mu}_x-s_{W_k}\mathbf{d}\right)\right)$$
$$\times\left(\frac{1}{L\sum_{k=1}^{K}s^2_{W_k}}\sum_{k=1}^{K}s_{W_k,l}\left(x^{(j)}_{W_k,l}-\mu^{(j)}_x\right)\right),$$

and $A_2 = W$. Then, we note that the gradient of $f_2$, $\nabla f_2(\widehat{\mathbf{x}}) = \mathbf{0}$ for $\widehat{\mathbf{x}} := \left\{ \widehat{x}^{(j)}_{W_k,l} = \phi^{-1}\left(\widetilde{n}^{(j)}_{W_k,l}\right) \right\}^{K,L,N}_{k,l,j=1}$, where $\phi^{-1}(z) := logit(z) = \ln(z/(1-z))$. Accordingly, by re-applying the saddle point method of asymptotic expansions, we evaluate the integral in *Equation 13* as:

$$h^{(1)}_{\text{num}}(\widehat{\mathbf{n}}) = r^{(1)}_{\text{num}}(\widehat{\mathbf{x}}) \times \exp\left(A_2 f_2(\widehat{\mathbf{x}})\right) \sqrt{\frac{(2\pi)^{KLN}}{-A_2 |H(f_2(\widehat{\mathbf{x}}))|}} \left(1 + \mathcal{O}\left(\frac{1}{A_2}\right)\right),$$ (14)

where the determinant of the Hessian, $|H(f_2(\widehat{\mathbf{x}}))| = -\prod_{k,l,j} \widetilde{n}^{(j)}_{W_k,l}\left(1 - \widetilde{n}^{(j)}_{W_k,l}\right) < 0$ when conditioned on event $\mathscr{A}_W$. The higher order terms in *Equation 14* will be bounded if the covariance of the saddle point approximation $\left(-(H(f_2(\widehat{\mathbf{x}})))^{-1}\right)$ is bounded, which we ensure by conditioning on event $\mathscr{A}_W$. This completes the evaluation of $I^{(1)}_{\text{num}}$.

Following the same sequence of arguments, we evaluate the denominator of *Equation 10* denoted by $I^{(1)}_{\text{den}}$. Accordingly, we derive:

$$I^{(1)}_{\text{den}} = h^{(1)}_{\text{den}}(\widehat{\mathbf{n}}) \times \exp\left(A_1 f_1(\widehat{\mathbf{n}})\right) \sqrt{\frac{(2\pi)^{TLN}}{-A_1 |H(f_1)|}} \left(1 + \mathcal{O}\left(\frac{1}{A_1}\right)\right),$$

$$h^{(1)}_{\text{den}}(\widehat{\mathbf{n}}) = r^{(1)}_{\text{den}}(\widehat{\mathbf{x}}) \times \exp\left(A_2 f_2(\widehat{\mathbf{x}})\right) \sqrt{\frac{(2\pi)^{KLN}}{-A_2 |H(f_2(\widehat{\mathbf{x}}))|}} \left(1 + \mathcal{O}\left(\frac{1}{A_2}\right)\right),$$ (15)

where $r^{(1)}_{\text{den}}(\mathbf{x}) = \frac{1}{\sqrt{(2\pi)^{(W+1)KLN} \sigma^{2TNL}_w |\Sigma_x|^{KL}}} \exp\left(-\frac{1}{2}\sum_{k,l}\left(\mathbf{x}_{W_k,l} - \boldsymbol{\mu}_x - s_{W_k}\mathbf{d}\right)^\top \Sigma^{-1}_x\left(\mathbf{x}_{W_k,l} - \boldsymbol{\mu}_x - s_{W_k}\mathbf{d}\right)\right)$. Finally, by combining *Equation 12*, *Equation 14* and *Equation 15*, the maximum likelihood estimator in *Equation 10* takes the form:

$$\widehat{d}_j = \frac{I^{(1)}_{\text{num}}}{I^{(1)}_{\text{den}}} = \widetilde{d}_j \frac{\left(1 + \mathcal{O}\left(\frac{1}{A_1}\right)\right)\left(1 + \mathcal{O}\left(\frac{1}{A_2}\right)\right)}{\left(1 + \mathcal{O}\left(\frac{1}{A_1}\right)\right)\left(1 + \mathcal{O}\left(\frac{1}{A_2}\right)\right)} = \widetilde{d}_j\left(1 + \mathcal{O}(\sigma^2_w)\right)\left(1 + \mathcal{O}\left(\frac{1}{W}\right)\right).$$

Further, following the same sequence of reasoning, simplifying the numerator $(I^{(2)}_{\text{num}})$ and denominator $(I^{(2)}_{\text{den}})$ of *Equation 11* yields:

$$(\widehat{\boldsymbol{\Sigma}}_x)_{i,j} = \frac{I^{(2)}_{\text{num}}}{I^{(2)}_{\text{den}}} = (\widetilde{\boldsymbol{\Sigma}}_x)_{i,j} \frac{\left(1 + \mathcal{O}\left(\frac{1}{A_1}\right)\right)\left(1 + \mathcal{O}\left(\frac{1}{A_2}\right)\right)}{\left(1 + \mathcal{O}\left(\frac{1}{A_1}\right)\right)\left(1 + \mathcal{O}\left(\frac{1}{A_2}\right)\right)} = (\widetilde{\boldsymbol{\Sigma}}_x)_{i,j}\left(1 + \mathcal{O}(\sigma^2_w)\right)\left(1 + \mathcal{O}\left(\frac{1}{W}\right)\right).$$

This concludes the proof of Lemma 1. ∎

Given that $\phi^{-1}(z)$ is unbounded for $z = 0$ or $z = 1$, we consider another truncation: $\phi^{-1}_{B'}(z) := \min\{\max\{\phi^{-1}(z), -B'\}, B'\}$, where $B' = 2\log(2\exp(B) + 1)$. This choice of $B'$ guarantees that over $\mathscr{A}_W$, $\left|\phi^{-1}_{B'}\left(\bar{n}^{(j)}_{W_k,l}\right)\right| < B'$ for all $j = 1, \cdots, N$, $k = 1, \cdots, K$ and $l = 1, \cdots, L$: and thus $\phi^{-1}_{B'}\left(\bar{n}^{(j)}_{W_k,l}\right) = \phi^{-1}\left(\bar{n}^{(j)}_{W_k,l}\right)$ on $\mathscr{A}_W$.

From Lemma 1, the bias and variance of the maximum likelihood estimators, $\widehat{d}_j$ and $(\widehat{\boldsymbol{\Sigma}}_x)_{i,j}$ are upper-bounded, if those of $\widetilde{d}_j$ and $(\widetilde{\boldsymbol{\Sigma}}_x)_{i,j}$ are bounded:

$$\left|bias\left(\widehat{d}_j\right)\right| \leq \left|bias\left(\widetilde{d}_j\right)\right| + \zeta_j, \quad Var\left(\widehat{d}_j\right) \leq Var\left(\widetilde{d}_j\right) + \widetilde{\zeta}_j,$$ (16)

and

$$\left|bias\left((\widehat{\boldsymbol{\Sigma}}_x)_{i,j}\right)\right| \leq \left|bias\left((\widetilde{\boldsymbol{\Sigma}}_x)_{i,j}\right)\right| + \upsilon_{i,j}, \quad Var\left((\widehat{\boldsymbol{\Sigma}}_x)_{i,j}\right) \leq Var\left((\widetilde{\boldsymbol{\Sigma}}_x)_{i,j}\right) + \widetilde{\upsilon}_{i,j},$$ (17)

where $\zeta_j$, $\widetilde{\zeta}_j$, $\upsilon_{i,j}$ and $\widetilde{\upsilon}_{i,j}$ represent terms that are $\mathcal{O}(\sigma^2_w)$ or $\mathcal{O}(\frac{1}{W})$. Thus, we seek to derive the performance bounds of $\widetilde{d}_j$ and $(\widetilde{\boldsymbol{\Sigma}}_x)_{i,j}$.

## Bounding the bias of $\widehat{d}_j$

Let us first consider $\widetilde{d}_j$. Note that:

$$
\begin{aligned}
\left| bias\left(\widetilde{d}_j\right) \right| &:= \left| \mathbb{E}\left[\widetilde{d}_j\right] - d_j \right| \\
&\overset{(a)}{=} \left| \mathbb{E}\left[\widetilde{d}_j - (d_{\mathrm{Oracle}})_j\right] \right| \\
&\overset{(b)}{\leq} \frac{1}{L\sum_{k=1}^{K} s_{W_k}^2} \sum_{k,l=1}^{K,L} |s_{W_k}| \mathbb{E}\left[\left| \phi_{B'}^{-1}\left(\widetilde{n}_{W_k,l}^{(j)}\right) - x_{W_k,l}^{(j)} \right|\right],
\end{aligned}
$$

(18)

where $(a)$ holds since the Oracle estimator, $(d_{\mathrm{Oracle}})_j = \frac{1}{L\sum_{k=1}^{K} s_{W_k}^2} \sum_{k,l=1}^{K,L} s_{W_k}\left(x_{W_k,l}^{(j)} - \mu_x^{(j)}\right)$ (i.e., observing $\mathbf{x}_{t,l}$ directly) is unbiased and $(b)$ follows through the application of Jensen's inequality and triangle inequality. To simplify this bound, the triangle inequality yields:

$$
\mathbb{E}\left[\left| \phi_{B'}^{-1}\left(\widetilde{n}_{W_k,l}^{(j)}\right) - x_{W_k,l}^{(j)} \right|\right] \leq \mathbb{E}\left[\left| \phi_{B'}^{-1}\left(\widetilde{n}_{W_k,l}^{(j)}\right) - \phi_{B'}^{-1}\left(\overline{n}_{W_k,l}^{(j)}\right) \right|\right] + \mathbb{E}\left[\left| \phi_{B'}^{-1}\left(\overline{n}_{W_k,l}^{(j)}\right) - x_{W_k,l}^{(j)} \right|\right].
$$

(19)

Then, to bound each of these terms, we establish a piece-wise linear Lipschitz-type bound on $\phi_{B'}^{-1}(z)$. First, consider the first term $\mathbb{E}\left[\left| \phi_{B'}^{-1}\left(\widetilde{n}_{W_k,l}^{(j)}\right) - \phi_{B'}^{-1}\left(\overline{n}_{W_k,l}^{(j)}\right) \right|\right]$. We seek to upper-bound this expectation by bounding $\left| \phi_{B'}^{-1}\left(\widetilde{n}_{W_k,l}^{(j)}\right) - \phi_{B'}^{-1}\left(\overline{n}_{W_k,l}^{(j)}\right) \right|$ via the following technical lemma:

Lemma 2. Conditioned on event $\mathscr{A}_W$, the following bound holds for all $j = 1, \cdots, N$, $k = 1, \cdots, K$ and $l = 1, \cdots, L$:

$$
\varepsilon\left(\widetilde{n}_{W_k,l}^{(j)}, \overline{n}_{W_k,l}^{(j)}\right) := \left| \phi_{B'}^{-1}\left(\widetilde{n}_{W_k,l}^{(j)}\right) - \phi_{B'}^{-1}\left(\overline{n}_{W_k,l}^{(j)}\right) \right| \leq g(B)\left| \widetilde{n}_{W_k,l}^{(j)} - \overline{n}_{W_k,l}^{(j)} \right|,
$$

where

$$
g(B) = \max\left\{ 4(1 + \exp(B))^2, 4\exp(-B)\log(2\exp(B) + 1)\left(1 + (2\exp(B) + 1)^2\right) \right\}.
$$

Proof of Lemma 2. First, consider the case $\overline{n}_{W_k,l}^{(j)} \leq 0.5$. We bound the function $\varepsilon\left(\widetilde{n}_{W_k,l}^{(j)}, \overline{n}_{W_k,l}^{(j)}\right)$ in a piece-wise fashion as follows. Note that $\phi_{B'}^{-1}\left(\widetilde{n}_{W_k,l}^{(j)}\right)$ is convex for $\widetilde{n}_{W_k,l}^{(j)} \geq 0.5$ and concave for $\widetilde{n}_{W_k,l}^{(j)} \leq 0.5$. Thus, it immediately follows that for $\widetilde{n}_{W_k,l}^{(j)} \leq \overline{n}_{W_k,l}^{(j)}$, $\varepsilon\left(\widetilde{n}_{W_k,l}^{(j)}, \overline{n}_{W_k,l}^{(j)}\right)$ is convex and hence:

$$
\varepsilon\left(\widetilde{n}_{W_k,l}^{(j)}, \overline{n}_{W_k,l}^{(j)}\right) \leq \frac{\left| B' + \phi_{B'}^{-1}\left(\overline{n}_{W_k,l}^{(j)}\right) \right|}{\left| \overline{n}_{W_k,l}^{(j)} - \frac{1}{1+\exp(B')} \right|} \left(\overline{n}_{W_k,l}^{(j)} - \widetilde{n}_{W_k,l}^{(j)}\right).
$$

(20)

Furthermore, for $\overline{n}_{W_k,l}^{(j)} \leq \widetilde{n}_{W_k,l}^{(j)} \leq 0.5$, $\varepsilon\left(\widetilde{n}_{W_k,l}^{(j)}, \overline{n}_{W_k,l}^{(j)}\right)$ is concave, and hence is bounded by the tangent at $\overline{n}_{W_k,l}^{(j)}$:

$$
\varepsilon\left(\widetilde{n}_{W_k,l}^{(j)}, \overline{n}_{W_k,l}^{(j)}\right) \leq \frac{1}{\overline{n}_{W_k,l}^{(j)}(1 - \overline{n}_{W_k,l}^{(j)})}\left(\widetilde{n}_{W_k,l}^{(j)} - \overline{n}_{W_k,l}^{(j)}\right).
$$

(21)

Finally, for the case of $\widetilde{n}_{W_k,l}^{(j)} \geq 0.5$, consider the line,

$$
h\left(\widetilde{n}_{W_k,l}^{(j)}, \overline{n}_{W_k,l}^{(j)}\right) := \frac{\left| B' - \phi_{B'}^{-1}\left(\overline{n}_{W_k,l}^{(j)}\right) \right|}{\left| \frac{1}{1+\exp(-B')} - \overline{n}_{W_k,l}^{(j)} \right|} \left(\widetilde{n}_{W_k,l}^{(j)} - \overline{n}_{W_k,l}^{(j)}\right).
$$

(22)

From the convexity of $\varepsilon\left(\widetilde{n}_{W_k,l}^{(j)}, \overline{n}_{W_k,l}^{(j)}\right)$, $h\left(\widetilde{n}_{W_k,l}^{(j)}, \overline{n}_{W_k,l}^{(j)}\right)$ upper bounds $\varepsilon\left(\widetilde{n}_{W_k,l}^{(j)}, \overline{n}_{W_k,l}^{(j)}\right)$ for $\widetilde{n}_{W_k,l}^{(j)} \geq 0.5$, since $h\left(0.5, \overline{n}_{W_k,l}^{(j)}\right) \geq \varepsilon\left(0.5, \overline{n}_{W_k,l}^{(j)}\right)$ for $\overline{n}_{W_k,l}^{(j)} \leq 0.5$. Combining the piece-wise bounds in *Equation 20*, *Equation 21* and *Equation 22*, we conclude that for $\overline{n}_{W_k,l}^{(j)} \leq 0.5$:

$$\varepsilon\left(\widetilde{n}_{W_k,l}^{(j)}, \overline{n}_{W_k,l}^{(j)}\right) \leq \widetilde{g}\left(\overline{n}_{W_k,l}^{(j)}, B'\right)\left|\widetilde{n}_{W_k,l}^{(j)} - \overline{n}_{W_k,l}^{(j)}\right|, \tag{23}$$

where

$$\widetilde{g}\left(\overline{n}_{W_k,l}^{(j)}, B'\right) = \max\left\{\frac{1}{\overline{n}_{W_k,l}^{(j)}(1 - \overline{n}_{W_k,l}^{(j)})}, \frac{\left|B' + \phi_{B'}^{-1}\left(\overline{n}_{W_k,l}^{(j)}\right)\right|}{\left|\overline{n}_{W_k,l}^{(j)} - \frac{1}{1+\exp(B')}\right|}, \frac{\left|B' - \phi_{B'}^{-1}\left(\overline{n}_{W_k,l}^{(j)}\right)\right|}{\left|\frac{1}{1+\exp(-B')} - \overline{n}_{W_k,l}^{(j)}\right|}\right\}.$$

Due to the symmetry of $\varepsilon\left(\widetilde{n}_{W_k,l}^{(j)}, \overline{n}_{W_k,l}^{(j)}\right)$, the same bound in *Equation 23* can be established for $\overline{n}_{W_k,l}^{(j)} > 0.5$ as well.

Then, using $\left|\phi_{B'}^{-1}\left(\overline{n}_{W_k,l}^{(j)}\right)\right| \leq B'$ and conditioning on event $\mathscr{A}_W$, we simplify this bound as:

$$\widetilde{g}\left(\overline{n}_{W_k,l}^{(j)}, B'\right) \leq \max\left\{4(1+\exp(B))^2, \frac{4B'(1+\exp(B'))(1+\exp(B))}{\exp(B') - (2\exp(B)+1)}\right\}.$$

Finally, based on the fact that $B' = 2\log(2\exp(B)+1)$, the latter is further upper bounded as:

$$\widetilde{g}\left(\overline{n}_{W_k,l}^{(j)}, B'\right) \leq g(B),$$

where

$$g(B) = \max\left\{4(1+\exp(B))^2, 4\exp(-B)\log(2\exp(B)+1)\left(1 + (2\exp(B)+1)^2\right)\right\}.$$

This concludes the proof of Lemma 2. ∎

Following Lemma 2, by conditioning on the event $\mathscr{A}_W$ we have:

$$\mathbb{E}_{\mathscr{A}_W}\left[\left|\phi_{B'}^{-1}\left(\widetilde{n}_{W_k,l}^{(j)}\right) - \phi_{B'}^{-1}\left(\overline{n}_{W_k,l}^{(j)}\right)\right|\right] \leq g(B)\mathbb{E}_{\mathscr{A}_W}\left[\left|\widetilde{n}_{W_k,l}^{(j)} - \overline{n}_{W_k,l}^{(j)}\right|\right]. \tag{24}$$

Then, we note that:

$$\mathbb{E}\left[\left|\widetilde{n}_{W_k,l}^{(j)} - \overline{n}_{W_k,l}^{(j)}\right|\right] \overset{(c)}{\leq} \sqrt{\mathbb{E}\left[\left|\widetilde{n}_{W_k,l}^{(j)} - \overline{n}_{W_k,l}^{(j)}\right|^2\right]} \overset{(d)}{=} \frac{\sigma_w\sqrt{1+\alpha^2}}{\sqrt{W}}, \tag{25}$$

where in $(c)$ we have used the Cauchy-Schwarz inequality, and in $(d)$ we have used the fact that the observation noise across the $W$ time instances is i.i.d. and white. From the bounds in *Equation 24* and *Equation 25*, we conclude that the first expectation in *Equation 19*, conditioned on event $\mathscr{A}_W$ is bounded as:

$$\begin{aligned}\mathbb{E}_{\mathscr{A}_W}\left[\left|\phi_{B'}^{-1}\left(\widetilde{n}_{W_k,l}^{(j)}\right) - \phi_{B'}^{-1}\left(\overline{n}_{W_k,l}^{(j)}\right)\right|\right] &\leq g(B)\mathbb{E}_{\mathscr{A}_W}\left[\left|\widetilde{n}_{W_k,l}^{(j)} - \overline{n}_{W_k,l}^{(j)}\right|\right] \\ &\leq g(B)\frac{\sigma_w\sqrt{1+\alpha^2}}{\sqrt{W}\mathbb{P}(\mathscr{A}_W)}.\end{aligned} \tag{26}$$

The foregoing sequence of reasoning similarly follows for $\mathbb{E}\left[\left|\phi_{B'}^{-1}\left(\overline{n}_{W_k,l}^{(j)}\right) - x_{W_k,l}^{(j)}\right|\right]$, since $\frac{1}{1+\exp(B)} \leq \phi\left(x_{W_k,l}^{(j)}\right) \leq 1 - \frac{1}{1+\exp(B)}$ for $k=1,\cdots,K$, $l=1,\cdots,L$ and $j=1,\cdots,N$ (as a consequence of $|x_{W_k,l}^{(j)}| < B$ for $k=1,\cdots,K$, $l=1,\cdots,L$ and $j=1,\cdots,N$, conditioned on $\mathscr{A}_W$). Accordingly, we derive the upper bound on the second term in *Equation 19*, conditioned on event $\mathscr{A}_W$:

$$\mathbb{E}_{\mathscr{A}_W}\left[\left|\phi_{B'}^{-1}\left(\bar{n}_{W_k,l}^{(j)}\right) - x_{W_k,l}^{(j)}\right|\right] \leq g(B)\mathbb{E}_{\mathscr{A}_W}\left[\left|\bar{n}_{W_k,l}^{(j)} - \phi\left(x_{W_k,l}^{(j)}\right)\right|\right]$$

$$\overset{(e)}{\leq} \frac{g(B)}{W\,\mathbb{P}(\mathscr{A}_W)}\sqrt{\mathbb{E}\left[\left(\sum_{w=1}^{W}n_{(k-1)W+w,l}^{(j)} - W\phi\left(x_{W_k,l}^{(j)}\right)\right)^2\right]}$$

$$\overset{(f)}{=} \frac{g(B)}{W\,\mathbb{P}(\mathscr{A}_W)}\sqrt{\mathbb{E}\left[W\phi\left(x_{W_k,l}^{(j)}\right)\left(1 - \phi\left(x_{W_k,l}^{(j)}\right)\right)\right]}$$

$$\overset{(g)}{\leq} \frac{g(B)}{2\sqrt{W}\,\mathbb{P}(\mathscr{A}_W)},$$

(27)

where $(e)$ follows from the application of Jensen's inequality, $(f)$ follows from the formula for the variance of a Binomial random variable, and $(g)$ follows from the inequality $\phi\left(x_{W_k,l}^{(j)}\right)\left(1 - \phi\left(x_{W_k,l}^{(j)}\right)\right) \leq 1/4$, for $\phi\left(x_{W_k,l}^{(j)}\right) \in [0,1]$. Combining the results in *Equation 26* and *Equation 27*, the overall expectation in *Equation 19*, conditioned on the event $\mathscr{A}_W$ is upper-bounded by:

$$\mathbb{E}_{\mathscr{A}_W}\left[\left|\phi_{B'}^{-1}\left(\tilde{n}_{W_k,l}^{(j)}\right) - x_{W_k,l}^{(j)}\right|\right] \leq \frac{2g(B)}{\sqrt{W}}\left(\sigma_w\sqrt{1+\alpha^2} + \frac{1}{2}\right),$$

(28)

where we have lower bounded the probability of the event $\mathscr{A}_W$ by $1/2$ (that is, $\mathbb{P}(\mathscr{A}_W) > 1/2$). Thus, from *Equation 18* and *Equation 28* we derive:

$$\left|bias_{\mathscr{A}_W}\left(\tilde{d}_j\right)\right| \leq \frac{2g(B)}{\sqrt{W}}\left(\sigma_w\sqrt{1+\alpha^2} + \frac{1}{2}\right)\frac{\sum_{k,l=1}^{K,L}|s_{W_k}|}{L\sum_{k=1}^{K}s_{W_k}^2} \overset{(h)}{\leq} \frac{2g(B)}{\sigma_s\sqrt{W}}\left(\sigma_w\sqrt{1+\alpha^2} + \frac{1}{2}\right),$$

where in $(h)$ we have used the Cauchy-Schwarz inequality $\sum_{k=1}^{K}|s_{W_k}| \leq \sqrt{K}\sqrt{\sum_{k=1}^{K}s_{W_k}^2}$ while defining $\sigma_s^2 := \frac{1}{K}\sum_{k=1}^{K}s_{W_k}^2$.

Then, for $B \geq 2.5$, we have $g(B) = 4(1 + \exp(B))^2$ and $B' = 2\log(2\exp(B) + 1) \leq 3B$. Let $B := \sigma_m\sqrt{8q\log W}$ for some $q > \frac{1}{64}$. Further, for some $\epsilon < 1/2$, suppose that:

$$\log W \geq \max\left\{\frac{\log(8KLN/\eta)}{q}, \frac{32\sigma_m^2 q}{\epsilon^2}, \frac{2\log(64q)}{1-2\epsilon}, \frac{\max\left\{6.25, 4\left(\|\boldsymbol{\mu}_x\|_\infty + \max_{k,j}\{|s_{W_k}d_j|\}\right)^2\right\}}{8q\sigma_m^2}, \log 2\right\}.$$

(29)

Under these conditions,

$$g(B) \leq 4\left(1 + \exp(\sigma_m\sqrt{8q\log W})\right)^2 \overset{(i)}{\leq} 16\exp\left(2\sigma_m\sqrt{8q\log W}\right) \leq 16W^\epsilon,$$

(30)

where in $(i)$ we have used the fact that $e^x \geq 1$ for $x \geq 0$. Thus, under the conditions in *Equation 29*, we have:

$$\left|bias_{\mathscr{A}_W}\left(\tilde{d}_j\right)\right| \leq \frac{32}{\sigma_s\sqrt{W^{1-2\epsilon}}}\left(\sigma_w\sqrt{1+\alpha^2} + \frac{1}{2}\right).$$

(31)

Finally, from *Equation 16* and *Equation 31*, we conclude that:

$$\left|bias_{\mathscr{A}_W}\left(\hat{d}_j\right)\right| \leq \frac{1}{\sqrt{W^{1-2\epsilon}}}C_1\left(2\sigma_w\sqrt{1+\alpha^2} + 1\right) + \mathcal{O}(\sigma_w^2) + \mathcal{O}\left(\frac{1}{W}\right),$$

where $C_1 := \frac{16}{\sigma_s}$.

## Bounding the variance of $\hat{d}_j$

Next, we prove the upper bound on the variance of the maximum likelihood estimator, $\hat{d}_j$. To that end, we upper-bound the variance of $\tilde{d}_j$. First, using the Cauchy-Schwarz inequality, we have:

$$Var\left(\widetilde{d}_j\right) := \mathbb{E}\left[\left|\widetilde{d}_j - \mathbb{E}\left[\widetilde{d}_j\right]\right|^2\right] \leq \left\{\sqrt{\mathbb{E}\left[\left|\widetilde{d}_j - (d_{\mathrm{Oracle}})_j\right|^2\right]} + \sqrt{Var\left((d_{\mathrm{Oracle}})_j\right)}\right\}^2. \tag{32}$$

Then, we upper-bound the conditional second moment of $\left|\widetilde{d}_j - (d_{\mathrm{Oracle}})_j\right|$ using the same techniques as we used in bounding the first moment. Accordingly, we get:

$$
\begin{aligned}
\mathbb{E}_{\mathscr{A}_W}\left[\left|\widetilde{d}_j - (d_{\mathrm{Oracle}})_j\right|^2\right] &= \frac{1}{\left(L\sum_{k=1}^K s_{W_k}^2\right)^2}\mathbb{E}_{\mathscr{A}_W}\left[\left|\sum_{k,l=1}^{K,L} s_{W_k}\left(\phi_{B'}^{-1}\left(\widetilde{n}_{W_k,l}^{(j)}\right) - x_{W_k,l}^{(j)}\right)\right|^2\right] \\
&\overset{(j)}{\leq} \frac{1}{\left(L\sum_{k=1}^K s_{W_k}^2\right)^2}\left\{\sum_{k,l=1}^{K,L}|s_{W_k}|\sqrt{\mathbb{E}_{\mathscr{A}_W}\left[\left|\phi_{B'}^{-1}\left(\widetilde{n}_{W_k,l}^{(j)}\right) - x_{W_k,l}^{(j)}\right|^2\right]}\right\}^2 \\
&\overset{(k)}{\leq} \left\{\frac{\sqrt{2}g(B)}{\sigma_s\sqrt{W}}\left(\sigma_w\sqrt{1+\alpha^2}+\frac{1}{2}\right)\right\}^2,
\end{aligned}
\tag{33}
$$

where in $(j)$, we have used the Cauchy-Schwarz inequality and $(k)$ follows from $\mathbb{E}_{\mathscr{A}_W}\left[\left|\phi_{B'}^{-1}\left(\widetilde{n}_{W_k,l}^{(j)}\right) - x_{W_k,l}^{(j)}\right|^2\right] \leq \frac{2(g(B))^2}{W}\left(\sigma_w\sqrt{1+\alpha^2}+\frac{1}{2}\right)^2$, which can be proven by the same techniques as before.

Next, we note that the variance of the Oracle estimator $(d_{\mathrm{Oracle}})_j$ is given by:

$$Var\left((d_{\mathrm{Oracle}})_j\right) = \frac{1}{\left(L\sum_{k=1}^K s_{W_k}^2\right)^2}\sum_{k,l=1}^{K,L} s_{W_k}^2\, Var\left(\left(x_{W_k,l}^{(j)} - \mu_x^{(j)}\right)\right) = \frac{(\mathbf{\Sigma}_x)_{j,j}}{L\sum_{k=1}^K s_{W_k}^2} = \frac{(\mathbf{\Sigma}_x)_{j,j}}{LK\sigma_s^2}. \tag{34}$$

Combining *Equation 32*, *Equation 33,* and *Equation 34*, we can upper-bound the conditional variance of $\widetilde{d}_j$ following *Equation 32* as:

$$\sqrt{Var_{\mathscr{A}_W}\left(\widetilde{d}_j\right)} \leq \sqrt{\frac{(\mathbf{\Sigma}_x)_{j,j}}{KL\sigma_s^2(1-\eta)}} + \frac{\sqrt{2}g(B)}{\sigma_s\sqrt{W}}\left(\sigma_w\sqrt{1+\alpha^2}+\frac{1}{2}\right).$$

Then, following *Equation 16*, under the conditions for $W$ in *Equation 29*, we conclude the proof of the conditional variance of $\widehat{d}_j$:

$$\sqrt{Var_{\mathscr{A}_W}\left(\widehat{d}_j\right)} \leq \sqrt{\frac{(\mathbf{\Sigma}_x)_{j,j}}{KL\sigma_s^2(1-\eta)}} + \frac{1}{\sqrt{W^{1-2\epsilon}}}C_2\left(2\sigma_w\sqrt{1+\alpha^2}+1\right) + \mathcal{O}(\sigma_w^2) + \mathcal{O}\left(\frac{1}{W}\right), \tag{35}$$

where $C_2 := \frac{8\sqrt{2}}{\sigma_s}$.

## Bounding the bias of $(\widehat{\mathbf{\Sigma}}_x)_{i,j}$

Next, following the foregoing techniques, we upper-bound the bias and variance of the noise covariance estimator $(\widehat{\mathbf{\Sigma}}_x)_{i,j}$. To that end, we first note:

$$
\begin{aligned}
\left|bias\left((\widetilde{\mathbf{\Sigma}}_x)_{i,j}\right)\right| &:= \left|\mathbb{E}\left[(\widetilde{\mathbf{\Sigma}}_x)_{i,j}\right] - (\mathbf{\Sigma}_x)_{i,j}\right| \\
&\overset{(l)}{\leq} \left|\mathbb{E}\left[(\widetilde{\mathbf{\Sigma}}_x)_{i,j} - (\mathbf{\Sigma}_{\mathrm{Oracle}})_{i,j}\right]\right| + \left|bias\left((\mathbf{\Sigma}_{\mathrm{Oracle}})_{i,j}\right)\right|,
\end{aligned}
\tag{36}
$$

where $(l)$ follows from the triangle inequality, with the Oracle noise covariance estimator (i.e., observing $\mathbf{x}_{t,l}$ directly), being defined as:

$$(\mathbf{\Sigma}_{\mathrm{Oracle}})_{i,j} = \frac{1}{KL}\sum_{k,l=1}^{K,L}\left(x_{W_k,l}^{(i)} - \mu_x^{(i)} - s_{W_k}(d_{\mathrm{Oracle}})_i\right)\left(x_{W_k,l}^{(j)} - \mu_x^{(j)} - s_{W_k}(d_{\mathrm{Oracle}})_j\right).$$

Then, to simplify the first term in *Equation 36*, we use similar techniques as before. Accordingly,

$$\left| \mathbb{E}\left[ (\widetilde{\Sigma}_x)_{i,j} - (\Sigma_{\text{Oracle}})_{i,j} \right] \right| = \left| \mathbb{E}\left[ \frac{1}{KL} \sum_{k,l=1}^{KL} (\phi_{B'}^{-1}(\widetilde{n}_{W_k,l}^{(i)}) - \mu_x^{(i)} - s_{W_k}\widetilde{d}_i)(\phi_{B'}^{-1}(\widetilde{n}_{W_k,l}^{(j)}) - \mu_x^{(j)} - s_{W_k}\widetilde{d}_j) \right. \right.$$

$$\left. \left. - \frac{1}{KL} \sum_{k,l=1}^{KL} (x_{W_k,l}^{(i)} - \mu_x^{(i)} - s_{W_k}(d_{\text{Oracle}})_i)(x_{W_k,l}^{(j)} - \mu_x^{(j)} - s_{W_k}(d_{\text{Oracle}})_j) \right] \right|$$

$$\overset{(m)}{\leq} \frac{1}{KL} \sum_{k,l=1}^{K,L} \mathbb{E}\left[ \left| (\phi_{B'}^{-1}(\widetilde{n}_{W_k,l}^{(i)}) - \mu_x^{(i)})(\phi_{B'}^{-1}(\widetilde{n}_{W_k,l}^{(j)}) - \mu_x^{(j)}) \right. \right.$$

$$\left. \left. - (x_{W_k,l}^{(i)} - \mu_x^{(i)})(x_{W_k,l}^{(j)} - \mu_x^{(j)}) \right| \right]$$

$$+ \frac{1}{KL^2 \sum_{k=1}^{K} s_{W_k}^2} \mathbb{E}\left[ \left| \sum_{k,l=1}^{K,L} s_{W_k}(\phi_{B'}^{-1}(\widetilde{n}_{W_k,l}^{(i)}) - \mu_x^{(i)}) \sum_{k',l'=1}^{K,L} s_{W_{k'}}(\phi_{B'}^{-1}(\widetilde{n}_{W_{k'},l'}^{(j)}) - \mu_x^{(j)}) \right. \right.$$

$$\left. \left. - \sum_{k,l=1}^{K,L} s_{W_k}(x_{W_k,l}^{(i)} - \mu_x^{(i)}) \sum_{k',l'=1}^{K,L} s_{W_{k'}}(x_{W_{k'},l'}^{(j)} - \mu_x^{(j)}) \right| \right], \quad (37)$$

where $(m)$ follows through the application of Jensen's inequality and triangle inequality. Next, conditioned on the event $\mathscr{A}_W$ we have:

$$\mathbb{E}_{\mathscr{A}_W}\left[ \left| \left( \phi_{B'}^{-1}\left( \widetilde{n}_{W_k,l}^{(i)} \right) - \mu_x^{(i)} \right)\left( \phi_{B'}^{-1}\left( \widetilde{n}_{W_k,l}^{(j)} \right) - \mu_x^{(j)} \right) - \left( x_{W_k,l}^{(i)} - \mu_x^{(i)} \right)\left( x_{W_k,l}^{(j)} - \mu_x^{(j)} \right) \right| \right]$$

$$\leq \mathbb{E}_{\mathscr{A}_W}\left[ \left| \phi_{B'}^{-1}\left( \widetilde{n}_{W_k,l}^{(i)} \right)\phi_{B'}^{-1}\left( \widetilde{n}_{W_k,l}^{(j)} \right) - x_{W_k,l}^{(i)} x_{W_k,l}^{(j)} \right| \right]$$

$$+ \mu_x^{(j)} \mathbb{E}_{\mathscr{A}_W}\left[ \left| \phi_{B'}^{-1}\left( \widetilde{n}_{W_k,l}^{(i)} \right) - x_{W_k,l}^{(i)} \right| \right] + \mu_x^{(i)} \mathbb{E}_{\mathscr{A}_W}\left[ \left| \phi_{B'}^{-1}\left( \widetilde{n}_{W_k,l}^{(j)} \right) - x_{W_k,l}^{(j)} \right| \right]$$

$$\leq \mathbb{E}_{\mathscr{A}_W}\left[ \left| \phi_{B'}^{-1}\left( \widetilde{n}_{W_k,l}^{(i)} \right)\phi_{B'}^{-1}\left( \widetilde{n}_{W_k,l}^{(j)} \right) - \phi_{B'}^{-1}\left( \overline{n}_{W_k,l}^{(i)} \right)\phi_{B'}^{-1}\left( \overline{n}_{W_k,l}^{(j)} \right) \right| \right]$$

$$+ \mathbb{E}_{\mathscr{A}_W}\left[ \left| \phi_{B'}^{-1}\left( \overline{n}_{W_k,l}^{(i)} \right)\phi_{B'}^{-1}\left( \overline{n}_{W_k,l}^{(j)} \right) - x_{W_k,l}^{(i)} x_{W_k,l}^{(j)} \right| \right] + 2\mu_m \mathbb{E}_{\mathscr{A}_W}\left[ \left| \phi_{B'}^{-1}\left( \widetilde{n}_{W_k,l}^{(j)} \right) - x_{W_k,l}^{(j)} \right| \right]$$

$$\leq 2g(B)\frac{\sigma_w\sqrt{1+\alpha^2}}{\sqrt{W}}\left( g(B)\frac{\sigma_w\sqrt{1+\alpha^2}}{\sqrt{W}} + 4\log(2\exp(B)+1) \right) + \frac{2g(B)}{\sqrt{W}}\left\{ \frac{g(B)}{4\sqrt{W}} + B \right\}$$

$$+ \frac{4\mu_m g(B)}{\sqrt{W}}\left( \sigma_w\sqrt{1+\alpha^2} + \frac{1}{2} \right), \quad (38)$$

where $\mu_m = \|\mu_x\|_\infty$ and we have used $B' = 2\log(2\exp(B)+1)$. Similarly, conditioned on the event $\mathscr{A}_W$ the second term in *Equation 37* can be bounded as:

$$\mathbb{E}_{\mathscr{A}_W}\left[ \left| \sum_{k,l=1}^{K,L} s_{W_k}\left( \phi_{B'}^{-1}\left( \widetilde{n}_{W_k,l}^{(i)} \right) - \mu_x^{(i)} \right) \sum_{k',l'=1}^{K,L} s_{W_{k'}}\left( \phi_{B'}^{-1}\left( \widetilde{n}_{W_{k'},l'}^{(j)} \right) - \mu_x^{(j)} \right) \right. \right.$$

$$\left. \left. - \sum_{k,l=1}^{K,L} s_{W_k}\left( x_{W_k,l}^{(i)} - \mu_x^{(i)} \right) \sum_{k',l'=1}^{K,L} s_{W_{k'}}\left( x_{W_{k'},l'}^{(j)} - \mu_x^{(j)} \right) \right| \right]$$

$$\leq \sum_{k,k',l,l'=1}^{K,K,L,L} \left| s_{W_k} s_{W_{k'}} \right| \mathbb{E}_{\mathscr{A}_W}\left[ \left| \phi_{B'}^{-1}\left( \widetilde{n}_{W_k,l}^{(i)} \right)\phi_{B'}^{-1}\left( \widetilde{n}_{W_{k'},l'}^{(j)} \right) - x_{W_k,l}^{(i)} x_{W_{k'},l'}^{(j)} \right| \right] \quad (39)$$

$$+ \mu_x^{(i)} \sum_{k,k',l,l'=1}^{K,K,L,L} \left| s_{W_k} s_{W_{k'}} \right| \mathbb{E}_{\mathscr{A}_W}\left[ \left| \phi_{B'}^{-1}\left( \widetilde{n}_{W_{k'},l'}^{(j)} \right) - x_{W_{k'},l'}^{(j)} \right| \right]$$

$$+ \mu_x^{(j)} \sum_{k,k',l,l'=1}^{K,K,L,L} \left| s_{W_k} s_{W_{k'}} \right| \mathbb{E}_{\mathscr{A}_W}\left[ \left| \phi_{B'}^{-1}\left( \widetilde{n}_{W_k,l}^{(i)} \right) - x_{W_k,l}^{(i)} \right| \right]$$

$$\leq \left( L\sum_{k=1}^{K}|s_{W_k}| \right)^2 \left\{ 2g(B)\frac{\sigma_w\sqrt{1+\alpha^2}}{\sqrt{W}}\left( g(B)\frac{\sigma_w\sqrt{1+\alpha^2}}{\sqrt{W}} + 4\log(2\exp(B)+1) \right) \right.$$

$$\left. + \frac{2g(B)}{\sqrt{W}}\left\{ \frac{g(B)}{4\sqrt{W}} + B \right\} + \frac{4\mu_m g(B)}{\sqrt{W}}\left( \sigma_w\sqrt{1+\alpha^2} + \frac{1}{2} \right) \right\}. \quad (40)$$

Then, by combining the bounds in *Equation 38* and *Equation 40* and using an instance of Cauchy-Schwarz inequality $\left( \sum_{k=1}^{K}|s_{W_k}| \right)^2 \leq K\sum_{k=1}^{K} s_{W_k}^2$, we see that the bound in *Equation 37* conditioned on the event $\mathscr{A}_W$ can be expressed as:

$$\left|\mathbb{E}_{\mathscr{A}_w}\left[(\widetilde{\boldsymbol{\Sigma}}_x)_{i,j} - (\boldsymbol{\Sigma}_{\text{Oracle}})_{i,j}\right]\right| \leq 4g(B)\frac{\sigma_w\sqrt{1+\alpha^2}}{\sqrt{W}}\left(g(B)\frac{\sigma_w\sqrt{1+\alpha^2}}{\sqrt{W}} + 4\log\left(2\exp\left(B\right)+1\right) + 2\mu_m\right)$$
$$+ \frac{4g(B)}{\sqrt{W}}\left\{\frac{g(B)}{4\sqrt{W}} + B + \mu_m\right\}. \tag{41}$$

Next, we see that the oracle estimator follows an Inverse Wishart distribution, that is $KL\boldsymbol{\Sigma}_{\text{Oracle}} \sim InvWish_N(\boldsymbol{\Sigma}_x, KL-1)$. Therefore, we get:

$$\mathbb{E}[\boldsymbol{\Sigma}_{\text{Oracle}}] = \frac{(KL-1)}{KL}\boldsymbol{\Sigma}_x.$$

Thus, the bias of the oracle estimator is given by:

$$\left|bias\left((\boldsymbol{\Sigma}_{\text{Oracle}})_{i,j}\right)\right| = \frac{1}{KL}|(\boldsymbol{\Sigma}_x)_{i,j}|. \tag{42}$$

Combining the results in *Equation 41* and *Equation 42*, the bias of $(\widetilde{\boldsymbol{\Sigma}}_x)_{i,j}$ can be bounded as:

$$\left|bias_{\mathscr{A}_w}\left((\widetilde{\boldsymbol{\Sigma}}_x)_{i,j}\right)\right| \leq \frac{|(\boldsymbol{\Sigma}_x)_{i,j}|}{KL(1-\eta)} + 4g(B)\frac{\sigma_w\sqrt{1+\alpha^2}}{\sqrt{W}}\left(4\log\left(2\exp\left(B\right)+1\right) + 2\mu_m\right)$$
$$+ \frac{4g(B)}{\sqrt{W}}(B+\mu_m) + \mathcal{O}\left(\frac{g(B)^2}{W}\right). \tag{43}$$

Finally, under the conditions for $W$ in *Equation 29*, the latter inequality simplifies to:

$$\left|bias_{\mathscr{A}_w}\left((\widetilde{\boldsymbol{\Sigma}}_x)_{i,j}\right)\right| \overset{(n)}{\leq} \frac{|(\boldsymbol{\Sigma}_x)_{i,j}|}{KL(1-\eta)} + \frac{Bg(B)}{\sqrt{W}}\left(28\sigma_w\sqrt{1+\alpha^2}+6\right) + \mathcal{O}\left(\frac{g(B)^2}{W}\right)$$
$$\overset{(o)}{\leq} \frac{|(\boldsymbol{\Sigma}_x)_{i,j}|}{KL(1-\eta)} + 64\sigma_m\sqrt{\frac{2q\log W}{W^{1-2\epsilon}}}\left(14\sigma_w\sqrt{1+\alpha^2}+3\right) + \mathcal{O}\left(\frac{1}{W^{1-2\epsilon}}\right), \tag{44}$$

where in $(n)$ we have used $2\log(2\exp(B)+1) \leq 3B$ and $B>2\mu_m$ and in $(o)$ we have used $Bg(B) \leq 16L^\epsilon\sigma_m\sqrt{8q\log L}$, which follows from *Equation 30*. Thus, following *Equation 17* we derive the bound on the bias of the maximum likelihood estimator:

$$\left|bias_{\mathscr{A}_w}\left((\widehat{\boldsymbol{\Sigma}}_x)_{i,j}\right)\right| \leq \frac{|(\boldsymbol{\Sigma}_x)_{i,j}|}{KL(1-\eta)} + \sqrt{\frac{\log W}{W^{1-2\epsilon}}}C_3\left(14\sigma_w\sqrt{1+\alpha^2}+3\right) + \mathcal{O}(\sigma_w^2) + \mathcal{O}\left(\frac{1}{W^{1-2\epsilon}}\right),$$

where $C_3 := 64\sigma_m\sqrt{2q}$.

## Bounding the variance of $(\widehat{\boldsymbol{\Sigma}}_x)_{i,j}$

Next, we establish an upper bound on the variance of the maximum likelihood estimator of the noise covariance. To that end, we upper-bound the variance of $(\widetilde{\boldsymbol{\Sigma}}_x)_{i,j}$. First, using the Cauchy-Schwarz inequality, we get:

$$Var\left((\widetilde{\boldsymbol{\Sigma}}_x)_{i,j}\right) := \mathbb{E}\left[\left|(\widetilde{\boldsymbol{\Sigma}}_x)_{i,j} - \mathbb{E}\left[(\widetilde{\boldsymbol{\Sigma}}_x)_{i,j}\right]\right|^2\right]$$
$$\leq \left\{\sqrt{\mathbb{E}\left[\left|(\widetilde{\boldsymbol{\Sigma}}_x)_{i,j} - (\boldsymbol{\Sigma}_{\text{Oracle}})_{i,j}\right|^2\right]} + \sqrt{Var\left((\boldsymbol{\Sigma}_{\text{Oracle}})_{i,j}\right)}\right\}^2. \tag{45}$$

Then, we upper-bound the conditional second moment of $\left|(\widetilde{\boldsymbol{\Sigma}}_x)_{i,j} - (\boldsymbol{\Sigma}_{\text{Oracle}})_{i,j}\right|$ using the same techniques used in bounding its first moment. Accordingly, we derive:

$$\mathbb{E}_{\mathscr{A}_W}\left[\left|\left(\widetilde{\mathbf{\Sigma}}_x\right)_{i,j}-\left(\mathbf{\Sigma}_{\text{Oracle}}\right)_{i,j}\right|^2\right]$$

$$=\frac{1}{K^2L^2}\mathbb{E}_{\mathscr{A}_W}\left[\left\{\sum_{k,l=1}^{K,L}\left(\left(\phi_{B'}^{-1}\left(\widetilde{n}_{W_k,l}^{(i)}\right)-\mu_x^{(i)}-s_{W_k}\widetilde{d}_i\right)\left(\phi_{B'}^{-1}\left(\widetilde{n}_{W_k,l}^{(j)}\right)-\mu_x^{(j)}-s_{W_k}\widetilde{d}_j\right)\right.\right.\right.$$

$$\left.\left.\left.-\left(x_{W_k,l}^{(i)}-\mu_x^{(i)}-s_{W_k}(d_{\text{Oracle}})_i\right)\left(x_{W_k,l}^{(j)}-\mu_x^{(j)}-s_{W_k}(d_{\text{Oracle}})_j\right)\right)\right\}^2\right]\tag{46}$$

$$\leq\frac{1}{K^2L^2}\left\{\sum_{k,l=1}^{K,L}\left\{\mathbb{E}_{\mathscr{A}_W}\left[\left(\left(\phi_{B'}^{-1}\left(\widetilde{n}_{W_k,l}^{(i)}\right)-\mu_x^{(i)}-s_{W_k}\widetilde{d}_i\right)\left(\phi_{B'}^{-1}\left(\widetilde{n}_{W_k,l}^{(j)}\right)-\mu_x^{(j)}-s_{W_k}\widetilde{d}_j\right)\right.\right.\right.\right.$$

$$\left.\left.\left.\left.-\left(x_{W_k,l}^{(i)}-\mu_x^{(i)}-s_{W_k}(d_{\text{Oracle}})_i\right)\left(x_{W_k,l}^{(j)}-\mu_x^{(j)}-s_{W_k}(d_{\text{Oracle}})_j\right)\right)^2\right]\right\}^{\frac{1}{2}}\right\}^2,$$

where the last bound follows from the Cauchy-Schwarz inequality. Then, we derive:

$$\mathbb{E}_{\mathscr{A}_W}\left[\left\{\left(\phi_{B'}^{-1}\left(\widetilde{n}_{W_k,l}^{(i)}\right)-\mu_x^{(i)}-s_{W_k}\widetilde{d}_i\right)\left(\phi_{B'}^{-1}\left(\widetilde{n}_{W_k,l}^{(j)}\right)-\mu_x^{(j)}-s_{W_k}\widetilde{d}_j\right)\right.\right.$$

$$\left.\left.-\left(x_{W_k,l}^{(i)}-\mu_x^{(i)}-s_{W_k}(d_{\text{Oracle}})_i\right)\left(x_{W_k,l}^{(j)}-\mu_x^{(j)}-s_{W_k}(d_{\text{Oracle}})_j\right)\right\}^2\right]$$

$$=\mathbb{E}_{\mathscr{A}_W}\left[\left\{\left(\phi_{B'}^{-1}\left(\widetilde{n}_{W_k,l}^{(i)}\right)-\mu_x^{(i)}-s_{W_k}\frac{1}{L\sum_{k'=1}^{K}s_{W_{k'}}^2}\sum_{k',l'=1}^{K,L}s_{W_{k'}}\left(\phi_{B'}^{-1}\left(\widetilde{n}_{W_{k'},l'}^{(i)}\right)-\mu_x^{(i)}\right)\right)\right.\right.$$

$$\times\left(\phi_{B'}^{-1}\left(\widetilde{n}_{W_k,l}^{(j)}\right)-\mu_x^{(j)}-s_{W_k}\frac{1}{L\sum_{k''=1}^{K}s_{W_{k''}}^2}\sum_{k'',l''=1}^{K,L}s_{W_{k''}}\left(\phi_{B'}^{-1}\left(\widetilde{n}_{W_{k''},l''}^{(j)}\right)-\mu_x^{(j)}\right)\right)$$

$$-\left(x_{W_k,l}^{(i)}-\mu_x^{(i)}-s_{W_k}\frac{1}{L\sum_{k'=1}^{K}s_{W_{k'}}^2}\sum_{k',l'=1}^{K,L}s_{W_{k'}}\left(x_{W_{k'},l'}^{(i)}-\mu_x^{(i)}\right)\right)$$

$$\left.\left.\times\left(x_{W_k,l}^{(j)}-\mu_x^{(j)}-s_{W_k}\frac{1}{L\sum_{k''=1}^{K}s_{W_{k''}}^2}\sum_{k'',l''=1}^{K,L}s_{W_{k''}}\left(x_{W_{k''},l''}^{(j)}-\mu_x^{(j)}\right)\right)\right\}^2\right]$$

$$=\mathbb{E}_{\mathscr{A}_W}\left[\left\{\left(\left(\phi_{B'}^{-1}\left(\widetilde{n}_{W_k,l}^{(i)}\right)-\mu_x^{(i)}\right)\left(\phi_{B'}^{-1}\left(\widetilde{n}_{W_k,l}^{(j)}\right)-\mu_x^{(j)}\right)-\left(x_{W_k,l}^{(i)}-\mu_x^{(i)}\right)\left(x_{W_k,l}^{(j)}-\mu_x^{(j)}\right)\right)\right.\right.$$

$$-\frac{s_{W_k}}{L\sum_{k''=1}^{K}s_{W_{k''}}^2}\sum_{k'',l''=1}^{K,L}s_{W_{k''}}\left\{\left(\phi_{B'}^{-1}\left(\widetilde{n}_{W_k,l}^{(i)}\right)-\mu_x^{(i)}\right)\left(\phi_{B'}^{-1}\left(\widetilde{n}_{W_{k''},l''}^{(j)}\right)-\mu_x^{(j)}\right)\right.\tag{47}$$

$$\left.-\left(x_{W_k,l}^{(i)}-\mu_x^{(i)}\right)\left(x_{W_{k''},l''}^{(j)}-\mu_x^{(j)}\right)\right\}$$

$$-\frac{s_{W_k}}{L\sum_{k'=1}^{K}s_{W_{k'}}^2}\sum_{k',l'=1}^{K,L}s_{W_{k'}}\left\{\left(\phi_{B'}^{-1}\left(\widetilde{n}_{W_{k'},l'}^{(i)}\right)-\mu_x^{(i)}\right)\left(\phi_{B'}^{-1}\left(\widetilde{n}_{W_k,l}^{(j)}\right)-\mu_x^{(j)}\right)\right.$$

$$\left.-\left(x_{W_{k'},l'}^{(i)}-\mu_x^{(i)}\right)\left(x_{W_k,l}^{(j)}-\mu_x^{(j)}\right)\right\}$$

$$+\frac{s_{W_k}^2}{L^2\sum_{k',k''=1}^{K,K}s_{W_{k'}}^2s_{W_{k''}}^2}\sum_{k',k'',l',l''=1}^{K,L}s_{W_{k'}}s_{W_{k''}}\left\{\left(\phi_{B'}^{-1}\left(\widetilde{n}_{W_{k'},l'}^{(i)}\right)-\mu_x^{(i)}\right)\left(\phi_{B'}^{-1}\left(\widetilde{n}_{W_{k''},l''}^{(j)}\right)-\mu_x^{(j)}\right)\right.$$

$$\left.\left.\left.-\left(x_{W_{k'},l'}^{(i)}-\mu_x^{(i)}\right)\left(x_{W_{k''},l''}^{(j)}-\mu_x^{(j)}\right)\right\}\right\}^2\right]$$

$$\leq\frac{2(g(B))^2}{W}\left(\sigma_w\sqrt{1+\alpha^2}+\frac{1}{2}\right)^2\left(\frac{g(B)}{\sqrt{W}}\left(\sigma_w\sqrt{1+\alpha^2}+\frac{1}{2}\right)+2(B+\mu_m)\right)^2$$

$$\times\left(1+\frac{s_{W_k}L\sum_{k'=1}^{K}s_{W_{k'}}}{L\sum_{k'=1}^{K}s_{W_{k'}}^2}+\frac{s_{W_k}L\sum_{k''=1}^{K}s_{W_{k''}}}{L\sum_{k''=1}^{K}s_{W_{k''}}^2}+\frac{s_{W_k}^2L\sum_{k'=1}^{K}s_{W_{k'}}L\sum_{k''=1}^{K}s_{W_{k''}}}{L^2\sum_{k'=1}^{K}s_{W_{k'}}^2\sum_{k''=1}^{K}s_{W_{k''}}^2}\right)^2.$$

Using the final bound of *Equation 47* in *Equation 46*, we get:

$$\sqrt{\mathbb{E}_{\mathscr{A}_W}\left[\left|(\widetilde{\boldsymbol{\Sigma}}_x)_{i,j}-(\boldsymbol{\Sigma}_{\text{Oracle}})_{i,j}\right|^2\right]} \leq \frac{\sqrt{2}g(B)}{\sqrt{W}}\left(\sigma_w\sqrt{1+\alpha^2}+\frac{1}{2}\right)\left(\frac{g(B)}{\sqrt{W}}\left(\sigma_w\sqrt{1+\alpha^2}+\frac{1}{2}\right)+2(B+\mu_m)\right)$$

$$\times\frac{1}{KL}\left(KL+\frac{L\sum_{k=1}^{K}|s_{W_k}|\sum_{k'=1}^{K}|s_{W_{k'}}|}{\sum_{k'=1}^{K}s_{W_{k'}}^2}+\frac{L\sum_{k=1}^{K}|s_{W_k}|\sum_{k''=1}^{K}|s_{W_{k''}}|}{\sum_{k''=1}^{K}s_{W_{k''}}^2}\right.$$

$$\left.+\frac{L\sum_{k=1}^{K}s_{W_k}^2\sum_{k'=1}^{K}|s_{W_{k'}}|\sum_{k''=1}^{K}|s_{W_{k''}}|}{\sum_{k'=1}^{K}s_{W_{k'}}^2\sum_{k''=1}^{K}s_{W_{k''}}^2}\right) \tag{48}$$

$$\leq\frac{4\sqrt{2}g(B)}{\sqrt{W}}\left(\sigma_w\sqrt{1+\alpha^2}+\frac{1}{2}\right)\left(\frac{g(B)}{\sqrt{W}}\left(\sigma_w\sqrt{1+\alpha^2}+\frac{1}{2}\right)+2(B+\mu_m)\right),$$

where the last inequality follows from an instance of the Cauchy-Schwarz inequality, that is $\left(\sum_{k=1}^{K}|s_{W_k}|\right)^2\leq K\sum_{k=1}^{K}s_{W_k}^2$.

Then, following the observation $KL\,\boldsymbol{\Sigma}_{\text{Oracle}}\sim InvWish_N(\boldsymbol{\Sigma}_x,KL-1)$, we derive the variance of $(\boldsymbol{\Sigma}_{\text{Oracle}})_{i,j}$:

$$Var\left((\boldsymbol{\Sigma}_{\text{Oracle}})_{i,j}\right)=\delta_{i,j}^2=\frac{(KL-1)\left((\boldsymbol{\Sigma}_x)_{i,j}^2+(\boldsymbol{\Sigma}_x)_{i,i}(\boldsymbol{\Sigma}_x)_{j,j}\right)}{K^2L^2}. \tag{49}$$

Combining *Equation 45*, *Equation 48* and *Equation 49*, we express the upper bound on the conditional variance of $(\widetilde{\boldsymbol{\Sigma}}_x)_{i,j}$ as:

$$\sqrt{Var_{\mathscr{A}_W}\left((\widetilde{\boldsymbol{\Sigma}}_x)_{i,j}\right)}\leq\frac{1}{\sqrt{1-\eta}}\delta_{i,j}+\frac{8\sqrt{2}g(B)}{\sqrt{W}}(B+\mu_m)\left(\sigma_w\sqrt{1+\alpha^2}+\frac{1}{2}\right)+\mathcal{O}\left(\frac{g(B)^2}{W}\right).$$

Then, following *Equation 17* and the conditions in *Equation 29*, we conclude the proof of the upper bound on the conditional variance of $(\widehat{\boldsymbol{\Sigma}}_x)_{i,j}$:

$$\sqrt{Var_{\mathscr{A}_W}\left((\widehat{\boldsymbol{\Sigma}}_x)_{i,j}\right)}\leq\frac{1}{\sqrt{1-\eta}}\delta_{i,j}+\sqrt{\frac{\log W}{W^{1-2\epsilon}}}C_4\left(2\sigma_w\sqrt{1+\alpha^2}+1\right)+\mathcal{O}(\sigma_w^2)+\mathcal{O}\left(\frac{1}{W^{1-2\epsilon}}\right),$$

where $C_4:=384\sigma_m\sqrt{q}$.

Finally, it only remains to prove that the event $\mathscr{A}_W$ occurs with high probability for sufficiently large $W$:

Lemma 3. The probability of occurrence of the event

$$\mathscr{A}_W=\left\{\left|x_{W_k,l}^{(j)}\right|\leq B\text{ and }\frac{1}{2(1+\exp(B))}\leq\bar{n}_{W_k,l}^{(j)}\leq1-\frac{1}{2(1+\exp(B))}\text{ for }j=1,\cdots,N,k=1,\cdots,K\text{ and }l=1,\cdots,L\right\}$$

is upper-bounded as follows:

$$\mathbb{P}(\mathscr{A}_W)\geq1-\eta,$$

for some constant $0<\eta\leq1/2$ satisfying the conditions of *Equation (29)*.

Proof of Lemma 3.

First, using the union bound, we have:

$$\mathbb{P}(\mathscr{A}_W)\geq1-\sum_{k,l,j=1}^{K,L,N}\left\{\mathbb{P}\left(\left|x_{W_k,l}^{(j)}\right|>B\right)+\mathbb{P}\left(\bar{n}_{W_k,l}^{(j)}<\frac{1}{2(1+\exp(B))}\right)+\mathbb{P}\left(\bar{n}_{W_k,l}^{(j)}>1-\frac{1}{2(1+\exp(B))}\right)\right\}. \tag{50}$$

Next, we bound the probabilities on the right hand side using Chernoff's inequality (*Boucheron et al., 2013*). First, note that:

$$\mathbb{P}\left(x_{W_k,l}^{(j)}>B\right) = \mathbb{P}\left(x_{W_k,l}^{(j)}-\mu_x^{(j)}-s_{W_k}d_j>B-\mu_x^{(j)}-s_{W_k}d_j\right)$$
$$\overset{(p)}{\leq}\mathbb{P}\left(x_{W_k,l}^{(j)}-\mu_x^{(j)}-s_{W_k}d_j>\frac{B}{2}\right)$$
$$\overset{(q)}{\leq}\exp\left(-\frac{B^2}{8\sigma_m^2}\right),$$

where $(p)$ follows if $B>2\left(\|\boldsymbol{\mu}_x\|_\infty+\max_{k,j}\{|s_{W_k}d_j|\}\right)$ (which will hold under the conditions in *Equation 29*) and $(q)$ has been derived by applying the Chernoff's bound on the Gaussian random variable $x_{W_k,l}^{(j)}$. From the same reasoning, we see that $\mathbb{P}\left(x_{W_k,l}^{(j)}<-B\right)\leq\exp\left(-\frac{B^2}{8\sigma_m^2}\right)$. Combining these two results, we get the upper bound:

$$\mathbb{P}\left(\left|x_{W_k,l}^{(j)}\right|>B\right)\leq 2\exp\left(-\frac{B^2}{8\sigma_m^2}\right). \tag{51}$$

Next, note that:

$$\mathbb{P}\left(\bar{n}_{W_k,l}^{(j)}<\frac{1}{2(1+\exp(B))}\right) \overset{(r)}{\leq}\mathbb{P}\left(\bar{n}_{W_k,l}^{(j)}-\phi\left(x_{W_k,l}^{(j)}\right)<\frac{-1}{2(1+\exp(B))}\right)$$
$$\overset{(s)}{\leq}\exp\left(-\frac{W}{16(1+\exp(B))^2}\right), \tag{52}$$

where $(r)$ follows from the observation $\frac{1}{1+\exp(B)}<\phi\left(x_{W_k,l}^{(j)}\right)$ (which is a consequence of $|x_{W_k,l}^{(j)}|<B$). Then, we note that the zero-mean random variable $\bar{n}_{W_k,l}^{(j)}-\phi\left(x_{W_k,l}^{(j)}\right)$ is sub-Gaussian with variance factor $\frac{2}{W}$. Thus, using the Chernoff's inequality on sub-Gaussian random variables (*Boucheron et al., 2013*), we derive the upper-bound $(s)$ in *Equation 52*. In a similar fashion, based on the observation $\phi\left(x_{W_k,l}^{(j)}\right)<1-\frac{1}{1+\exp(B)}$, we conclude the bound:

$$\mathbb{P}\left(\bar{n}_{W_k,l}^{(j)}>1-\frac{1}{2(1+\exp(B))}\right)\leq\exp\left(-\frac{W}{16(1+\exp(B))^2}\right). \tag{53}$$

By combining the bounds in *Equation 51*, *Equation 52,* and *Equation 53*, the upper bound on $\mathbb{P}(\mathscr{A}_W)$ in *Equation 50* takes the form:

$$\mathbb{P}(\mathscr{A}_W)\geq 1-2KLN\exp\left(-\frac{W}{16(1+\exp(B))^2}\right)-2KLN\exp\left(-\frac{B^2}{8\sigma_m^2}\right).$$

Finally, under the assumptions in *Equation 29*, we further simplify this bound as:

$$\mathbb{P}(\mathscr{A}_W)\geq 1-2KLN\exp\left(-\frac{W^{1-\epsilon}}{64}\right)-\frac{2KLN}{W^q}\geq 1-\frac{4KLN}{W^q},$$

where we have used $W\geq 2$ (which gives $\log W\geq 2\log\log W$) and $\log W\geq\frac{2\log(64q)}{1-2\epsilon}$ to show that $\frac{W^{1-\epsilon}}{64}\geq q\log W$. Thus, $\log W\geq\frac{\log(8KLN/\eta)}{q}$ ensures that $\mathbb{P}(\mathscr{A}_W)\geq 1-\eta$, for $0<\eta\leq\frac{1}{2}$. ∎

This concludes the proof of Theorem 1.∎

## Appendix 3

## Adapting the proposed signal and noise correlation estimates to spiking observations

While Algorithm 1 takes two-photon fluorescence observations as input and produces estimates of signal and noise correlation as output, it is possible to adapt it to spiking observations obtained by electrophysiology recordings. The resulting algorithm is obtained by simplifying the variational inference procedure in Algorithm 1 and is given below for completeness:

---

**Algorithm 2 Estimation of $\Sigma_x$ and $\mathbf{D}$ from spiking observations**

---

**Inputs:** Ensemble of spiking observations $\left\{\mathbf{n}_{t,l}\right\}_{t,l=1}^{T,L}$, constant $\boldsymbol{\mu}_x$, hyper-parameters $\boldsymbol{\psi}_x$ and $\rho_x$, tolerance at convergence $\delta$ and the external stimulus $\mathbf{s}_t$

**Outputs:** $\widehat{\boldsymbol{\Sigma}}_x$ and $\widehat{\mathbf{D}}$

**Initialization:** Initial choice of $\mathbf{P}_x$, $\widetilde{\boldsymbol{\Omega}}_t$, $\widehat{\boldsymbol{\Sigma}}_x$ and $\widehat{\mathbf{D}}$, residual $= 10\delta$, $\gamma_x = \rho_x + LT$

1. **while** residual $\geq \delta$ **do**

Update variational parameters

2. **for** $t = 1, ..., T$ and $l = 1, ..., L$ **do**

3. $\mathbf{Q}_{\mathbf{x}_{t,l}} = (\widetilde{\boldsymbol{\Omega}}_{t,l} + \gamma_x \mathbf{P}_x^{-1})^{-1}$

4. $\mathbf{m}_{\mathbf{x}_{t,l}} = \mathbf{Q}_{\mathbf{x}_{t,l}}(\mathbf{n}_{t,l} - \frac{1}{2}\mathbf{1} - \widetilde{\boldsymbol{\Omega}}_{t,l}\widehat{\mathbf{D}}^{\top}\mathbf{s}_t + \gamma_x \mathbf{P}_x^{-1}\boldsymbol{\mu}_x)$

5. **for** $j = 1, \cdots, N$ **do**

6. $c_{t,l}^{(j)} = \sqrt{\left(\mathbf{Q}_{\mathbf{x}_{t,l}}\right)_{j,j} + \left(m_{\mathbf{x}_{t,l}}^{(j)} + \widehat{\mathbf{d}}_j^{\top}\mathbf{s}_t\right)^2}$

7. $(\widetilde{\boldsymbol{\Omega}}_{t,l})_{j,j} := \frac{1}{2c_{t,l}^{(j)}}tanh\left(\frac{c_{t,l}^{(j)}}{2}\right)$

8. **end for**

9. **end for**

10. $\mathbf{P}_x := \boldsymbol{\psi}_x + \sum_{t,l=1}^{T,L}\left\{\mathbf{Q}_{\mathbf{x}_{t,l}} + \mathbf{m}_{\mathbf{x}_{t,l}}\mathbf{m}_{\mathbf{x}_{t,l}}^{\top} - \boldsymbol{\mu}_x\mathbf{m}_{\mathbf{x}_{t,l}}^{\top} - \mathbf{m}_{\mathbf{x}_{t,l}}\boldsymbol{\mu}_x^{\top} + \boldsymbol{\mu}_x\boldsymbol{\mu}_x^{\top}\right\}$

Update outputs and the convergence criterion

11. **for** $j = 1, \cdots, N$ **do**

12. $\widehat{\mathbf{d}}_j = \left(\sum_{t,l=1}^{T,L}\left((\widetilde{\boldsymbol{\Omega}}_{t,l})_{j,j}\mathbf{s}_t\mathbf{s}_t^{\top}\right)\right)^{-1}\left(\sum_{t,l=1}^{T,L}\left\{\left(n_{t,l}^{(j)} - \frac{1}{2}\right)\mathbf{s}_t - (\widetilde{\boldsymbol{\Omega}}_{t,l})_{j,j}m_{\mathbf{x}_{t,l}}^{(j)}\mathbf{s}_t\right\}\right)$

13. **end for**

14. $(\widehat{\mathbf{D}})_{prev} = \widehat{\mathbf{D}}$, $\widehat{\mathbf{D}} = \left[\widehat{\mathbf{d}}_1, \widehat{\mathbf{d}}_2, \cdots, \widehat{\mathbf{d}}_N\right]$

15. $(\widehat{\boldsymbol{\Sigma}}_x)_{prev} = \widehat{\boldsymbol{\Sigma}}_x$, $\widehat{\boldsymbol{\Sigma}}_x = \frac{\mathbf{P}_x}{\gamma_x + N + 1}$

16. residual $= \|(\widehat{\boldsymbol{\Sigma}}_x)_{prev} - \widehat{\boldsymbol{\Sigma}}_x\|_2/\|(\widehat{\boldsymbol{\Sigma}}_x)_{prev}\|_2 + \|(\widehat{\mathbf{D}})_{prev} - \widehat{\mathbf{D}}\|_2/\|(\widehat{\mathbf{D}})_{prev}\|_2$

17. **end while**

18. Return $\widehat{\boldsymbol{\Sigma}}_x$ and $\widehat{\mathbf{D}}$

---

