## [Decision Letter]

**Acceptance summary:**

This study establishes a new method for more precise estimation of pairwise noise and signal correlations in two-photon calcium imaging, by modeling generically the influence of calcium dynamics and subtracting the interaction between response signals and variability when the trial number is low. The accuracy of this new estimator is demonstrated here for the mouse auditory cortex, but this tool will find useful applications on a large diversity of datasets.

**Decision letter after peer review:**

Thank you for submitting your article "Direct Extraction of Signal and Noise Correlations from Two-Photon Calcium Imaging of Ensemble Neuronal Activity" for consideration by *eLife*. Your article has been reviewed by 2 peer reviewers, including Brice Bathellier as the Reviewing Editor and Reviewer #1. and the evaluation has been overseen by Barbara Shinn-Cunningham as the Senior Editor. The reviewers have discussed their reviews with one another, and the Reviewing Editor has drafted this to help you prepare a revised submission.

Essential revisions:

Both reviewers agree on the quality of the method proposed in this study and it potential for assessing correlations in two-photon calcium imaging data. However, two points should be carefully addressed according to the comments below. First the authors should better explain what aspect of your method is key for your approach to outperform others methods. Second, the authors should better explore the behavior of your method on data that does not fulfil the core assumptions you make (uncorrelated noise, LN response model).

1. Surrogate data use a measurement noise model that has no temporal correlations. This is not the case in real two-photon imaging data, which include both correlation-free noise (photon count statistics) and temporally (and spatially) correlated artefacts. As in Deneux et al. 2016 (cited in the paper), the authors should provide simulations with different types of noise and show how this affects their correlation estimates. Is it still more robust than more classical methods?

2. Another simplifying assumption used by the authors is their model of spiking activity which is composed of a linear receptive field followed by a non-linear mapping function. Several papers have shown that this only imperfectly models neural responses (e.g. to sound in auditory cortex). There are, in fact in real neural data, non-linearities that are more complex than what the mapping function can capture. How does this impact their estimate? The authors should simulate this with e.g. a multilayer network (two-layer linear non-linear cascade, deep net) or simulate neurons that respond to the quadratic sum of the output of several linear filter (see e.g. recent work by the Shamma and colleagues).

3. There is a lack of intuition about the key aspects in their approach that makes it overperform other methods. This should be introduced in the results and/or discussion to better guide and convince the reader.

It is crucial that any end-user be able to get a clear picture of the conditions under which the method can or cannot be applied before diving in. The fact that such an applicability domain is not well defined is a major concern. Notably, each Real Data Study presented in the paper uses a preliminary selection of "highly active cells" (1rst study: N = 16; 2nd study: N = 10; 3rd study: N~20 per field), as the authors succinctly discuss that performance is expected to degrade "in the regime of extremely low spiking rate and high observation noise" (l. 518-519). But no precise criteria are provided to specify what is meant by "highly active cells". On the other hand, the authors also assume that there is at most one spiking event per time frame for each neuron, which seems to exclude bursting neurons. The latter assumption seems to be a challenge with respect to the example traces shown on Figure 4C (∆F/F reaches 400%) and on Figure 6C (∆F/F reaches 100%), considering that the GCaMP6s signal for a single spike is expected to peak below 10-20%. This forces the authors to take a scaling factor of the observations A = 1 x I (Real Data Study 1 and 3) or A = 0.75 x I (Real Data Study 2) compared to the A = 0.1 x I taken in the Simulation Studies. Therefore, it looks like if the Real Data Studies were performed on mainly bursting cells and each burst was counted as one spiking event. A detailed discussion of the usable range of firing rates, whether in spike or burst units, as well as the usable range of SNR should be added to the main text to allow future users to assess the suitability of their data for this analysis.

4. Another parameter seems to be set by the authors on a criterion that is unclear to me: the number of time lags R to be included in the sound stimulus vector st. It seems to act as a memory of the past trajectory of the stimulus and probably serves to enhance the effect of stimulus onset/offset relative to the rest of the sound presentation. It is consistent with the known tendency of neurons in the primary auditory cortex to respond to these abrupt changes in sound power. However, this R is set at 2 in the Simulation Study 1, whereas it is set at 25, in the Real Data Studies 1 and 3, and to 40 in the Real Data Study 2. What leads to these differences escaped to me and should be explained more clearly.

5. This memory of the past stimulus trajectory appears to be specific to the proposed method and is not accounted for in the 2-stage Pearson estimation, for example. Since it probably helps to reflect the common sensitivity of neurons to onset/offset, it alone provides an advantage to the proposed method over the 2-stage Pearson estimation. It would be instructive to also perform this comparison with R set to 1 to get an idea of the magnitude of this advantage.

*Reviewer #1 (Recommendations for the authors):*

1. Surrogate data use a measurement noise model that has no temporal correlations. This is not the case in real two-photon imaging data, which include both correlation-free noise (photon count statistics) and temporally (and spatially) correlated artefacts. As in Deneux et al. 2016 (cited in the paper), the authors should provide simulations with different types of noise and show how this affects their correlation estimates. Is it still more robust than more classical methods?

2. Another simplifying assumption used by the authors is their model of spiking activity which is composed of a linear receptive field followed by a non-linear mapping function. Several papers have shown that this only imperfectly models neural responses (e.g. to sound in auditory cortex). There are, in fact in real neural data, non-linearities that are more complex than what the mapping function can capture. How does this impact their estimate? The authors should simulate this with e.g. a multilayer network (two-layer linear non-linear cascade, deep net) or simulate neurons that respond to the quadratic sum of the output of several linear filter (see e.g. recent work by the Shamma and colleagues).

3. There is a lack of intuition about the key aspects in their approach that makes it overperform other methods. This should be introduced in the results and/or discussion to better guide and convince the reader.

---

## [Author Response]

Essential revisions:Both reviewers agree on the quality of the method proposed in this study and it potential for assessing correlations in two-photon calcium imaging data. However, two points should be carefully addressed according to the comments below. First the authors should better explain what aspect of your method is key for your approach to outperform others methods. Second, the authors should better explore the behavior of your method on data that does not fulfil the core assumptions you make (uncorrelated noise, LN response model).

We would like to thank the reviewing editor for his supportive stance towards our work, and for clearly summarizing the feedback from both reviewers. Based on the comments and suggestions of the two reviewers, we have made several changes to our manuscript that substantially and constructively addresses both of these key points, as we describe next.

1. Surrogate data use a measurement noise model that has no temporal correlations. This is not the case in real two-photon imaging data, which include both correlation-free noise (photon count statistics) and temporally (and spatially) correlated artefacts. As in Deneux et al. 2016 (cited in the paper), the authors should provide simulations with different types of noise and show how this affects their correlation estimates. Is it still more robust than more classical methods?

To address this comment, we performed extensive simulations to evaluate the robustness of different algorithms under model mismatch conditions induced by temporal correlations of observation noise. These new analyses are included in a new subsection called “Analysis of Robustness with respect to Modeling Assumptions” (Pages 6-7).

2. Another simplifying assumption used by the authors is their model of spiking activity which is composed of a linear receptive field followed by a non-linear mapping function. Several papers have shown that this only imperfectly models neural responses (e.g. to sound in auditory cortex). There are, in fact in real neural data, non-linearities that are more complex than what the mapping function can capture. How does this impact their estimate? The authors should simulate this with e.g. a multilayer network (two-layer linear non-linear cascade, deep net) or simulate neurons that respond to the quadratic sum of the output of several linear filter (see e.g. recent work by the Shamma and colleagues).

To address this comment, we performed a new simulation study to evaluate the robustness of different algorithms under model mismatch conditions induced by non-linear stimulus integration model. This result is included in a new subsection called “Analysis of Robustness with respect to Modeling Assumptions” (Pages 6-7).

3. There is a lack of intuition about the key aspects in their approach that makes it overperform other methods. This should be introduced in the results and/or discussion to better guide and convince the reader.It is crucial that any end-user be able to get a clear picture of the conditions under which the method can or cannot be applied before diving in. The fact that such an applicability domain is not well defined is a major concern. Notably, each Real Data Study presented in the paper uses a preliminary selection of "highly active cells" (1rst study: N = 16; 2nd study: N = 10; 3rd study: N~20 per field), as the authors succinctly discuss that performance is expected to degrade "in the regime of extremely low spiking rate and high observation noise" (l. 518-519). But no precise criteria are provided to specify what is meant by "highly active cells". On the other hand, the authors also assume that there is at most one spiking event per time frame for each neuron, which seems to exclude bursting neurons. The latter assumption seems to be a challenge with respect to the example traces shown on Figure 4C (∆F/F reaches 400%) and on Figure 6C (∆F/F reaches 100%), considering that the GCaMP6s signal for a single spike is expected to peak below 10-20%. This forces the authors to take a scaling factor of the observations A = 1 x I (Real Data Study 1 and 3) or A = 0.75 x I (Real Data Study 2) compared to the A = 0.1 x I taken in the Simulation Studies. Therefore, it looks like if the Real Data Studies were performed on mainly bursting cells and each burst was counted as one spiking event. A detailed discussion of the usable range of firing rates, whether in spike or burst units, as well as the usable range of SNR should be added to the main text to allow future users to assess the suitability of their data for this analysis.

To address this comment, we have now included the sources of performance gap between our proposed method and existing ones in the revised Discussion section, highlighting the key aspects of our method that makes it outperform existing approaches (Pages 17-18). We have also added a new subsection to Methods called “Guidelines for model parameter settings” that includes our rationale and criteria for choosing the number of neurons (N), stimulus integration window length (R), observation noise covariance (Σw), scaling matrix A, state transition parameter (α), and mean of the latent noise process (μx) (Page 24). Finally, we have performed new simulation studies to evaluate the effects of SNR and firing rate on the performance of the proposed method (Pages 6-7), and closely inspected the performance of our method under rapid increase of firing rate (Page 10).

4. Another parameter seems to be set by the authors on a criterion that is unclear to me: the number of time lags R to be included in the sound stimulus vector st. It seems to act as a memory of the past trajectory of the stimulus and probably serves to enhance the effect of stimulus onset/offset relative to the rest of the sound presentation. It is consistent with the known tendency of neurons in the primary auditory cortex to respond to these abrupt changes in sound power. However, this R is set at 2 in the Simulation Study 1, whereas it is set at 25, in the Real Data Studies 1 and 3, and to 40 in the Real Data Study 2. What leads to these differences escaped to me and should be explained more clearly.

To address this comment, we have added a new subsection to Methods called “Guidelines for model parameter settings” that includes our rationale for choosing the stimulus integration window length R (Page 24) and have performed a new analysis to evaluate the effect of R on the performance of the proposed method in real data study 1 (Page 10).

5. This memory of the past stimulus trajectory appears to be specific to the proposed method and is not accounted for in the 2-stage Pearson estimation, for example. Since it probably helps to reflect the common sensitivity of neurons to onset/offset, it alone provides an advantage to the proposed method over the 2-stage Pearson estimation. It would be instructive to also perform this comparison with R set to 1 to get an idea of the magnitude of this advantage.

To address this comment, we have now discussed the advantage of including the stimulus history in our model and probed the sensitivity of our estimates to the choice of R (including R = 1) in Figure 4—figure supplement 1 (Page 10).

Reviewer #1 (Recommendations for the authors):1. Surrogate data use a measurement noise model that has no temporal correlations. This is not the case in real two-photon imaging data, which include both correlation-free noise (photon count statistics) and temporally (and spatially) correlated artefacts. As in Deneux et al. 2016 (cited in the paper), the authors should provide simulations with different types of noise and show how this affects their correlation estimates. Is it still more robust than more classical methods?

Thank you for this suggestion. As explained earlier in response to the public portion of Reviewer 1’s comments, motivated by this suggestion, we have substantially enhanced our performance analyses in the revised manuscript and compiled them in a new subsection titled “Analysis of Robustness with respect to Modeling Assumptions” for better clarity and consistency.

Specifically, we considered two observation noise model mismatch conditions, namely, white noise + low frequency drift and pink noise, similar to the treatment in Deneux et al. (2016). For each noise mismatch model, we also varied the SNR level and firing rate and compared the performance of the different algorithms as reported in Figure 2—figure supplement 6. These new analyses demonstrate that our proposed estimates outperform the existing methods, under correlated generative noise models, and also with respect to varying levels of SNR and firing rate. As clearly evident in panels C and F of Figure 2—figure supplement 6, even though the estimated calcium concentrations are contaminated by the temporally correlated fluctuations in observation noise, the putative spikes estimated as a byproduct of our iterative method closely match the ground truth spikes, which in turn results in accurate estimates of signal and noise correlations.

To address this comment, we performed extensive simulations to evaluate the robustness of different algorithms under model mismatch conditions induced by temporal correlations of observation noise. These new analyses are included in a new subsection called “Analysis of Robustness with respect to Modeling Assumptions” (Pages 6-7).

2. Another simplifying assumption used by the authors is their model of spiking activity which is composed of a linear receptive field followed by a non-linear mapping function. Several papers have shown that this only imperfectly models neural responses (e.g. to sound in auditory cortex). There are, in fact in real neural data, non-linearities that are more complex than what the mapping function can capture. How does this impact their estimate? The authors should simulate this with e.g. a multilayer network (two-layer linear non-linear cascade, deep net) or simulate neurons that respond to the quadratic sum of the output of several linear filter (see e.g. recent work by the Shamma and colleagues).nt,l(j)∼Bernoulli(logistic(xt,l(j)+dj⊤st+(d~j,1⊤st)2+(d~j,2⊤st)2))

Thank you for this suggestion. As explained earlier in response to the public portion of Reviewer 1’s comments, we have addressed this comment in the revised manuscript in the new subsection titled “Analysis of Robustness with respect to Modeling Assumptions”. To examine the robustness of our method with respect to model mismatch in stimulus integration, as suggested, we generated data according to a non-linear (i.e., quadratic sum of linear filters) receptive field model:but assumed a linear stimulus integration model in our inference procedure (i.e., d~j,1=d~j,2=0). The comparison of the correlations estimated under this setting by each method are shown in Figure 2 —figure supplement 3. While the performance of our proposed signal correlation estimates under this setting degrade as compared to that in Figure 2 with no model mismatch, our proposed estimates still outperform the other methods and recovers the ground truth signal correlation structure reasonably well.

It is noteworthy that the model mismatch in the stimulus integration component does not affect the accuracy of noise correlation estimates in our method, as is evident from the noise correlation estimates in Figure 2—figure supplement 3. In comparison, the biases induced in the other methods due to model mismatch and various other factors such as observation noise, temporal blurring, undermining non-linear mappings between spikes and underlying covariates, results in significantly larger errors in both signal and noise correlation estimates.

Finally, we would like to note that since our model is a baseline framework for signal and noise correlation estimation from two-photon imaging data, it has the inherent limitations of the underlying modeling assumptions such as linear receptive fields and binary spiking. However, it is possible to generalize our approach beyond linear stimulus integration by extending to non-linear models such as those parameterized by neural networks, as we have also outlined towards the end of the Discussion.

To address this comment, we performed a new simulation study to evaluate the robustness of different algorithms under model mismatch conditions induced by non-linear stimulus integration model. This result is included in a new subsection called “Analysis of Robustness with respect to Modeling Assumptions” (Pages 6-7).

3. There is a lack of intuition about the key aspects in their approach that makes it overperform other methods. This should be introduced in the results and/or discussion to better guide and convince the reader.

Thank you for pointing out this source of ambiguity. The two main sources for the observed performance gap between our proposed method and existing approaches can be summarized as follows:

1) Favorable *soft* decisions on the timing of spikes achieved by our method, as a byproduct of the iterative variational inference procedure: an accurate probabilistic decoding of spikes results in better estimates of the signal/noise correlations, and conversely having more accurate estimates of the signal/noise covariances improves the probabilistic characterization of spiking events. This is in contrast with both the Pearson and Two-Stage methods: in the Pearson method, spike timing is heavily blurred by the calcium decay; in the two-stage methods, erroneous *hard* (i.e., binary) decisions on the timing of spiking events result in biases that propagate to and contaminate the downstream signal and noise correlation estimation and thus result in significant errors.

2) Explicit modeling of the non-linear mapping from stimulus and latent noise covariates to spiking through a canonical point process model (which is in turn tied to a two-photon observation model in a multi-tier Bayesian fashion) results in robust performance under limited number of trials and observation duration. As we have shown in Appendix 1, as the number of trials L and trial duration T tend to infinity, conventional notions of signal and noise correlation indeed recover the ground truth signal and noise correlations, as the biases induced by non-linearities average out across trial repetitions. However, as shown in Figure 2—figure supplement 2, in order to achieve comparable performance to our method using 20 trials, the conventional correlation estimates require ~1000 trials.

To address this comment, we have now included the aforementioned sources of performance gap in the revised Discussion section, highlighting the key aspects of our method that makes it outperform existing approaches (Pages 17-18).